# K&L: Penetrating Backdoor Defense with Key and Locks

## Abstract

Backdoor attacks in machine learning create hidden vulnerability by manipulating the model behaviour with specific triggers. Such attacks often remain unnoticed as the model operates as expected for normal input. Thus, it is imperative to understand the intricate mechanism of backdoor attacks. To address this challenge, in this work, we introduce three key requirements that a backdoor attack must meet. Moreover, we note that current backdoor attack algorithms, whether employing fixed or input-dependent triggers, exhibit a high binding with model parameters, rendering them easier to defend against. To tackle this issue, we propose the Key-Locks algorithm, which separates the backdoor attack process into embedding locks and employing a key for unlocking. This method enables the adjustment of unlocking levels to counteract diverse defense mechanisms. Extensive experiments are conducted to evaluate the effective of our proposed algorithm. Our code is available at: https://anonymous.4open.science/r/KeyLocks-FD85

## 1 Introduction

Deep neural networks (DNNs) require extensive data and training resources for a decent performance, which are not available for most companies (Jeon et al., 2019). As a result, many organizations resort to outsourcing model training or using public models (Chen et al., 2022). However, verifying a model as free from backdoor attacks with stealthy triggers is challenging. These triggers remain dormant, allowing the model to perform normally, but causing it for incorrect results when activated. Based on (Gu et al., 2017; Chen et al., 2017; Nguyen & Tran, 2021; Wang et al., 2022a; Nguyen & Tran, 2020; Li et al., 2021b), in this paper, we first systematically delineate three requirements for a successful backdoor attack:

***Requirement 1:*** An effective backdoor trigger cannot affect the semantic information of the original image.

***Requirement 2:*** The trigger should be able to manipulate the model for incorrect outputs.

***Requirement 3:*** Victim model must operate normally in the absence of the backdoor trigger.

Requirement 1 emphasises the backdoor stealth, as a big disruption of the the semantic information would make it more vulnerable to defense. Requirements 2 and 3 ensure the trigger's efficacy and prevent the backdoor from interfering with regular task execution. This is crucial for maintaining the concealment. Backdoor attack methods should meet three requirements without compromising model integrity. We further analyze how these requirements influence the defense process in Section 3.2.

Normally the backdoor attack algorithms are depicted as fixed trigger backdoors (Gu et al., 2017; Chen et al., 2017; Wang et al., 2022a; Nguyen & Tran, 2021) and input-dependent backdoors (Nguyen & Tran, 2020; Li et al., 2021b). In Figure 1, we note that fixed trigger backdoors exhibit high binding with the model parameters. Specifically, to satisfy the three *requirements*, the parameters responsible for activating these backdoors are sensitive and fragile, making them susceptible to existing defense algorithms. Although the input-dependent backdoor algorithms can adaptively generate triggers as needed, they are also highly bound to the generator parameters. Interestingly, it is this high binding nature that is crucial for the success of backdoor defenses. Defenders can weaken or eliminate the backdoor effect by adjusting model parameters or generator parameters, thereby reducing the success rate of backdoor attacks.

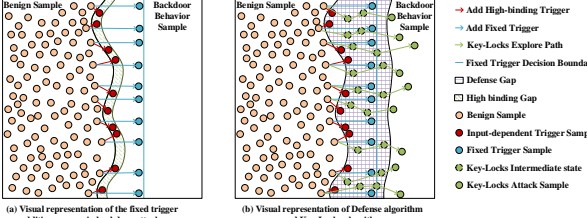

Figure 1: Schematic illustration of different backdoor attack and defense processes. Subfigure (a) delineates the process of fixed trigger integration (blue points) in techniques like BadNet (Gu et al., 2017). This method essentially constructs an additional decision boundary that must be wholly encoded within the model's parameters. (b) visualizes defense strategies (green points) such as ANP (Wu & Wang, 2021) that manipulate the decision boundary to form a 'defense gap', high binding attack samples will be positioned within the defense gap, thereby facilitating the defense against backdoor attacks. The two green dots in Subfigure (b) are intended to represent different states: the dotted green dot indicates intermediate attack samples that are still in the process of iterating, while the solid green dot is the final attack sample that has successfully penetrated the defense gap. In contrast, our Key-Locks algorithm (green points) introduces various unlocking levels to mitigate the issue of high binding, thereby enhancing the attack against different defensive shifts.

Generally, backdoor attacks incorporate a backdoor into the model, tightly coupled with a specific trigger. This tight coupling, referred to as high binding, is a primary factor exposing backdoor attacks to defensive measures. A detailed discussion on the association between the three requirements of backdoor attacks and high binding, as well as the reason why such design leads to high binding, is provided in Section 3.2. High binding implies that changes in the model parameters or the backdoor's operational context can render the trigger ineffective.

To circumvent the limitations imposed by high binding, we introduce a more flexible and less detectable approach that involves embedding a component within the model that is responsive to a broad range of trigger conditions (locks). We use **Appendix A** to show the structure of our approach. Furthermore, a mechanism capable of generating a range of triggers (keys) is developed. These triggers are designed to interact with the embedded component, effectively triggering the backdoor across various scenarios. The operational details of the backdoor are primarily encoded within these triggers. Consequently, the embedded component's sole function is to respond to the presence of trigger information, without the need to store or memorize specific backdoor details. Therefore, compared to other backdoor attack strategies, our method offers enhanced evasion of existing defense mechanisms.

Overall, our contributions are: •We summarise three *requirements* for backdoor attacks, elucidate the high binding nature between backdoor attack algorithms and model parameters. We analyse how backdoor defense leverages high binding property. •Our Key-Locks (K&L) algorithm decouples the attack algorithm from model parameters, successfully penetrating nearly all existing backdoor defense mechanisms. • We introduce a new metric: the Accuracy-ASR Curve (AAC). Extensive experiments validate the performance of the K&L algorithm, demonstrating its effectiveness against nearly all current backdoor defense strategies.

## 2 RELATED WORK

### 2.1 BACKDOOR ATTACKS

We firstly discuss existing backdoor attack methods that compromise DNNs by inserting stealthy triggers or altering training data, leading to incorrect outputs under specific conditions. We feature several methods such as BadNet (Gu et al., 2017), Blended (Chen et al., 2017), WaNet (Nguyen & Tran, 2021), Bit-per-pixel (Bpp) (Wang et al., 2022a), Input-Aware (Nguyen & Tran, 2020), and Sample-specific Backdoor Attack (SSBA) (Li et al., 2021b).

BadNet introduces malicious behavior by inserting backdoor samples during training, such as a specific pattern on an object. Blended seamlessly integrates a key pattern into inputs, creating poisoned samples by subtly blending the pattern. The key pattern is intensified as a stronger presence associated with a higher likelihood to trigger the backdoor. WaNet uses image warping to inject backdoors, making the images more natural and harder to detect. We will analyse how the fixed-trigger methods result in a high binding with the model parameters in Section 3.2. Bpp attack employs image quantization and dithering to create stealthy Trojan triggers, which reduces the color palette and uses contrastive learning and adversarial training to inject the Trojan. The model parameters must remember the attack pattern. Input-Aware attack generate unique triggers for each input, however the generator is highly related to the model parameters. SSBA embeds attack strings into benign images to create imperceptible noise as backdoor triggers, bypassing different backdoor defenses.

## 2.2 Backdoor Defenses

Here we review classic and state-of-the-art defense algorithms that mitigate backdoor threats, starting with testing-time defenses such as Strong Intentional Perturbation (STRIP). STRIP (Gao et al., 2019) defends against backdoor attacks by perturbing input images with a set of clean images and monitoring the entropy of prediction outputs. High entropy in predictions indicates a robust response to potential backdoor triggers, making STRIP an effective preliminary defense mechanism. Following this, we delve into strategies from neuron pruning to advanced techniques like attention distillation. Adversarial Neuron Pruning (ANP) (Wu & Wang, 2021) prunes sensitive neurons directly, avoiding extensive retraining and requiring minimal data. Similarly, Batch Normalization Statistics-based Pruning (BNP) (Zheng et al., 2022b) considers the altered BN layer statistics, utilizing divergence for pruning. Channel Lipschitzness based Pruning (CLP) (Zheng et al., 2022a) uses Channel Lipschitz Constant as a metric to identify and prune channels most affected by trigger patterns. Implicit Backdoor Adversarial Unlearning (I-BAU) (Zeng et al., 2021) removes embedded triggers using a minimax optimization with implicit hypergradients, streamlining the unlearning process. Neural Attention Distillation (NAD) (Li et al., 2021a) realigns the compromised network's attention through a teacher-student paradigm, akin to knowledge distillation. These methods collectively form a defense ecosystem that addresses different aspects of the backdoor threat, from direct intervention to subtle attention realignment, catering to a variety of defense scenarios.

Enhancing the credibility of AI systems through interpretability tools can also detect backdoor attacks (Selvaraju et al., 2017; Zhu et al., 2023). This paper employs the Boundary-based Integrated Gradient (BIG) (Wang et al., 2022b), an attribution interpretability tool that aggregates gradients along a path from input to the nearest decision boundary, leveraging local boundary information for more precise explanations. BIG is used here to investigate the detectability of backdoor samples generated by various attack techniques.

## 3 Method

### 3.1 Problem Definition

A backdoor attack on a neural network (NN) classifier aims to create a new model $f(:; W)$ that behaves normally on the standard input distribution but misclassifies inputs that contain a specific pattern $\tau$ to a target class $c_t$. $D$ is the training dataset. The poisoned samples are created by applying a perturbation $\tau$ to a subset of $D$, yielding a poisoned dataset $B$. The model $f(:; W)$ is trained using a dataset $D_B$ which includes both clean and poisoned samples.

$f(x; W)$ represents a mapping $\mathbb{R}^n \to \mathbb{R}^c$ for an input $x$ to output over $C$ classes. For an input-dependent backdoor attack, the progress of adding a trigger is defined by a perturbation function $T_\tau : \mathbb{R}^n \to \mathbb{R}^n$ which embeds a backdoor trigger $\tau$ into the benign input $x$, and a target label $c_t$, by optimizing model parameters $W$:

$$f(T_\tau(x); W) = c_t \quad \forall x \in B, \tag{1}$$

$$f(x; W) \approx p(y|x) \quad \forall x \in D, \tag{2}$$

where $p(y|x)$ denotes the true probability distribution over labels given input $x$, and $c_t$ is the target label specified by the attacker. Eq. 1 embodies **Requirement 2**, ensuring that the backdoored model misclassifies any input containing the trigger $\tau$ to the target class $c_t$. Eq. 2 aligns with **Requirement**

**3**, asserting that the model's behavior on clean inputs $x$ approximates the true label distribution $p(y|x)$. **Requirement 1** stipulates that the benign input $x$ and its triggered counterpart $T_\tau(x)$ should share similar semantics, a condition that can be quantitatively expressed using the $L2$ norm such that $\|x - T_\tau(x)\|_2 < \epsilon$, where $\epsilon$ is a small constant. It ensures the stealth of the trigger, preventing easy detection while preserving the original semantics of the input as much as possible.

## 3.2 RESEARCH PROBLEM

**Q1: What makes the fixed-trigger backdoor susceptible to effective defense?**

Fixed trigger backdoor attacks typically involve inserting identical triggers into an image, and to maintain semantic similarity before and after the attack (**Requirement 1**), these triggers are often designed to be imperceptible, such as being extremely small or nearly transparent. Consequently, the model needs to meticulously memorize the characteristics of these triggers. Hence, any parameters changes can lead to a loss of memory regarding these triggers, effectively neutralizing the backdoor attack. This principle underlies the rationale of most backdoor defense mechanisms.

Under the premise that Requirement 3 is used to ensure normal model function, due to Requirements 1 and 2, these parameters must be highly sensitive and fragile. The model is expected to trigger with only a minimal presence of harmful features without impacting its semantic information. We interpret this sensitivity and fragility as **high binding** between the backdoor attack and model parameters. **Any trigger addition method that does not rely on model parameters is highly bound to the parameters**, the reason being that the model's parameters must memorize these triggers or the methods of adding these triggers. This high binding is precisely why fixed trigger methods are susceptible to data-free defense methods like Clip, which detects and eliminates parameters that deviate from normal, often characterized by their high sensitivity.

Additionally, defense methods like ANP, RNP, FP, FT, I-BAU, and NAD use clean datasets to fine-tune or distill the model. The conspicuous sensitivity of anomalous parameters after gradient descent engenders a shift in the original backdoor trigger conditions, rendering conventional backdoor attack methods useless.

**Q2: What factors contribute to the defensibility of current input-dependent backdoor attacks?**

Input-aware backdoor attacks serve as a paradigmatic instance of input-dependent backdoor strategies. This approach entails the concurrent training of a Generator alongside the primary model, with the objective of fabricating a distinct Trigger for each input sample. To comply with **Requirement 1**, which necessitates minimal perturbation to the original input features, the perturbation of the generated Trigger must be rigorously regulated. Fundamentally, the input-aware methodology transitions the binding from a fixed trigger and model parameters to a dynamic association between the model parameters and those of the Generator. For instance, a trigger generated by the Generator for a given input image is a one-off creation, thus establishing a pronounced binding between this trigger and the model parameters. Consequently, any modification to the model could potentially misalign the specifically tailored trigger, leading to the nullification of the backdoor, signifying that the Generator exhibits a **high binding** to the model parameters. The Bpp algorithm embodies input-dependence and **Requirement 1** through methods such as image quantization and dithering. Nonetheless, these techniques are inherently sensitive and decoupled from the model parameters, resonating with the high binding scenario posited in Question 1, where Trigger generation is not contingent on model parameters.

**Definition of high binding:** A backdoor attack method that exhibits high binding to model parameters must meet at least one of the conditions: 1. A fixed Trigger is generated. 2. The process of generating the Trigger is independent of the model's parameters. 3. The Trigger generation process is parametric and is trained in conjunction with the model.

**Q3: What properties should a backdoor attack possess to surpass current defensive strategies?**

(a)Initially, it is expected to conform to all requirements 1-3. (b) The approach must adhere to an Input-dependent condition while avoiding high binding to the model parameters. Specifically, the generation of Triggers should be parameter-dependent, with the Trigger addition method not resulting from joint training with the model.(c) Compromising the effectively embedded backdoor results in affecting the model's performance on standard tasks.

### 3.3 Key-Locks (K&L) Backdoor Attack

In order to attain characteristics impervious to the existing defense methods outlined aforementioned, we introduce a novel attack strategy, named the K&L backdoor attack algorithm. The K&L algorithm is divided into two principal components: *Embedding Locks* and *Use the Key to Open the Door*. Following sections will provide the details of the functions of these two components and their relationship to the properties discussed in Section 3.2.

The training for Embedding Locks represents a unique form of adversarial training employed to implant a backdoor in the model's parameters. The loss for Embedding Locks is composed of two parts: the maintain loss and the locks loss. The maintain loss ensures the model's performance on normal inputs, while the locks loss facilitates the embedding of the backdoor. This is formulated as:

$$L = \underbrace{L(x, y; W)}_{\text{maintain loss}} + \underbrace{L\left(x', c_t; W\right)}_{\text{locks loss}} \tag{3}$$

$$x' = \underbrace{x - \eta \cdot \text{sign}\left(\frac{\partial L(x, y; W)}{\partial x}\right)}_{\text{reserve keyhole}} \tag{4}$$

Eq. 3 represents the loss expression for Embedding Locks, where $L$ denotes the loss function, and $x'$ in Eq. 4 denotes the sample generation process representing Backdoor Behavior. Since both clean samples and backdoor samples are updated within the same loss function, there is no issue of imbalance between clean and backdoor samples during the training process. This process involves a single-step gradient descent towards the backdoor category. During training, detailed in Algorithm 1 lines 7, 10 and 11, Backdoor Behavior samples from each iteration are further descended based on the previous iteration, aiming to iteratively expand the range of Locks. After several iterations of Embedding Locks, we acquire new model parameters, denoted as $W'$, which act as the Key in our parameters. It is noted that, training typically occurs in a very high-dimensional yet confined part of the space, focusing on learning distribution within this minimal space. Samples outside this space are considered as Out of Distribution (OOD) samples. The purpose of generating $x'$ is to intentionally include the post-gradient descent samples as part of OOD samples that the original model did not learn, thereby enabling the model to interpret the space near $x$ and $x'$ as latent space. This makes it easy to achieve backdoor target label during the gradient descent process.

#### 3.3.1 Use the Key to Open the Door

Following the Embedding Locks process, we obtain new model parameters $W'$, which are utilized as the Key. The principles for generating backdoor samples using the Key are outlined in Eqs. 5 and 6:

$$grad = \frac{\partial L(x, c_t; W')}{\partial x} \tag{5}$$

$$x = x - \alpha \cdot sign(grad) \tag{6}$$

We control the image perturbation using clip function, $x = \text{clip}(x, \min, \max)$, where $\min = \max(x - \epsilon^l, 0)$ and $\max = \min(x + \epsilon^l, 1)$ if the valuable range of input features normalized to between 0-1. Here $\epsilon^l$ is the perturbation constant based on level $\epsilon^l = \frac{1}{255} \times level$.

The generation process employs gradient descent, altering the sample's category to the Backdoor target label under model parameters $W'$. During this process, we specify a *level* and its corresponding perturbation limit $\epsilon^l$, where *level* indicates the number of gradient descent iterations controlling the intensity of using the key to open the door. A higher *level* signifies stronger Backdoor capability but also implies greater perturbation. Due to the presence of Locks loss, our samples can easily transition to the target category during gradient descent. Therefore, we keep the *level* within a small range in our algorithm, ensuring that the image pixel value deviation is nearly imperceptible to the human eye.

#### 3.3.2 In-depth Analysis of Indefensibility

Next, we analyze why K&L algorithm satisfies the three properties discussed in Section 3.2.

*Property (a) Analysis*: Since the pixel value deviation generated by the K&L algorithm is very low, it meets the criterion of minimal disruption to original features. Moreover, due to the presence of

maintain loss, the category remains consistent with the original true category when not transitioning to a Backdoor example. Furthermore, as analyzed in Section 3.3.2, gradient descent can easily achieve the Backdoor category with the presence of locks loss.

*Property (b) Analysis*: The process of converting samples to Backdoor samples utilizes gradient descent, which employs model parameters but differs from the generator in Input-aware methods (Nguyen & Tran, 2020) in that it does not have parameters and does not require training. Also, due to the existence of different *levels*, the high binding relationship is dissolved. Even if defense methods use in-distribution (IDD) samples to modify the decision boundary, they cannot completely erase the multiple Embedding Locks processes, as this behavior lies in the OOD space.

*Property (c) Analysis*: During backdoor training, the use of Locks loss merely ensures that gradient descent can convert samples into the target label with lower perturbation, rather than relying solely on the latent space implanted by Locks loss. This means that defense algorithms using IDD samples for defense or CLP (Zheng et al., 2022a) for data-free defense to destroy the latent space actually shift the original IDD space significantly, impacting the normal functionality, which is an unacceptable cost for defense.

### 3.4 DIFFERENTIATING K&L FROM ADVERSARIAL ATTACKS

Adversarial attacks aim to perturb inputs in a manner that deceives the target model without altering its parameters. These attacks typically introduce minimal perturbations to the input data, denoted by $\delta$, leading to an adversarially modified input $x_{adv} = x + \delta$. The objective is to cause the model $f(\cdot; W)$ to misclassify $x_{adv}$, such that $f(x_{adv}; W) \neq y$, where $x$ is the original input, $W$ represents the model parameters, and $y$ is the true label.

In contrast, our approach involves embedding a backdoor into the model during the training phase by adjusting its parameters $W$ to induce a high binding phenomenon. This is achieved through a dual-objective optimization process, where the model is trained to minimize the loss on clean inputs while simultaneously ensuring that inputs containing a specific pattern or trigger $\tau$ are misclassified to a target class $c_t$. The optimization can be formalized as:

$$W^* = \arg\min_W L(x, y; W) + \lambda L(x', c_t; W), \tag{7}$$

where $L$ denotes the loss function, $\lambda$ is a regularization term that balances the performance on clean and poisoned inputs, and $x'$ represents the function embedding the backdoor trigger into a benign input $x$. The goal is to find optimal model parameters $W^*$ that ensure the model performs accurately on legitimate inputs while classifying inputs with trigger as the target class $c_t$.

Our method's distinction from adversarial attacks lies in its focus on **modifying the model parameters** during training to embed a backdoor, as opposed to crafting input perturbations at inference time. This high binding backdoor strategy not only renders the backdoor more difficult to detect and remove but also potentially increases the model's susceptibility to adversarial attacks, fundamentally altering the model's response to specific input patterns.

## 4 EXPERIMENTS

### 4.1 EXPERIMENTAL SETUP

**Datasets** To substantiate the efficacy of K&L algorithm, we employ four popular public datasets, namely CIFAR-10, CIFAR-100 (Krizhevsky et al., 2009), GTSRB (Houben et al., 2013), and Tiny ImageNet (Le & Yang, 2015).

**Evaluation Metrics and Parameters** In alignment with the prevailing standards in the domain (Gu et al., 2017; Chen et al., 2017; Nguyen & Tran, 2021; Wang et al., 2022a; Nguyen & Tran, 2020; Li et al., 2021b), We evaluated the K&L backdoor attack using Benign Accuracy (BA) and Attack Success Rate (ASR). We also introduce a new metric: the Accuracy-ASR Curve (AAC). Detailed introduction of metrics and the parameters in the comparison experiment can be found in the **Appendix D- E**.

**Models** Our empirical analysis utilizes three distinct neural network architectures: PreActResNet18 (He et al., 2016), VGG19 with Batch Normalization (VGG19-BN) (Simonyan & Zisserman,

Table 1: Comparison results of attack methods against various defense algorithms using PreActResNet18. The model's *original accuracy* on CIFAR-10, CIFAR-100, GTSRB, and Tiny ImageNet datasets are 94.08%, 70.72%, 98.27%, and 57.39%, respectively. This table presents a detailed comparison of several attack methods, including K&L (ours), across different datasets. The performance is evaluated in terms of Benign Accuracy (BA) and Attack Success Rate (ASR) under different defense mechanisms, including ANP, BNP, FP, FT, I-BAU, NAD, CLP, and RNP. Bold entries indicate a successful attack under the corresponding defense algorithm. Notably, only K&L method consistently penetrates defense across all scenarios.

| Datasets | Attack Methods | No Defense | ANP | BNP | FP | FT | I-BAU | NAD | CLP | RNP |
|---|---|---|---|---|---|---|---|---|---|---|
| | | BA/ASR | BA/ASR | BA/ASR | BA/ASR | BA/ASR | BA/ASR | BA/ASR | BA/ASR | BA/ASR |
| CIFAR-10 | BadNet | 91.82/93.79 | 83.55/0.0 | **91.72/94.34** | 91.91/0.9 | 90.34/1.6 | 84.58/4.49 | 88.82/1.96 | **91.45/94.46** | 91.88/5.34 |
| | Blended | 93.27/97.58 | 86.82/0.22 | **93.17/95.3** | 92.63/5.67 | **92.39/73.01** | 89.36/1.12 | 91.87/55.07 | **90.75/62.24** | **93.27/97.58** |
| | BppAttack | 91.39/99.19 | 85.1/0.16 | 90.88/3.83 | **93.32/50.4** | 93.3/2.7 | 90.7/16.59 | 93.11/2.31 | 90.02/2.79 | 91.42/8.68 |
| | Input-Aware | 89.79/93.71 | 86.72/0.54 | 90.02/1.36 | 93.15/10.64 | **93.06/52.36** | **89.35/38.69** | 92.84/5.81 | 90.23/2.49 | 90.31/1.7 |
| | SSBA | 93.34/100.0 | 89.53/0.1 | 93.2/8.46 | 92.61/38.93 | 92.79/7.0 | 87.59/1.3 | 92.11/6.49 | 92.68/1.26 | 93.01/0.68 |
| | WaNet | 90.57/96.93 | 81.68/0.22 | 47.03/84.07 | 93.27/0.84 | 93.16/6.01 | 91.79/10.29 | 92.81/3.41 | **88.99/66.82** | 90.67/1.47 |
| | K&L | 93.87/99.74 | **86.59/60.79** | **93.38/99.43** | **93.29/82.53** | **93.83/98.24** | **89.81/74.21** | **93.24/95.96** | **92.12/99.08** | **93.87/99.74** |
| CIFAR-100 | BadNet | 67.36/86.68 | 62.24/0.0 | **66.38/86.81** | 64.36/0.57 | 66.13/0.34 | 61.36/0.06 | 65.69/0.14 | **63.95/69.27** | 67.36/2.6 |
| | Blended | 69.07/96.73 | **65.48/64.3** | **68.77/96.21** | 62.77/7.69 | **68.04/86.46** | 62.66/1.12 | **67.87/87.59** | **63.61/77.1** | **69.07/96.73** |
| | BppAttack | 65.51/99.37 | 60.92/0.07 | 64.79/0.11 | 68.66/0.09 | 69.82/0.21 | 66.39/15.26 | 69.74/0.29 | 63.05/0.01 | **65.51/99.37** |
| | Input-Aware | 64.87/95.34 | 60.92/3.78 | 63.1/4.8 | 66.99/0.12 | 69.25/1.76 | **65.74/37.55** | 68.8/2.82 | **57.61/96.78** | **64.63/96.11** |
| | SSBA | 69.7/99.99 | 63.68/1.26 | 68.98/72.59 | 62.7/47.59 | 68.42/98.17 | **64.44/64.22** | **67.92/99.76** | **66.89/99.83** | 69.58/54.28 |
| | WaNet | 63.16/98.44 | 60.7/20.58 | **27.07/99.41** | 68.45/0.02 | 68.73/0.03 | 65.24/18.55 | 68.48/0.03 | 60.08/56.49 | **62.59/98.04** |
| | K&L | 69.3/99.97 | **66.34/83.36** | **68.87/99.93** | **66.19/71.02** | **69.73/99.25** | **64.64/53.69** | **69.67/99.65** | **67.88/99.77** | **69.3/99.97** |
| GTSRB | BadNet | 96.35/95.02 | 93.86/0.01 | **96.5/90.86** | 98.12/0.0 | **97.73/79.24** | 95.6/0.0 | **97.53/79.9** | 96.44/31.5 | **96.55/85.82** |
| | Blended | 98.38/99.75 | 95.1/0.0 | **98.01/99.78** | 98.37/53.39 | **98.5/98.73** | **93.97/58.0** | **98.22/99.35** | **98.3/99.67** | **98.38/99.75** |
| | BppAttack | 98.34/92.12 | 98.56/0.0 | 98.17/0.0 | **99.06/28.76** | 99.04/0.88 | 96.84/0.04 | 98.1/0.0 | 98.0/2.04 | 98.35/0 |
| | Input-Aware | 97.58/97.12 | 96.86/0.0 | 96.9/0.58 | 98.11/0.46 | **98.31/82.64** | 96.98/0.06 | **98.1/43.91** | **97.06/80.05** | 96.38/1.89 |
| | SSBA | 97.78/100.0 | 95.53/0.0 | **97.3/100.0** | 98.01/89.36 | **97.65/100.0** | 89.39/27.17 | **97.67/100.0** | **97.13/99.98** | 97.78/100 |
| | WaNet | 97.05/96.16 | 93.56/0.0 | **2.8/100.0** | 98.99/0.08 | 98.84/1.81 | 97.4/0.14 | 98.72/0.7 | **0.48/100.0** | 96.3/0.02 |
| | K&L | 97.66/99.85 | **89.78/2.03** | **97.39/88.93** | **98.27/44.36** | **98.16/88.22** | **92.03/39.96** | **98.08/92.87** | **97.57/99.53** | **97.66/99.85** |
| Tiny | BadNet | 55.94/99.95 | **55.19/1.59** | **55.94/99.96** | 50.52/0.56 | 55.1/0.1 | **52.61/97.56** | 48.16/0.19 | **55.94/99.95** | 55.94/0.48 |
| | Blended | 56.13/97.72 | **50.58/64.39** | **56.13/97.72** | 50.52/28.44 | **55.17/89.03** | **53.49/79.13** | **49.02/72.16** | **56.22/85.39** | 56.14/44.72 |
| | BppAttack | 58.29/99.99 | **55.38/99.56** | **57.51/99.95** | 50.55/0.39 | 57.7/0.23 | **55.91/80.06** | 49.49/0.32 | 57.73/0.21 | 58.18/0.1 |
| | Input-Aware | 57.51/99.44 | 53.35/0.3 | 56.9/0.0 | **57.42/99.47** | 52.15/0.16 | 57.14/0.39 | 49.54/0.34 | **57.4/24.31** | 57.41/0.2 |
| | SSBA | 55.45/99.89 | 49.96/1.61 | **55.45/99.89** | 50.11/79.16 | 54.56/4.57 | 48.06/0.4 | 47.0/1.61 | **55.45/99.89** | 55.03/1.26 |
| | WaNet | 57.9/95.36 | 55.01/0.08 | 57.52/1.16 | 50.45/0.69 | 57.05/0.28 | **54.07/18.0** | 48.31/0.57 | **57.46/72.37** | 57.82/0.1 |
| | K&L | 56.82/99.92 | **52.47/99.74** | **56.82/99.92** | **51.89/70.54** | **57.02/98.26** | **53.91/91.14** | **50.38/40.52** | **56.41/99.84** | **56.53/99.92** |

2014), MobileNet-v3-large (Howard et al., 2019), and EfficientNet-B3 (Tan & Le, 2019). Each architecture embodies a unique aspect of contemporary neural network design, offering a comprehensive platform for assessing the effectiveness of the K&L backdoor attack in varied image processing contexts. However, due to space constraints, we present the analysis results for PreActResNet18 within the main text. The experimental outcomes for the other three models are detailed in the **Appendix E**.

**Baselines** We included BppAttack, the state-of-the-art methodology in backdoor attacks, as the foundational baseline. Additionally, three other backdoor attack algorithms were deployed in a comparative capacity, specifically BadNet (Gu et al., 2017), Blended (Chen et al., 2017), Input-Aware (Nguyen & Tran, 2020), SSBA (Li et al., 2021b), and WaNet (Nguyen & Tran, 2021), each embodying a distinct attack strategy. To ensure a fair and uniform evaluation terrain, we implemented the assaults using the hyperparameters as specified in previous scholarly endeavors for all rival techniques (Wu et al., 2022).

**Defense Methods** To evaluate the K&L attack against existing defense mechanisms, a range of state-of-the-art defense algorithms were employed. It includes ANP (Wu & Wang, 2021), BNP (Zheng et al., 2022b), FP (Liu et al., 2018), standard fine-tuning (FT), I-BAU (Zeng et al., 2021), NAD (Li et al., 2021a), CLP (Zheng et al., 2022a), and RNP (Li et al., 2023). Each defense method was implemented using the default parameters as outlined in prior research (Wu et al., 2022). This selection encompasses a diverse array of approaches, from pruning and finetuning to more intricate adversarial and network distillation techniques, thereby providing a comprehensive evaluation of the K&L attack's effectiveness against current defensive strategies.

## 4.2 RESULTS

The analysis of the K&L method's performance, especially in the context of breaching defense algorithms, reveals its significant superiority over other attack methods across various datasets. Our method is particularly challenging to defend against, as evidenced by the fact that our K&L method

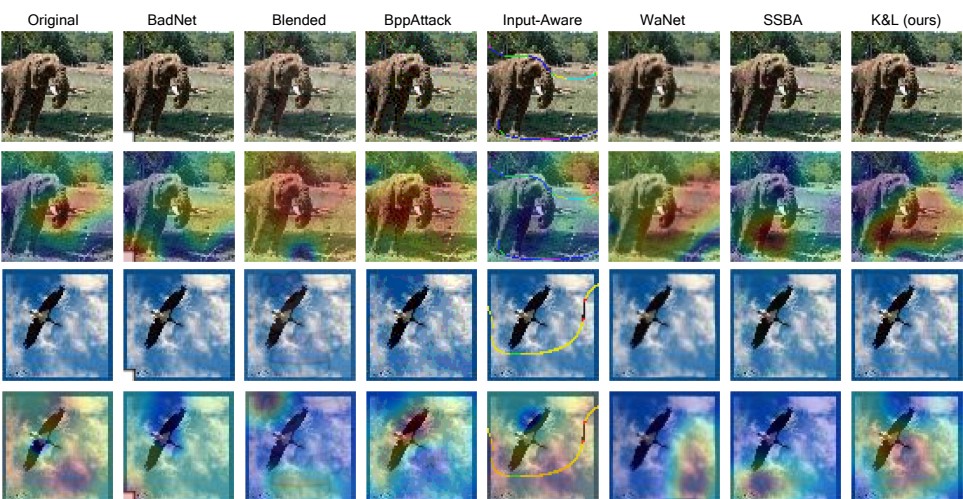

Figure 2: Attribution visualization comparing the stealthiness of our K&L method against other methods. Our method manages to hide the trigger close to the main features under BIG (Wang et al., 2022b), while other methods show clear feature shifts or diffusion across the image.

Table 2: Backdoor samples similarity rates

| Method | BadNet | Blended | Bpp | Input-Aware | WaNet | SSBA | K&L |
|---|---|---|---|---|---|---|---|
| Similarity Rate | 0.0442 | 0.1086 | 0.0879 | 0.0744 | 0.0006 | 0.0584 | 0.136 |

is the only one capable of **circumventing all tested defense mechanisms**. In contrast, methods like BadNet and Blended, while easy to implement, fail against certain defenses and produce samples with clear attribution shifts that are easily detected by post-hoc algorithms. Our K&L method, however, generates samples **without noticeable attribution shift**, making it significantly harder for detection algorithms to identify.

As shown in Table 1, in the context of ANP defense mechanism, K&L achieves a breakthrough performance. Specifically, on the CIFAR-10 dataset, it attains an ASR of 71.59%, significantly higher than its competitors, indicating its superior capability to breach defenses. Moreover, K&L maintains a high BA, ensuring the attack's stealthiness. This dual achievement of high ASR and BA is not commonly observed in other methods. Particularly noteworthy is the performance of K&L on the Tiny ImageNet dataset under the RNP defense mechanism. It achieves an ASR of 99.92% while maintaining a BA of 56.53%. This contrasts starkly with other methods, which lag considerably behind K&L both in terms of ASR and the ability to maintain a reasonable BA, further illustrating the effectiveness of K&L in balancing attack aggressiveness with stealthiness.

Furthermore, we use attribution visualization to analyze the stealthiness of our method as compared to others. As illustrated in Figure 2, it is evident that while our method retains high defense breaching capabilities, it can conceal the trigger within the vicinity of the main features, as evidenced by advanced attribution methods. In contrast, features in other methods show a noticeable deviation from the original image. For instance, in BadNet, the features are entirely attributed to the trigger in the bottom-right corner. Blended and BppAttack, as well as WaNet, have features diffusing across the entire image. Input-Aware and SSBA exhibit feature shifts that result in attribution outcomes deviating from the target subject. We quantify the similarity between the attribution maps in Figure 2 using statistical metrics. We apply Softmax to the attribution results of both the attacked and original images and then calculate Cosine similarity. Table 2 shows that K&L has the highest similarity, indicating our method's samples have better attribution results to the original image.

Moreover, Table 3 shows the AAV corresponding to the AAC of Figure 3, including our K&L approach, across different AAC scenarios. K&L outperforms others with the highest AAC values (0.9415, 0.9111, and 0.8772 for AAC1, AAC3, and AAC5, respectively), indicating its superior

ability to penetrate defenses while maintaining accuracy. Unlikely, BadNet, Blended, and SSBA methods show lower performance, especially under stringent accuracy constraints (AAC1).

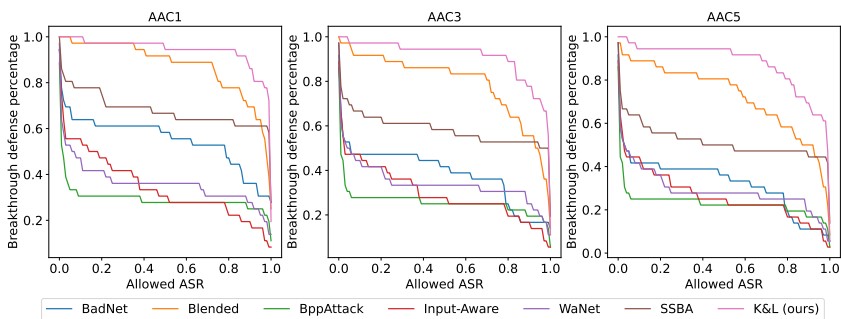

Figure 3: AAC of different attack methods on PreActResNet18

Table 3: AAV of different attack methods on PreActResNet18

|        | BadNet | Blended | BppAttack | Input-Aware | WaNet  | SSBA   | K&L (ours) |
|--------|--------|---------|-----------|-------------|--------|--------|------------|
| AAC1   | 0.5503 | 0.8749  | 0.2928    | 0.3443      | 0.3603 | 0.6800 | **0.9415** |
| AAC3   | 0.3847 | 0.7929  | 0.2576    | 0.2986      | 0.3413 | 0.5842 | **0.9111** |
| AAC5   | 0.3215 | 0.7244  | 0.2300    | 0.2657      | 0.2907 | 0.5153 | **0.8772** |

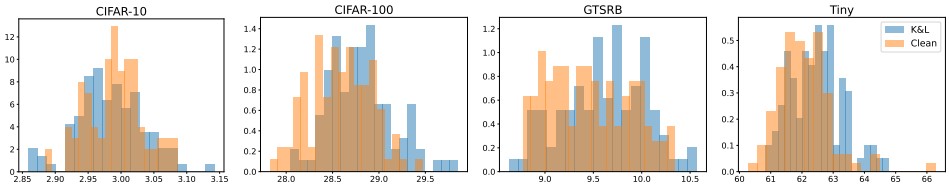

Figure 4: STRIP

In the Figure 4 across the four datasets the overlapping distributions of the K&L and Clean samples indicate a challenge for the STRIP method to discern between benign and backdoored data. The substantial overlap suggests that the K&L backdoor attack can effectively mimic the statistical profile of clean data, thus eluding STRIP's detection capabilities. This conveys that the K&L method possesses the potential to circumvent the defensive mechanism of STRIP, demonstrating a limitation in the STRIP's robustness against this particular type of trojan attack.

Overall, the results show that K&L not only excels in attacking without defense mechanisms but also have remarkable resilience and potency in evading various defense algorithms. It stands out as the most effective method in penetrating defenses while maintaining high attack stealthiness across all tested datasets.

### 4.3 ABLATION RESULTS

Our ablation study assesses the effect of four parameters (epochs, learning rate $\eta$, *level*, and attack step size $\alpha$) on the K&L backdoor attack's efficacy, using PreActResNet18 on CIFAR-10. Default settings for these parameters are epochs and *level* at 4, learning rate at 0.01, and attack step size at 1. During ablation, only the parameter under investigation is altered, with the others held at their defaults. Due to space constraints, full results of the tables are included in the **Appendix F- G**.

**Ablation study on Epochs**

As depicted in Table 8, increasing epochs from 2 to 12 enhances BA, suggesting improved model performance on non-adversarial inputs. Conversely, the ASR initially high, diminishes slightly with more epochs, pointing to a trade-off between extended training and attack effectiveness.

**Ablation study on Learning Rate** $\eta$

From Table 9, we can see that altering the learning rate from 0.001 to 0.1 impacts both BA and ASR. A learning rate of 0.01 is the optimal, upholding a high ASR with minimal compromise to BA, which is pivotal for calibrating the backdoor attack.

**Ablation Study on** *level*

Our findings, as summarized in Table 10, varying *level* from 2 to 12, an upsurge is noted in ASR, and the model's capability to evade defenses is heightened. The BA remains mostly unaffected, indicating that increasing generate steps predominantly bolsters the backdoor's potency.

**Ablation Study on Attack Step Size** $\alpha$

As illustrated in Table 11, Adjusting the attack step size between 0.25 and 1.5 suggests inadequacy of smaller sizes for effective backdoor activation, whereas larger sizes sustain a high ASR. However, gains in ASR plateau beyond a step size of 1, accentuating the necessity to pinpoint an optimal magnitude.

## 5 CONCLUSION

In this work, we present the key requirements for successful backdoor attacks and address the high binding nature prevalent in existing methods. Our novel Key-Locks Backdoor Attack algorithm effectively circumvents this challenge, proving resilience against nearly all current defense methods while maintaining minimal perturbation. Extensive experiments validate our approach, showcasing the K&L algorithm's effectiveness in penetrating backdoor defense methods. However, The process of the K&L attack requires keeping the key at hand. Although other backdoor attack algorithms may also store the trigger or the trained model, the K&L approach increases the demands of computing resources, which is directly related to the model size. Despite this, we believe the effectiveness of the attack justifies the additional cost.

## CODE OF ETHICS AND ETHICS STATEMENT

All authors of this paper have read and adhered to the ICLR Code of Ethics, as outlined in the conference guidelines (https://iclr.cc/public/CodeOfEthics). We affirm that no human subjects were involved in the experiments presented in this work. Additionally, no potentially harmful methodologies or applications are proposed in this paper, and all datasets used are publicly available with proper citations. The research complies with legal, privacy, and security standards, and we have disclosed no conflicts of interest. Any insights derived from this work are intended for the scientific advancement of backdoor defense mechanisms and are not designed to be misused. We welcome open discussions during the review process to ensure compliance with ethical standards.

## REPRODUCIBILITY STATEMENT

We have made every effort to ensure the reproducibility of our work. Detailed descriptions of the datasets, models, and evaluation metrics used in our experiments can be found in the main paper and appendix. The pseudocode for our proposed Key-Locks algorithm is provided in Appendix B, and the detailed experimental setup, including hyperparameters, can be found in Section 4.1. The source code for our experiments, including model training and evaluation scripts, will be made available anonymously upon acceptance. Additionally, the appendix includes all theoretical proofs, data preprocessing steps, and further clarification of any assumptions made in our methodology to facilitate reproducibility.

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

## A    FLOWCHART OF KEY-LOCKS BACKDOOR ATTACK

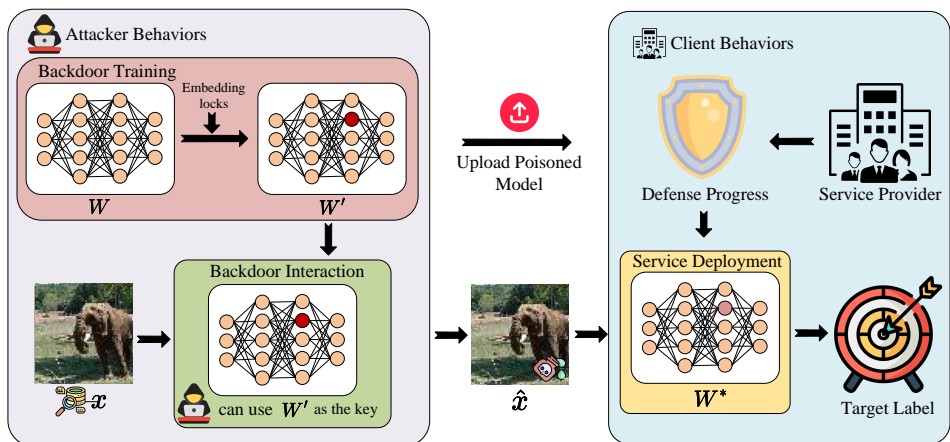

Figure 5: Flowchart of Key-Locks Backdoor Attack

## B    PSEUDOCODE

### B.1    EMBEDDING LOCKS

---

**Algorithm 1** Embedding locks

---

**Input:** training datasets $D$, training steps $epoches$, backdoor datasets $B$, loss function $L$, learning rate $\eta$,
     backdoor target label $c_t$
**Output:** Trained parameters $W$
1: **Initial:** $D = \{\{x_1, y_1\}, \{x_2, y_2\}, \cdots, \{x_n, y_n\}\}$
2: **Initial:** $B = \{\}$, initial parameters $W$ if the pretrained model is not provide.
3: **for** $e$ in range ($epoches$) **do**
4:     **if** $e == 0$ **then**
5:         **for** $(x, y)$ in sample_batch (part($D$)) **do**
6:             $x' = x - \eta \cdot sign \left( \frac{\partial L(x, c_t; W)}{\partial x} \right)$
7:             $B$.append($x', c_t$)
8:         **end for**
9:     **else**
10:         **for** $(x', y)$ in sample_batch ($B$) **do**
11:             $x' = x' - \eta \cdot sign \left( \frac{\partial L(x, c_t; W)}{\partial x} \right)$
12:         **end for**
13:     **end if**
14:     **for** $(x, y, x', c_t)$ in sample_batch ($D, B$) **do**
15:         $L_{total} = L(x, y; W) + L(x', c_t; W)$
16:         update $W$ by using the gradient descent based on $L_{total}$
17:     **end for**
18: **end for**
19: **return** $W$

---

## B.2 GENERATE BACKDOOR ATTACK EXAMPLES

---

**Algorithm 2** Generate backdoor attack examples

---

**Input:** $\epsilon^l$ is the allowed perturbation base on $level$, Learning rate $\alpha$, backdoor parameter $W'$, attack levels $level$

**Output:** the backdoor attack sample $x$

1: $x$ normalized to $(0, 1)$
2: $\min = \max(x - \epsilon^l, 0)$
3: $\max = \min(x + \epsilon^l, 1)$
4: **for** $l$ in range $(levels)$ **do**
5:     $grad = \frac{\partial L(x, t; W')}{\partial x}$
6:     $x = x - \alpha \cdot sign(grad)$
7:     $x = clip(x, \min, \max)$
8: **end for**
9: **return** $x$

---

## C THREAT MODEL

Suppose an attacker's goal is to implant a backdoor into an artificial intelligence model used in an automated image recognition system. This backdoor would trigger incorrect behavior or outputs when specific image features are detected. The attacker has the ability to access or influence the model's training data and possesses sufficient permissions and opportunities to participate in the training to implant a backdoor. By embedding this backdoor, AI model will perform predetermined erroneous actions, such as misclassification, when it detects images containing specific triggers. The attacker utilizes our Key and Locks (K&L) attack method to generate triggers that are difficult for the human eye to detect, achieving a more covert attack. A successful backdoor attack could lead to the system making incorrect decisions in practical applications, such as a security monitoring system failing to correctly identify threats, resulting in security vulnerabilities. The hypothesis is feasible, as many companies currently do not possess the resources to train models themselves, and therefore, they tend to outsource model training to third parties. This situation provides opportunities for attackers. Additionally, there are numerous platforms for open-source models, such as Hugging Face, where attackers could upload their backdoored models. These factors make the threat model feasible.

## D EVALUATION METRICS

### D.1 BENIGN ACCURACY (BA) AND ATTACK SUCCESS RATE (ASR)

BA, calculated as the ratio of correct predictions on clean test instances, reflects the model's normal operation, ensuring the backdoor does not impair primary task performance. ASR, measuring the rate at which backdoored samples are misclassified into a targeted class, assesses the backdoor's effectiveness. High ASR indicates effective trigger recognition, while BA assures the attack's stealth by demonstrating unaltered performance on regular inputs. Both metrics are crucial for a comprehensive assessment of the backdoor attack's impact and stealth.

### D.2 ACCURACY-ASR CURVE (AAC)

To facilitate the evaluation of backdoor attack algorithms, we introduce a new metric: Accuracy-ASR Curve (AAC). The y-axis represents the percentage of the attack method penetrating the defense algorithms, while the x-axis corresponds to the ASR threshold for considering a defense successful. A higher ASR on this curve signifies a more lenient criterion for successful defense. As shown in Figure 3, the AAC metric requires setting a parameter for the permissible loss in accuracy. We employ AAC3 to denote the defense scenario where an accuracy loss of up to 3% is allowed. We compute the area under the AAC and term it the Accuracy-ASR Value (AAV); a higher AAC value implies a more effective attack method under the defined accuracy loss constraint.

# E    COMPARISON EXPERIMENT

All experiments in this study are conducted on 2 NVIDIA RTX 6000 Ada graphics cards. Each attack method was employed with a poison rate of 10%, meaning that 10% of the training data was subtly altered to include the backdoor trigger. The target class for all attacks was set to Class 0. The L&K attack method specifically involves tuning four key parameters: two during training (the number of training epochs and the learning rate), and two during the attack phase (the number of steps referred to as generate steps and the attack step size). For the PreActResNet18 model, we set the epochs to 10, learning rate to 0.001 for CIFAR-10 and 0.01 for other three datasets, generate steps to 4, step size to $\frac{1}{255}$, and The disturbance rate $\epsilon$ is $\frac{4}{255}$.

## E.1    EXPERIMENT ON VGG19-BN

**Parameters**

In attacking the VGG19-BN model, the epochs, learning rate, generate steps, and step size were configured as 10, 0.001, 10, and 1/255, respectively.

**Result**

As shown in Table 4, it is evident that our K&L Backdoor Attack method can breach nearly all existing defense mechanisms on the VGG19-BN model across the CIFAR-10, CIFAR-100, GTSRB, and Tiny ImageNet datasets.

Table 4: Comparison results of attack methods against various defense algorithms using VGG19-BN. The model's original accuracy on CIFAR-10, CIFAR-100, GTSRB, and Tiny ImageNet datasets are 92.42%, 65.4%, 98.01%, and 54.01%, respectively. This table presents a detailed comparison of several attack methods, including K&L (ours), across different datasets. The performance is evaluated in terms of Benign Accuracy (BA) and Attack Success Rate (ASR) under different defense mechanisms, including ANP, BNP, FP, FT, I-BAU, NAD, CLP, and RNP.

| Datasets | Attack Methods | No Defense | ANP | BNP | FP | FT | I-BAU | NAD | CLP | RNP |
|---|---|---|---|---|---|---|---|---|---|---|
| | | BA/ASR | BA/ASR | BA/ASR | BA/ASR | BA/ASR | BA/ASR | BA/ASR | BA/ASR | BA/ASR |
| CIFAR-10 | BadNet | 90.7/95.17 | 84.14/0.29 | 90.48/95.46 | 89.37/24.91 | 88.94/37.33 | 86.42/6.51 | 87.07/7.99 | 88.8/12.53 | 90.58/27.29 |
| | Blended | 91.48/96.16 | 88.63/1.68 | 90.94/95.63 | 89.55/67.97 | 89.83/63.01 | 89.4/32.77 | 88.65/61.64 | 89.01/53.76 | 91.48/96.16 |
| | BppAttack | 89.49/99.67 | 83.46/0.11 | 87.95/2.41 | 91.42/9.67 | 91.3/2.07 | 90.38/6.83 | 90.98/2.04 | 86.54/2.52 | 89.02/1.72 |
| | Input-Aware | 89.04/94.51 | 86.93/0.08 | 87.32/0.17 | 90.97/36.43 | 91.01/18.4 | 90.0/3.72 | 91.02/6.41 | 85.25/15.43 | 88.02/33.81 |
| | SSBA | 88.28/67.9 | 84.54/0.24 | 72.29/11.84 | 91.49/2.04 | 91.42/1.47 | 87.31/2.11 | 91.41/1.33 | 87.94/3.08 | 89.11/2.8 |
| | WaNet | 91.89/100.0 | 84.86/0.02 | 91.9/99.99 | 90.34/23.14 | 90.07/3.5 | 88.07/67.12 | 88.58/3.04 | 83.92/14.11 | 90.83/1.29 |
| | K&L | 91.88/99.46 | 86.05/44.38 | 91.88/99.26 | 89.79/46.57 | 89.16/47.78 | 86.67/51.72 | 83.87/44.87 | 88.78/89.94 | 81.79/47.07 |
| CIFAR-100 | BadNet | 60.89/88.51 | 60.42/0.0 | 58.79/87.91 | 60.95/27.47 | 58.99/0.13 | 54.95/0.64 | 57.8/0.06 | 56.74/64.98 | 60.08/2.77 |
| | Blended | 64.12/92.37 | 58.99/24.64 | 60.29/81.22 | 62.67/60.92 | 61.24/53.38 | 58.02/55.69 | 59.18/43.35 | 59.89/48.44 | 64.09/79.83 |
| | BppAttack | 60.77/96.46 | 55.63/0.04 | 56.16/0.49 | 64.9/0.1 | 64.65/0.03 | 62.55/4.89 | 64.25/0.04 | 56.17/0.13 | 42.17/95.76 |
| | Input-Aware | 60.48/89.56 | 57.52/0.17 | 58.63/0.11 | 64.34/43.71 | 64.25/98.58 | 59.19/93.3 | 63.81/98.71 | 57.02/8.08 | 59.32/89.93 |
| | SSBA | 55.49/98.3 | 51.99/0.45 | 16.62/99.62 | 65.05/0.11 | 64.65/0.34 | 61.67/0.57 | 64.07/0.11 | 52.57/37.88 | 59.86/54.37 |
| | WaNet | 64.06/99.89 | 59.68/0.07 | 63.33/89.55 | 62.51/87.98 | 60.9/1.93 | 57.41/11.59 | 59.12/36.77 | 57.24/99.86 | 61.45/0.22 |
| | K&L | 64.55/99.84 | 60.78/45.89 | 64.5/99.76 | 64.7/77.52 | 64.21/88.02 | 59.24/41.88 | 64.75/98.44 | 57.81/92.35 | 24.25/13.03 |
| GTSRB | BadNet | 97.74/94.83 | 88.63/0.11 | 97.47/94.83 | 98.09/2.04 | 97.88/5.47 | 94.98/0.06 | 97.62/86.4 | 97.44/0.85 | 97.93/92.63 |
| | Blended | 96.86/99.24 | 94.17/0.0 | 96.99/98.57 | 97.46/96.99 | 97.13/97.33 | 96.86/69.6 | 96.9/95.49 | 96.29/98.71 | 96.86/99.24 |
| | BppAttack | 97.82/97.24 | 97.65/0.0 | 97.55/96.41 | 98.57/51.54 | 98.63/0.06 | 97.49/0.01 | 98.38/0.03 | 97.64/0.35 | 97.9/0.02 |
| | Input-Aware | 96.66/74.11 | 96.36/0.0 | 96.46/0.0 | 98.08/21.65 | 97.78/5.65 | 96.61/30.26 | 97.58/0.43 | 96.38/60.0 | 96.31/0.0 |
| | SSBA | 94.89/95.56 | 96.73/0.0 | 0.55/100.0 | 98.53/15.67 | 98.57/0.72 | 97.6/0.26 | 97.99/1.0 | 7.75/99.98 | 96.19/0.02 |
| | WaNet | 97.75/99.69 | 92.61/23.09 | 97.85/99.62 | 98.38/83.37 | 98.19/98.81 | 95.06/20.2 | 98.05/99.43 | 97.69/99.58 | 95.3/97.81 |
| | K&L | 97.28/100.0 | 95.19/0.0 | 97.08/98.23 | 97.36/100.0 | 97.3/98.75 | 94.78/45.16 | 97.41/100.0 | 96.69/90.45 | 97.28/100.0 |
| Tiny | BadNet | 51.72/99.99 | 47.8/0.21 | 50.77/100.0 | 50.36/2.68 | 50.92/98.77 | 43.75/97.5 | 41.27/0.27 | 51.95/98.58 | 51.57/99.99 |
| | Blended | 40.41/95.13 | 36.43/74.19 | 39.74/93.13 | 32.11/5.62 | 30.07/10.0 | 10.32/0.01 | 18.15/0.06 | 40.41/95.13 | 40.41/95.13 |
| | BppAttack | 54.91/99.97 | 55.0/0.0 | 54.93/99.96 | 54.1/0.16 | 54.91/0.09 | 46.37/89.31 | 33.14/0.2 | 55.05/0.2 | 54.98/0.0 |
| | Input-Aware | 53.58/99.88 | 53.45/0.0 | 53.46/0.03 | 52.65/0.29 | 53.13/0.05 | 46.75/0.14 | 41.13/0.25 | 53.6/0.16 | 53.27/0.0 |
| | SSBA | 55.09/99.95 | 53.39/0.11 | 54.84/99.96 | 54.32/66.09 | 54.57/0.11 | 50.76/94.1 | 37.63/0.24 | 55.37/92.52 | 54.96/0.04 |
| | WaNet | 52.61/99.92 | 48.1/0.03 | 52.39/99.84 | 51.61/0.49 | 51.39/0.44 | 45.56/1.79 | 37.7/0.48 | 52.4/13.84 | 51.78/0.08 |
| | K&L | 53.06/100.0 | 52.9/85.53 | 53.07/100.0 | 53.93/98.77 | 54.16/99.94 | 47.75/36.69 | 31.56/4.0 | 53.09/100.0 | 42.11/54.95 |

## E.2    EXPERIMENT ON MOBILENET-V3-LARGE

**Parameters**

For the MobileNet-v3-large model, on the simplest dataset, GTSRB, the parameters, epochs, learning rate, generate steps, and step size, are set to 4, 0.01, 4, and 1/255, respectively. On the other three relatively complex datasets, these parameters are each set to 10, 0.001, 10, and 1/255.

**Result**

Table 5: Comparison results of attack methods against various defense algorithms using MobileNet-v3-large. The model's original accuracy on CIFAR-10, CIFAR-100, GTSRB, and Tiny ImageNet datasets are 84.25%, 53.59%, 94.71%, and 39.96%, respectively. This table presents a detailed comparison of several attack methods, including K&L (ours), across different datasets. The performance is evaluated in terms of Benign Accuracy (BA) and Attack Success Rate (ASR) under different defense mechanisms, including ANP, BNP, FP, FT, I-BAU, NAD, CLP, and RNP.

| Datasets | Attack Methods | No Defense | ANP | BNP | FP | FT | I-BAU | NAD | CLP | RNP |
|---|---|---|---|---|---|---|---|---|---|---|
|  |  | BA/ASR | BA/ASR | BA/ASR | BA/ASR | BA/ASR | BA/ASR | BA/ASR | BA/ASR | BA/ASR |
| CIFAR-10 | BadNet | 82.52/93.61 | 82.51/93.46 | 54.9/10.18 | 78.02/18.92 | 78.36/13.09 | -/- | 77.94/10.53 | 81.97/1.83 | 82.47/2.37 |
|  | Blended | 82.27/90.0 | 80.13/73.24 | 65.29/91.32 | 77.44/4.44 | 77.86/5.73 | -/- | 73.54/5.31 | 62.84/93.97 | 78.31/8.56 |
|  | BppAttack | 73.94/99.13 | 70.35/1.97 | 73.03/4.92 | 80.63/13.57 | 79.99/4.14 | -/- | 81.69/4.43 | 58.39/1.67 | 76.09/3.06 |
|  | Input-Aware | 78.37/85.46 | 71.95/2.94 | 78.35/6.82 | 79.4/11.44 | 79.37/25.02 | -/- | 81.44/10.01 | 78.08/31.22 | 78.85/5.6 |
|  | SSBA | 69.5/93.59 | 70.74/5.87 | 61.63/93.81 | 80.23/1.64 | 80.88/3.8 | -/- | 82.02/4.56 | 35.98/98.43 | 79.22/8.73 |
|  | WaNet | 83.34/99.93 | 77.57/4.42 | 78.95/99.97 | 78.76/13.03 | 78.46/3.94 | -/- | 77.73/2.54 | 74.0/99.97 | 81.96/4.64 |
|  | K&L | 83.92/100.0 | 83.92/100.0 | 80.3/99.51 | 79.11/93.93 | 78.5/95.81 | -/- | 77.28/98.03 | 82.46/99.92 | 55.0/64.61 |
| CIFAR-100 | BadNet | 50.16/91.55 | 49.49/0.38 | 44.17/92.02 | 41.97/1.32 | 44.11/2.57 | -/- | 43.4/7.26 | 45.64/5.14 | 50.25/14.77 |
|  | Blended | 49.16/89.08 | 45.83/76.68 | 42.36/48.84 | 42.7/2.46 | 43.97/15.51 | -/- | 42.43/16.92 | 36.56/2.45 | 47.93/5.55 |
|  | BppAttack | 47.19/60.06 | 43.1/5.82 | 37.3/3.55 | 47.09/80.4 | 47.35/49.82 | -/- | 51.59/3.33 | 44.9/53.35 | 47.08/15.13 |
|  | Input-Aware | 47.61/87.7 | 42.73/0.05 | 35.75/54.09 | 46.35/15.66 | 47.09/9.3 | -/- | 50.74/0.31 | 44.87/53.87 | 43.09/3.39 |
|  | SSBA | 30.57/98.38 | 29.95/9.83 | 23.9/96.21 | 47.64/0.48 | 48.46/0.65 | -/- | 51.26/4.55 | 32.38/95.21 | 44.31/8.15 |
|  | WaNet | 49.53/99.99 | 48.31/99.96 | 44.11/99.99 | 42.76/2.75 | 44.46/1.45 | -/- | 42.63/2.41 | 39.05/0.14 | 49.11/0.69 |
|  | K&L | 53.87/99.97 | 50.92/99.95 | 41.78/97.93 | 45.0/91.63 | 46.27/93.17 | -/- | 45.59/93.38 | 45.01/99.26 | 30.48/74.79 |
| GTSRB | BadNet | 93.63/91.54 | 93.94/0.0 | 91.54/92.06 | 95.71/4.49 | 95.68/33.04 | -/- | 95.36/25.45 | 82.93/0.05 | 93.95/0.0 |
|  | Blended | 91.86/93.84 | 89.48/60.52 | 90.82/92.41 | 95.0/15.77 | 94.41/72.2 | -/- | 93.86/69.58 | 86.32/93.29 | 88.38/44.8 |
|  | BppAttack | 93.37/78.64 | 93.63/0.0 | 79.4/1.04 | 95.88/0.03 | 95.76/0.02 | -/- | 95.0/0.0 | 92.43/19.26 | 93.33/0.78 |
|  | Input-Aware | 90.48/65.85 | 85.25/0.0 | 89.19/48.55 | 94.1/0.06 | 93.81/0.45 | -/- | 92.97/1.9 | 85.79/26.95 | 90.8/5.7 |
|  | SSBA | 56.33/80.27 | 57.02/79.47 | 86.54/0.83 | 96.08/0.0 | 95.92/0.0 | -/- | 95.47/0.02 | 46.47/85.1 | 91.59/2.1 |
|  | WaNet | 94.43/99.98 | 89.53/12.35 | 92.49/99.74 | 95.72/34.78 | 95.55/56.66 | -/- | 94.73/46.01 | 85.86/0.02 | 90.22/0.76 |
|  | K&L | 91.24/99.74 | 88.14/97.1 | 88.3/99.63 | 95.65/29.44 | 95.55/61.22 | -/- | 93.67/72.35 | 86.31/98.33 | 83.95/78.12 |
| Tiny | BadNet | 40.44/99.86 | 38.96/0.89 | 39.38/99.36 | 39.68/0.43 | 40.59/0.6 | -/- | 34.9/0.73 | 40.45/0.44 | 40.42/0.83 |
|  | Blended | 23.07/91.27 | 21.8/67.66 | 19.84/89.58 | 24.8/22.53 | 22.21/21.47 | -/- | 13.41/0.22 | 23.14/91.28 | 12.6/39.23 |
|  | BppAttack | 45.1/96.59 | 41.29/0.05 | 42.25/97.39 | 38.62/0.33 | 40.08/0.22 | -/- | 23.35/0.23 | 45.38/0.27 | 43.34/0.04 |
|  | Input-Aware | 42.89/99.57 | 41.34/0.04 | 34.89/99.34 | 37.0/0.26 | 38.58/0.18 | -/- | 25.54/0.36 | 43.07/0.53 | 43.11/0.06 |
|  | SSBA | 45.33/97.48 | 43.46/65.12 | 35.04/93.28 | 37.19/0.74 | 39.82/0.61 | -/- | 25.08/0.24 | 42.59/97.93 | 45.6/0.09 |
|  | WaNet | 38.09/99.97 | 38.19/2.2 | 37.38/99.94 | 39.54/7.06 | 40.34/3.24 | -/- | 34.91/0.72 | 38.08/50.69 | 38.18/0.55 |
|  | K&L | 47.52/99.99 | 47.45/99.97 | 42.33/99.93 | 37.77/96.78 | 39.46/98.08 | -/- | 27.92/29.3 | 47.48/99.99 | 34.48/66.16 |

Table 6: Performance of backdoor attacks with and without embedding locks. BN represents benign accuracy, and ASR represents attack success rate.

|  | BN | ASR |
|---|---|---|
| Without Embedding Locks | 94.08% | 90.71% |
| With Embedding Locks | 93.71% | 99.88% |

As shown in Table 5, it is evident that our K&L Backdoor Attack method can breach nearly all existing defense mechanisms on the MobileNet-v3-large model across the CIFAR-10, CIFAR-100, GTSRB, and Tiny ImageNet datasets.

**The contribution between adversarial and backdoored perturbation** Firstly, our adversarial noise varies with each step, making it non-unique to each sample. Furthermore, we have supplemented our findings in the PreActResNet-18 model by comparing the performance changes before and after embedding locks to demonstrate the effectiveness of our backdoor mechanism. As shown in Table 6, the performance of the backdoor attacks has significantly improved through the embedding locks mechanism, showing a clear advantage over other algorithms.

### E.3    EXPERIMENT ON EFFICIENTNET-B3

**Parameters**

For the EfficientNet_B3 model, parameter settings were differentiated based on dataset complexity. On simpler datasets like CIFAR-10 and GTSRB, epochs, learning rate, generate steps, and step size were set to 4, 0.01, 4, and 1/255, respectively. Conversely, for more complex datasets such as CIFAR-100 and Tiny ImageNet, these parameters were adjusted to 10, 0.001, 10, and 1/255.

**Result**

Table 7: Comparison results of attack methods against various defense algorithms using EfficientNet-B3. The model's original accuracy on CIFAR-10, CIFAR-100, GTSRB, and Tiny ImageNet datasets are 69.21%, 50.45%, 84.69%, and 46.66%, respectively. This table presents a detailed comparison of several attack methods, including K&L (ours), across different datasets. The performance is evaluated in terms of Benign Accuracy (BA) and Attack Success Rate (ASR) under different defense mechanisms, including ANP, BNP, FP, FT, I-BAU, NAD, CLP, and RNP.

| Datasets | Attack Methods | No Defense | ANP | BNP | FP | FT | I-BAU | NAD | CLP | RNP |
|---|---|---|---|---|---|---|---|---|---|---|
| | | BA/ASR | BA/ASR | BA/ASR | BA/ASR | BA/ASR | BA/ASR | BA/ASR | BA/ASR | BA/ASR |
| CIFAR-10 | BadNet | 55.75/10.39 | 55.79/10.39 | 54.36/11.89 | 53.0/4.94 | 53.07/5.58 | 54.14/5.36 | 53.29/5.38 | 25.14/24.59 | 55.75/10.42 |
| | Blended | 55.5/25.46 | 55.71/23.72 | 54.43/28.71 | 53.03/4.1 | 52.04/6.31 | 55.0/5.76 | 53.01/4.0 | 17.68/70.39 | 55.5/25.46 |
| | BppAttack | 70.03/78.73 | 69.33/57.74 | 53.4/7.07 | 70.12/8.6 | 70.68/5.13 | 72.33/21.1 | 72.07/3.52 | 10.65/83.38 | 64.15/4.71 |
| | Input-Aware | 61.84/78.72 | 54.89/68.94 | 52.68/56.21 | 61.55/22.04 | 61.4/20.92 | 61.93/53.37 | 62.56/28.63 | 24.04/5.8 | 60.37/30.44 |
| | SSBA | 66.49/71.82 | 65.73/14.16 | 65.99/73.01 | 69.97/3.09 | 69.65/5.39 | 71.89/4.06 | 71.7/4.27 | 38.31/3.97 | 68.22/3.21 |
| | WaNet | 54.77/9.59 | 54.77/9.61 | 53.39/10.13 | 53.44/3.74 | 53.07/5.08 | 52.96/5.87 | 52.45/5.33 | 22.39/43.94 | 54.78/9.63 |
| | K&L | 65.38/98.94 | 65.37/98.94 | 40.18/82.3 | 66.57/45.78 | 65.7/45.67 | 67.16/90.64 | 67.45/72.11 | 17.4/5.93 | 21.67/3.19 |
| CIFAR-100 | BadNet | 45.84/88.98 | 45.82/88.52 | 44.18/88.38 | 34.84/4.2 | 36.83/7.72 | 38.99/68.14 | 37.4/9.14 | 2.69/0.53 | 45.01/2.82 |
| | Blended | 48.02/81.03 | 47.89/80.01 | 39.05/37.19 | 38.28/0.88 | 38.71/3.06 | 41.89/34.05 | 39.69/5.69 | 10.48/19.86 | 38.36/5.41 |
| | BppAttack | 48.77/78.86 | 48.21/0.78 | 45.45/76.43 | 43.58/0.15 | 44.65/0.08 | 49.13/52.75 | 46.45/1.91 | 10.62/0.25 | 41.03/0.83 |
| | Input-Aware | 44.68/91.53 | 42.12/1.64 | 24.78/6.3 | 40.79/0.37 | 40.81/0.34 | 46.51/58.88 | 46.45/1.91 | 13.53/0.12 | 46.26/0.55 |
| | SSBA | 45.45/91.92 | 49.78/36.08 | 38.11/0.01 | 44.86/0.31 | 44.69/1.51 | 49.42/14.15 | 49.45/5.36 | 10.1/38.91 | 50.34/2.12 |
| | WaNet | 48.05/99.79 | 46.82/53.43 | 39.09/0.12 | 37.85/26.28 | 39.58/63.6 | 41.46/94.3 | 40.27/63.67 | 16.99/4.51 | 48.13/8.28 |
| | K&L | 49.7/99.33 | 48.47/99.29 | 41.6/96.85 | 39.37/54.48 | 40.29/71.02 | 43.73/95.12 | 41.67/84.31 | 20.2/24.89 | 20.41/77.39 |
| GTSRB | BadNet | 80.82/82.31 | 77.55/71.82 | 80.4/80.55 | 83.36/9.29 | 82.11/26.01 | 81.65/58.97 | 81.92/42.12 | 14.18/40.39 | 78.73/23.21 |
| | Blended | 78.0/79.36 | 70.85/67.52 | 76.05/73.05 | 81.05/3.41 | 79.9/30.48 | 76.74/43.68 | 78.04/49.32 | 37.71/76.44 | 39.11/10.92 |
| | BppAttack | 82.83/23.64 | 76.44/9.81 | 82.13/23.85 | 88.54/5.58 | 87.61/4.54 | 85.83/8.81 | 87.32/8.07 | 33.8/23.44 | 81.77/14.63 |
| | Input-Aware | 63.63/5.45 | 63.25/3.44 | 64.51/4.33 | 68.93/0.68 | 68.27/0.08 | 66.56/0.48 | 68.57/0.04 | 38.54/18.3 | 65.34/3.38 |
| | SSBA | 77.13/4.02 | 77.13/4.02 | 77.43/4.21 | 82.14/0.1 | 81.62/0.27 | 79.06/0.76 | 81.09/0.18 | 26.71/37.46 | 77.12/4.03 |
| | WaNet | 82.95/96.94 | 82.08/81.11 | 80.25/94.38 | 88.22/1.29 | 85.95/63.99 | 84.59/88.77 | 85.46/85.01 | 22.72/72.52 | 83.1/97.3 |
| | K&L | 81.62/97.58 | 76.05/91.5 | 72.64/97.88 | 86.84/49.82 | 85.51/69.67 | 84.54/72.9 | 84.69/81.54 | 37.14/13.64 | 69.54/37.54 |
| Tiny | BadNet | 38.42/100.0 | 36.48/3.62 | 30.07/100.0 | 42.45/0.0 | 43.23/0.15 | 40.46/99.08 | 35.57/0.01 | 38.42/100.0 | 38.2/1.83 |
| | Blended | 38.47/94.61 | 36.33/64.61 | 29.48/88.6 | 32.51/1.84 | 28.68/9.96 | 35.44/50.19 | 17.47/0.48 | 38.48/92.59 | 31.90/48.99 |
| | BppAttack | 48.18/99.93 | 48.21/0.1 | 48.09/5.91 | 43.79/0.17 | 44.05/0.5 | 44.45/94.55 | 27.34/0.3 | 45.59/68.6 | 48.0/0.11 |
| | Input-Aware | 46.22/99.73 | 46.37/0.32 | 38.39/0.68 | 40.6/0.48 | 44.38/0.57 | 40.34/86.07 | 30.25/0.54 | 45.05/93.37 | 46.44/0.14 |
| | SSBA | 46.72/99.07 | 46.05/0.03 | 46.33/0.0 | 43.01/0.41 | 43.19/0.24 | 43.72/82.73 | 28.67/0.64 | 45.38/97.28 | 46.69/0.07 |
| | WaNet | 44.29/99.99 | 44.34/1.74 | 39.09/95.05 | 41.35/1.78 | 43.85/0.85 | 42.91/99.43 | 34.6/0.53 | 44.17/99.99 | 44.15/2.16 |
| | K&L | 46.23/99.99 | 46.25/99.99 | 41.22/99.99 | 40.4/60.4 | 42.63/98.84 | 40.49/99.85 | 30.97/38.85 | 44.56/99.99 | 38.32/98.08 |

As shown in Table 7, it is evident that our K&L Backdoor Attack method can breach nearly all existing defense mechanisms on the EfficientNet-B3 model across the CIFAR-10, CIFAR-100, GTSRB, and Tiny ImageNet datasets.

# F ABLATION RESULTS IN THE MAIN TEXT

## F.1 ABLATION STUDY ON EPOCHS

Table 8: Ablation study on the impact of epochs on the performance of K&L method. The table shows the variation in BA and ASR as the epoch changes.

| epochs | No Defense | ANP | BNP | FP | FT | I-BAU | NAD | CLP | RNP |
|---|---|---|---|---|---|---|---|---|---|
| | BA/ASR | BA/ASR | BA/ASR | BA/ASR | BA/ASR | BA/ASR | BA/ASR | BA/ASR | BA/ASR |
| 2 | 87.47/100.0 | 80.94/5.46 | 87.17/100.0 | 92.43/97.17 | 92.46/99.3 | 88.65/85.66 | 92.22/87.21 | 85.73/47.92 | 85.89/33.17 |
| 4 | 86.76/99.99 | 85.39/29.07 | 86.15/99.8 | 92.44/98.38 | 92.44/99.27 | 90.16/83.4 | 92.26/97.93 | 86.55/75.52 | 86.27/73.04 |
| 6 | 89.12/99.94 | 81.64/61.43 | 86.48/85.51 | 92.72/97.81 | 92.76/99.69 | 89.32/86.33 | 92.48/99.17 | 85.26/95.5 | 89.12/99.94 |
| 8 | 90.02/98.94 | 83.23/48.17 | 89.87/86.41 | 92.65/94.43 | 92.93/98.59 | 88.38/94.58 | 92.62/97.32 | 89.71/91.34 | 86.27/83.70 |
| 10 | 90.16/98.89 | 85.32/66.08 | 87.03/84.13 | 92.55/92.63 | 93.06/97.97 | 90.28/72.81 | 92.93/96.42 | 86.9/92.16 | 86.27/86.32 |
| 12 | 90.24/97.37 | 83.5/55.37 | 89.34/79.57 | 92.86/87.79 | 92.87/95.77 | 90.82/76.67 | 92.71/95.01 | 89.21/83.6 | 86.27/35.96 |

## F.2 ABLATION STUDY ON LEARNING RATE $\eta$

Table 9: Ablation study on the impact of learning rate on the performance of the K&L method. This table illustrates the variation in BA and ASR as the learning rate changes.

| LR | No Defense | ANP | BNP | FP | FT | I-BAU | NAD | CLP | RNP |
|---|---|---|---|---|---|---|---|---|---|
| | BA/ASR | BA/ASR | BA/ASR | BA/ASR | BA/ASR | BA/ASR | BA/ASR | BA/ASR | BA/ASR |
| 0.001 | 86.76/91.19 | 85.15/9.12 | 86.15/70.29 | 92.44/64.34 | 92.44/72.99 | 90.16/34.47 | 92.26/66.17 | 86.55/38.73 | 86.27/35.96 |
| 0.005 | 86.76/99.99 | 85.39/29.07 | 86.15/99.8 | 92.44/98.38 | 92.44/99.27 | 90.16/83.4 | 92.26/97.93 | 86.55/75.52 | 86.27/73.04 |
| 0.01 | 86.76/100.0 | 85.39/37.37 | 86.15/100.0 | 92.44/99.86 | 92.44/99.97 | 90.16/94.09 | 92.26/99.81 | 86.55/82.58 | 86.27/80.87 |
| 0.05 | 86.76/100.0 | 85.39/42.31 | 86.15/100.0 | 92.44/99.92 | 92.44/99.98 | 90.16/97.32 | 92.26/99.97 | 86.55/85.22 | 86.27/83.70 |
| 0.1 | 86.76/100.0 | 85.39/46.7 | 86.15/100.0 | 92.44/99.99 | 92.44/100.0 | 90.16/98.66 | 92.26/99.99 | 86.55/87.62 | 86.27/86.32 |

## F.3 ABLATION STUDY ON *level*

Table 10: Ablation study on the impact of *level* on the performance of the K&L method. This table illustrates the changes in BA and ASR as the generate steps parameter is varied.

| Generate Steps | No Defense | ANP | BNP | FP | FT | I-BAU | NAD | CLP | RNP |
|---|---|---|---|---|---|---|---|---|---|
| | BA/ASR | BA/ASR | BA/ASR | BA/ASR | BA/ASR | BA/ASR | BA/ASR | BA/ASR | BA/ASR |
| 2 | 86.76/91.19 | 85.15/9.12 | 86.15/70.29 | 92.44/64.34 | 92.44/72.99 | 90.16/34.47 | 92.26/66.17 | 86.55/38.73 | 86.27/35.96 |
| 4 | 86.76/99.99 | 85.39/29.07 | 86.15/99.8 | 92.44/98.38 | 92.44/99.27 | 90.16/83.4 | 92.26/97.93 | 86.55/75.52 | 86.27/73.04 |
| 6 | 86.76/100.0 | 85.39/37.37 | 86.15/100.0 | 92.44/99.86 | 92.44/99.97 | 90.16/94.09 | 92.26/99.81 | 86.55/82.58 | 86.27/80.87 |
| 8 | 86.76/100.0 | 85.39/42.31 | 86.15/100.0 | 92.44/99.92 | 92.44/99.98 | 90.16/97.32 | 92.26/99.97 | 86.55/85.22 | 86.27/83.70 |
| 10 | 86.76/100.0 | 85.39/46.7 | 86.15/100.0 | 92.44/99.99 | 92.44/100.0 | 90.16/98.66 | 92.26/99.99 | 86.55/87.62 | 86.27/86.32 |
| 12 | 86.76/100.0 | 85.39/50.5 | 86.15/100.0 | 92.44/100.0 | 92.44/100.0 | 90.16/99.32 | 92.26/100.0 | 86.55/89.41 | 86.27/88.36 |

## F.4 ABLATION STUDY ON ATTACK STEP SIZE $\alpha$

Table 11: Ablation study on the impact of attack step size on the performance of the K&L method. The table shows how BA and ASR vary with changes in step size.

| Step Size | No Defense | ANP | BNP | FP | FT | I-BAU | NAD | CLP | RNP |
|---|---|---|---|---|---|---|---|---|---|
| | BA/ASR | BA/ASR | BA/ASR | BA/ASR | BA/ASR | BA/ASR | BA/ASR | BA/ASR | BA/ASR |
| 0.25 | 85.65/84.77 | 77.7/2.84 | 82.96/80.97 | 92.17/15.16 | 92.05/29.13 | 88.61/25.87 | 92.12/28.27 | 76.63/5.84 | 84.43/30.76 |
| 0.5 | 88.69/99.8 | 81.26/1.6 | 87.69/90.77 | 92.71/78.8 | 92.58/95.2 | 89.23/77.22 | 92.53/94.37 | 86.61/21.11 | 87.01/18.86 |
| 0.75 | 88.51/99.87 | 84.41/7.22 | 87.11/40.62 | 92.54/96.08 | 92.61/99.3 | 90.28/82.93 | 92.27/98.79 | 87.08/49.69 | 87.82/55.58 |
| 1 | 86.76/99.99 | 85.39/29.07 | 86.15/99.8 | 92.44/98.38 | 92.44/99.27 | 90.16/83.4 | 92.26/97.93 | 86.55/75.52 | 86.27/73.04 |
| 1.25 | 88.37/99.99 | 82.67/48.77 | 87.52/98.87 | 92.63/98.88 | 92.87/99.86 | 88.71/97.3 | 92.48/99.4 | 87.64/85.76 | 87.97/81.11 |
| 1.5 | 89.15/99.93 | 81.35/14.83 | 87.61/46.87 | 92.65/99.8 | 92.44/99.96 | 89.6/95.38 | 92.38/99.92 | 86.92/65.71 | 88.20/71.72 |

# G ADDITIONAL ABLATION RESULTS

## G.1 ABLATION PARAMETERS SETTING

In the ablation experiments conducted, four models were evaluated across different datasets with specific parameter settings. As shown in Table 12, for the PreActResNet18 model, the parameters set for CIFAR-10, CIFAR-100, and Tiny ImageNet datasets were: epochs at 4, learning rate (LR) at 0.01, *level* at 4, and attack step size ($\alpha$) at 1. Notably, for the GTSRB dataset under the PreActResNet18 model, the learning rate was set to 0.001, diverging from the default setting. Similarly, for the VGG19-BN, MobileNet-v3-large, and EfficientNet-B3 models, the default parameter values were maintained across all datasets: CIFAR-10, CIFAR-100, GTSRB, and Tiny ImageNet, with epochs at 4, LR at 0.01, *level* at 4, and $\alpha$ at 1. In each section of the ablation study, all parameters except the one under investigation were kept at these default settings.

Table 12: Default Parameters Table in Ablation Study

| Models | Datasets | epochs | LR | *level* | $\alpha$ |
|---|---|---|---|---|---|
| PreActResNet18 | CIFAR-10 | 4 | 0.01 | 4 | 1 |
| | CIFAR-100 | 4 | 0.01 | 4 | 1 |
| | GTSRB | 4 | 0.001 | 4 | 1 |
| | Tiny | 4 | 0.01 | 4 | 1 |
| VGG19-BN | CIFAR-10 | 4 | 0.01 | 4 | 1 |
| | CIFAR-100 | 4 | 0.01 | 4 | 1 |
| | GTSRB | 4 | 0.01 | 4 | 1 |
| | Tiny | 4 | 0.01 | 4 | 1 |
| MobileNet-v3-large | CIFAR-10 | 4 | 0.01 | 4 | 1 |
| | CIFAR-100 | 4 | 0.01 | 4 | 1 |
| | GTSRB | 4 | 0.01 | 4 | 1 |
| | Tiny | 4 | 0.01 | 4 | 1 |
| EfficientNet-B3 | CIFAR-10 | 4 | 0.01 | 4 | 1 |
| | CIFAR-100 | 4 | 0.01 | 4 | 1 |
| | GTSRB | 4 | 0.01 | 4 | 1 |
| | Tiny | 4 | 0.01 | 4 | 1 |

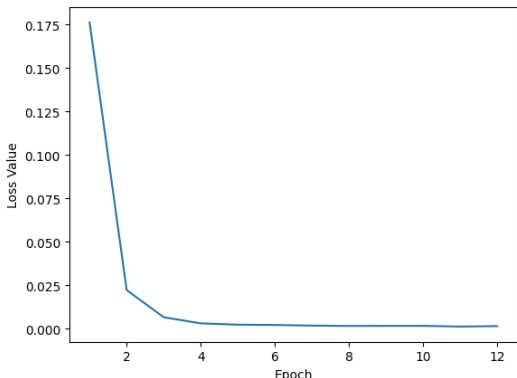

Figure 6: Training loss over 12 epochs using PreactResNet-18 on the CIFAR-10 dataset. The model converges by the fourth epoch, as indicated by the stabilization of the loss value.

## G.2 ABLATION STUDY ON EPOCHS

**PreActResNet18**: Given that an ablation study on the PreActResNet18 model using the CIFAR-10 dataset has already been conducted in the main text, this section will evaluate the impact of four parameters (epochs, learning rate $\eta$, *level*, and attack step size $\alpha$) on the efficacy of the K&L backdoor attack on the remaining three datasets: CIFAR-100, GTSRB, and Tiny ImageNet. The default settings for these parameters are epochs and *level* at 4, learning rate set to 0.01 for the GTSRB dataset and 0.01 for the others, and an attack step size of 1. During the ablation process, only the parameter under study is altered, while others are kept at their default values.

Analyzing Table 13 reveals that with increasing epochs, there is a consistent improvement in Benign Accuracy (BA) on the PreActResNet18 model, indicating enhanced model performance. Concurrently, Attack Success Rate (ASR) generally maintains a high level, signifying the robustness of the K&L Backdoor Attack method against various defense mechanisms over different epoch settings.

Table 13: Ablation Study Assessing the Effect of Epoch Variability on the Performance of the K&L Method with PreActResNet18 on CIFAR-100, GTSRB, and Tiny ImageNet - Comparison of Benign Accuracy (BA) and Attack Success Rate (ASR)

| Datasets | epochs | No Defense | ANP | BNP | FP | FT | I-BAU | NAD | CLP | RNP |
|---|---|---|---|---|---|---|---|---|---|---|
| | | BA/ASR | BA/ASR | BA/ASR | BA/ASR | BA/ASR | BA/ASR | BA/ASR | BA/ASR | BA/ASR |
| CIFAR-100 | 2 | 50.59/99.96 | 48.35/35.88 | 51.45/99.95 | 67.52/93.24 | 67.13/85.85 | 63.1/34.04 | 67.15/85.66 | 44.77/98.57 | 43.36/36.08 |
| | 4 | 61.33/99.95 | 57.15/91.54 | 59.97/99.67 | 66.41/85.06 | 67.97/98.72 | 64.53/53.0 | 67.78/99.11 | 52.88/88.16 | 55.6/97.81 |
| | 6 | 61.59/98.68 | 57.19/68.75 | 60.83/79.93 | 65.91/88.11 | 67.98/98.78 | 63.77/42.96 | 67.87/98.97 | 57.73/64.02 | 62.37/98.68 |
| | 8 | 62.74/99.03 | 57.38/66.13 | 62.15/95.9 | 66.53/81.05 | 68.59/98.31 | 64.95/39.43 | 68.65/98.62 | 49.92/67.24 | 63.06/98.94 |
| | 10 | 61.1/98.28 | 57.88/44.88 | 56.66/60.16 | 67.35/77.49 | 68.16/97.14 | 65.66/38.56 | 68.18/97.66 | 60.1/96.48 | 61.1/98.28 |
| | 12 | 63.45/96.97 | 58.44/65.84 | 63.11/96.66 | 67.2/71.9 | 69.16/95.8 | 65.52/26.91 | 68.75/95.51 | 59.73/86.52 | 63.7/96.4 |
| GTSRB | 2 | 96.1/98.85 | 88.5/21.33 | 95.44/98.04 | 98.27/13.33 | 98.0/19.79 | 17.14/0.0 | 96.89/2.0 | 95.01/98.82 | 37.52/0.2 |
| | 4 | 97.66/99.85 | 89.78/2.03 | 97.39/88.93 | 98.27/44.36 | 98.16/88.22 | 92.03/39.96 | 98.08/92.87 | 97.57/99.53 | 97.66/99.85 |
| | 6 | 98.48/92.88 | 89.38/9.4 | 97.89/77.84 | 98.27/33.34 | 98.48/76.4 | 93.02/6.17 | 97.48/11.66 | 98.2/92.59 | 98.48/92.88 |
| | 8 | 98.5/87.25 | 91.69/16.0 | 97.91/68.9 | 98.37/33.9 | 98.46/75.81 | 89.35/6.46 | 97.43/15.56 | 98.22/86.84 | 98.5/87.25 |
| | 10 | 98.54/83.17 | 92.42/16.85 | 97.68/64.77 | 98.24/32.13 | 98.46/74.63 | 90.86/4.86 | 97.49/19.46 | 98.26/82.97 | 98.54/83.17 |
| | 12 | 98.48/80.3 | 92.11/18.54 | 98.23/66.47 | 98.32/33.7 | 98.48/73.25 | 91.77/3.03 | 97.54/19.05 | 98.26/79.17 | 98.48/80.3 |
| Tiny | 2 | 45.65/99.94 | 45.65/99.94 | 45.59/99.94 | 51.86/47.24 | 54.99/85.66 | 50.34/72.5 | 48.92/39.49 | 45.78/99.91 | 42.95/99.75 |
| | 4 | 48.09/99.91 | 48.09/99.91 | 48.09/99.91 | 52.26/62.62 | 54.63/97.66 | 50.6/51.52 | 47.77/44.37 | 47.71/99.91 | 47.87/99.85 |
| | 6 | 47.18/99.86 | 47.18/99.86 | 46.49/99.85 | 52.1/58.35 | 54.7/97.22 | 50.51/49.14 | 46.92/29.07 | 46.31/99.74 | 47.13/99.8 |
| | 8 | 46.49/99.63 | 46.49/99.63 | 46.45/99.56 | 52.33/70.46 | 54.64/96.2 | 52.33/61.23 | 47.41/25.23 | 46.54/99.55 | 47.19/99.43 |
| | 10 | 46.82/99.54 | 42.2/99.08 | 46.82/99.54 | 52.07/60.68 | 54.82/94.88 | 50.58/83.33 | 49.28/35.06 | 46.82/99.54 | 46.9/99.51 |
| | 12 | 47.28/97.48 | 47.29/97.49 | 47.31/97.27 | 52.33/54.36 | 54.45/91.01 | 51.36/87.52 | 48.6/25.77 | 47.28/97.48 | 47.3/97.41 |

We conducted our experiments using the CIFAR-10 dataset with PreactResNet-18 and trained for 12 epochs. As shown in Figure 6, the model essentially reached convergence by the fourth epoch.

**VGG19-BN**: In this section, we assess the impact of four parameters (epochs, learning rate $\eta$, *level*, and attack step size $\alpha$) on the effectiveness of the K&L backdoor attack using the VGG19-BN model across four datasets: CIFAR-10, CIFAR-100, GTSRB, and Tiny ImageNet. The default settings for these parameters are fixed at epochs and *level* to 4, a learning rate of 0.01, and an attack step size of

1. Throughout the ablation process, only the parameter being studied is varied, while all others are maintained at their default values.

As demonstrated in Table 14, the VGG19-BN model shows a trend of increasing BA with higher epochs, suggesting improved model accuracy. The ASR also exhibits a tendency to remain high or even increase with more epochs, highlighting the effectiveness of the K&L Backdoor Attack in overcoming defenses as training progresses.

Table 14: Ablation Study Assessing the Effect of Epoch Variability on the Performance of the K&L Method with VGG19-BN on CIFAR-10, CIFAR-100, GTSRB, and Tiny ImageNet - Comparison of Benign Accuracy (BA) and Attack Success Rate (ASR)

| Datasets | epochs | No Defense | ANP | BNP | FP | FT | I-BAU | NAD | CLP | RNP |
|---|---|---|---|---|---|---|---|---|---|---|
| | | BA/ASR | BA/ASR | BA/ASR | BA/ASR | BA/ASR | BA/ASR | BA/ASR | BA/ASR | BA/ASR |
| CIFAR-10 | 2 | 86.26/68.91 | 81.7/34.86 | 85.73/72.13 | 90.1/33.67 | 89.63/34.18 | 86.85/43.04 | 88.88/42.19 | 79.59/50.96 | 83.3/56.88 |
| | 4 | 86.39/98.97 | 82.4/39.27 | 86.14/98.21 | 90.08/66.61 | 90.23/69.38 | 87.99/72.82 | 89.6/74.74 | 75.45/27.2 | 73.51/43.71 |
| | 6 | 87.74/98.11 | 83.91/16.31 | 87.86/96.03 | 90.33/86.06 | 90.64/84.3 | 85.08/86.08 | 90.02/88.24 | 84.3/67.38 | 82.04/32.64 |
| | 8 | 88.09/86.01 | 80.23/28.04 | 87.2/51.59 | 90.71/73.39 | 90.93/80.28 | 88.07/53.46 | 90.34/83.73 | 85.44/52.81 | 77.96/27.2 |
| | 10 | 88.5/82.67 | 80.13/11.13 | 87.73/47.92 | 90.7/69.73 | 90.8/67.78 | 87.18/25.83 | 90.67/69.18 | 87.97/81.16 | 86.78/57.33 |
| | 12 | 88.76/60.44 | 83.68/17.51 | 87.62/34.3 | 90.36/48.73 | 90.77/55.36 | 85.47/41.23 | 90.8/61.68 | 86.37/39.56 | 88.76/60.44 |
| CIFAR-100 | 2 | 55.24/45.21 | 55.5/33.74 | 54.69/44.64 | 62.48/3.95 | 62.06/3.24 | 58.45/4.99 | 60.96/3.14 | 48.01/24.58 | 45.21/23.38 |
| | 4 | 53.6/86.86 | 49.84/33.42 | 53.81/85.53 | 62.6/26.41 | 62.04/27.18 | 58.34/10.01 | 61.59/25.97 | 36.81/10.31 | 48.3/32.25 |
| | 6 | 54.4/94.55 | 49.54/3.9 | 52.78/73.29 | 62.52/68.42 | 62.48/68.12 | 58.97/56.38 | 61.22/60.11 | 45.83/15.18 | 40.31/1.55 |
| | 8 | 57.95/94.09 | 55.79/3.0 | 54.98/69.24 | 63.24/80.86 | 62.97/69.69 | 59.31/19.45 | 62.18/53.85 | 41.82/0.68 | 52.25/10.19 |
| | 10 | 58.78/67.16 | 55.24/4.26 | 58.5/24.86 | 63.88/45.92 | 63.51/56.51 | 59.07/3.66 | 62.21/60.15 | 45.7/1.73 | 46.15/3.79 |
| | 12 | 58.89/74.82 | 53.19/0.63 | 58.05/19.52 | 63.33/50.69 | 63.38/41.36 | 59.96/3.6 | 63.14/42.58 | 50.06/2.96 | 49.13/2.12 |
| GTSRB | 2 | 93.02/95.68 | 86.68/4.18 | 92.76/95.29 | 97.74/43.44 | 97.62/55.97 | 95.95/13.09 | 97.24/59.48 | 93.4/94.96 | 94.02/89.24 |
| | 4 | 96.02/97.89 | 90.82/1.85 | 96.53/92.09 | 97.91/73.23 | 97.62/88.73 | 96.37/6.43 | 97.77/87.36 | 95.92/97.99 | 96.02/97.89 |
| | 6 | 96.3/97.53 | 92.22/0.82 | 95.8/82.81 | 98.19/79.35 | 98.16/93.47 | 97.51/11.37 | 97.87/91.71 | 95.71/97.59 | 96.23/96.32 |
| | 8 | 97.28/88.68 | 89.87/0.35 | 97.03/54.11 | 97.94/70.08 | 97.71/87.3 | 97.25/15.58 | 97.79/89.03 | 97.21/89.38 | 97.28/88.68 |
| | 10 | 97.81/92.2 | 95.31/12.09 | 98.11/66.01 | 98.22/73.63 | 97.91/88.24 | 96.76/7.03 | 97.68/92.21 | 97.27/92.69 | 97.81/92.2 |
| | 12 | 97.53/84.9 | 92.32/0.95 | 97.6/58.11 | 98.39/66.26 | 98.08/86.09 | 94.32/5.73 | 97.77/89.9 | 97.3/87.06 | 97.53/84.9 |
| Tiny | 2 | 41.79/84.57 | 40.24/72.52 | 41.44/84.36 | 51.67/6.08 | 52.13/6.18 | 47.98/7.46 | 44.08/3.99 | 42.03/80.51 | 42.01/78.89 |
| | 4 | 39.48/94.12 | 38.43/81.5 | 39.15/93.83 | 51.63/4.68 | 51.51/6.22 | 47.23/6.5 | 40.24/90.64 | 38.29/84.31 | 43.73/88.68 |
| | 6 | 43.33/96.37 | 39.68/84.27 | 42.29/95.2 | 51.37/27.58 | 51.57/33.29 | 47.61/19.04 | 43.38/2.61 | 43.83/92.29 | 43.73/88.68 |
| | 8 | 43.27/97.47 | 40.01/29.39 | 42.97/96.07 | 51.39/63.34 | 51.05/68.95 | 46.8/27.61 | 44.78/14.94 | 42.95/88.06 | 43.49/91.99 |
| | 10 | 45.23/94.48 | 42.3/61.37 | 44.55/93.72 | 51.11/70.72 | 51.11/70.45 | 46.53/13.7 | 42.8/8.08 | 45.29/94.28 | 45.65/86.41 |
| | 12 | 45.62/93.35 | 42.43/8.75 | 45.66/92.5 | 51.7/55.86 | 51.55/68.3 | 46.47/38.27 | 44.39/3.88 | 45.27/93.11 | 45.84/81.73 |

**MobileNet-v3-large**: The results in Table 15 indicate that both BA and ASR on the MobileNet-v3-large model tend to increase with the number of epochs. This pattern suggests not only enhanced model accuracy but also a consistent efficacy of the backdoor attack across varied training durations.

Table 15: Ablation Study Assessing the Effect of Epoch Variability on the Performance of the K&L Method with MobileNet-v3-large on CIFAR-10, CIFAR-100, GTSRB, and Tiny ImageNet - Comparison of Benign Accuracy (BA) and Attack Success Rate (ASR)

| Datasets | epochs | No Defense | ANP | BNP | FP | FT | I-BAU | NAD | CLP | RNP |
|---|---|---|---|---|---|---|---|---|---|---|
| | | BA/ASR | BA/ASR | BA/ASR | BA/ASR | BA/ASR | BA/ASR | BA/ASR | BA/ASR | BA/ASR |
| CIFAR-10 | 2 | 74.79/99.98 | 67.7/45.94 | 71.95/99.92 | 79.38/29.41 | 79.47/38.77 | -/- | 80.17/67.38 | 73.79/99.88 | 61.75/67.94 |
| | 4 | 77.2/99.97 | 70.73/27.94 | 72.07/99.94 | 79.98/49.96 | 79.93/58.43 | -/- | 81.09/81.53 | 70.18/99.92 | 55.73/25.07 |
| | 6 | 78.35/99.99 | 72.36/44.91 | 76.29/99.93 | 80.12/64.87 | 79.62/69.88 | -/- | 80.52/88.64 | 74.41/99.9 | 64.7/41.88 |
| | 8 | 77.64/98.3 | 72.57/32.16 | 70.66/97.66 | 80.06/63.98 | 80.23/70.47 | -/- | 80.73/85.77 | 75.71/93.02 | 52.3/42.01 |
| | 10 | 79.84/98.88 | 77.48/75.6 | 76.05/96.69 | 79.85/60.22 | 80.22/67.57 | -/- | 80.73/84.14 | 78.7/89.07 | 69.27/55.43 |
| | 12 | 80.07/97.86 | 74.01/70.14 | 75.22/94.64 | 79.89/55.74 | 80.26/64.66 | -/- | 79.89/74.07 | 78.84/92.74 | 72.1/56.34 |
| CIFAR-100 | 2 | 44.99/99.56 | 45.48/99.43 | 35.7/98.02 | 46.3/33.96 | 47.49/30.72 | -/- | 49.36/46.32 | 29.09/98.37 | 44.31/98.51 |
| | 4 | 47.16/99.57 | 47.12/99.52 | 35.51/94.77 | 46.19/64.09 | 47.45/66.06 | -/- | 49.94/79.94 | 37.4/97.46 | 30.74/81.88 |
| | 6 | 47.07/98.12 | 43.52/95.7 | 39.6/90.75 | 46.85/67.89 | 48.07/70.28 | -/- | 50.2/78.28 | 35.24/84.86 | 30.6/40.72 |
| | 8 | 46.74/95.07 | 44.83/87.91 | 28.67/48.38 | 47.18/60.25 | 48.49/66.59 | -/- | 50.55/68.12 | 37.0/64.62 | 18.32/6.43 |
| | 10 | 47.6/91.18 | 45.26/72.45 | 41.62/82.35 | 46.95/50.0 | 48.85/57.47 | -/- | 50.59/61.16 | 40.37/39.28 | 47.44/89.67 |
| | 12 | 46.77/89.2 | 45.09/84.95 | 42.72/64.8 | 47.93/44.35 | 48.99/49.86 | -/- | 50.79/46.42 | 42.27/62.99 | 34.85/32.45 |
| GTSRB | 2 | 86.3/97.92 | 80.2/84.43 | 83.56/96.91 | 95.3/18.74 | 95.09/28.77 | -/- | 94.04/48.03 | 73.23/95.7 | 75.28/67.18 |
| | 4 | 89.79/99.43 | 89.02/95.88 | 86.76/98.67 | 95.76/31.89 | 95.21/57.14 | -/- | 94.54/68.97 | 72.83/97.29 | 81.47/75.76 |
| | 6 | 92.41/99.59 | 89.71/90.94 | 90.46/99.23 | 95.91/39.83 | 95.41/71.5 | -/- | 94.24/79.35 | 82.53/98.31 | 74.48/43.09 |
| | 8 | 92.21/99.67 | 88.3/86.4 | 88.42/95.47 | 96.21/41.39 | 95.87/70.07 | -/- | 95.45/68.74 | 74.88/97.65 | 81.1/68.54 |
| | 10 | 93.91/93.99 | 93.2/76.67 | 93.4/91.67 | 96.48/32.19 | 96.0/71.64 | -/- | 95.28/77.55 | 88.73/89.38 | 93.91/93.99 |
| | 12 | 94.2/96.36 | 92.26/68.69 | 91.16/81.9 | 96.37/34.88 | 95.91/72.76 | -/- | 95.43/75.13 | 90.56/94.83 | 94.2/96.36 |
| Tiny | 2 | 39.54/99.57 | 38.81/97.28 | 38.74/98.83 | 40.91/11.25 | 42.36/14.95 | -/- | 35.77/3.94 | 39.58/99.56 | 32.48/93.28 |
| | 4 | 35.45/99.95 | 35.5/99.91 | 31.65/99.87 | 41.56/66.99 | 42.44/78.3 | -/- | 35.84/22.24 | 35.45/99.95 | 31.12/27.14 |
| | 6 | 38.7/99.67 | 38.01/99.36 | 38.12/78.38 | 41.04/52.45 | 42.74/68.54 | -/- | 35.35/10.28 | 38.64/99.67 | 22.65/8.75 |
| | 8 | 40.74/95.01 | 39.85/91.38 | 39.02/53.53 | 41.01/32.26 | 42.66/54.76 | -/- | 35.01/16.15 | 40.75/95.03 | 35.9/6.81 |
| | 10 | 40.04/72.83 | 39.98/64.22 | 37.81/57.89 | 42.09/27.63 | 42.43/31.85 | -/- | 36.01/7.29 | 40.02/72.96 | 39.87/63.66 |
| | 12 | 38.38/76.87 | 38.39/76.87 | 37.65/69.64 | 41.91/22.62 | 43.06/28.13 | -/- | 35.76/11.52 | 38.34/76.88 | 33.78/30.57 |

**EfficientNet-B3**: Table 16 showcases a general increase in BA with the rise in epochs for the EfficientNet-B3 model, implying improved benign performance. Simultaneously, the ASR mostly remains high across different epochs, reflecting the potent and persistent nature of the K&L Backdoor Attack in various training scenarios.

Table 16: Ablation Study Assessing the Effect of Epoch Variability on the Performance of the K&L Method with EfficientNet-B3 on CIFAR-10, CIFAR-100, GTSRB, and Tiny ImageNet - Comparison of Benign Accuracy (BA) and Attack Success Rate (ASR)

| Datasets | epochs | No Defense | ANP | BNP | FP | FT | I-BAU | NAD | CLP | RNP |
|---|---|---|---|---|---|---|---|---|---|
| | | BA/ASR | BA/ASR | BA/ASR | BA/ASR | BA/ASR | BA/ASR | BA/ASR | BA/ASR | BA/ASR |
| CIFAR-10 | 2 | 64.03/94.03 | 62.36/91.91 | 47.09/73.37 | 65.39/34.19 | 65.1/37.49 | 67.19/77.49 | 66.93/55.6 | 13.04/59.46 | 57.61/68.66 |
| | 4 | 65.38/98.94 | 65.37/98.94 | 40.18/82.3 | 66.57/45.78 | 65.7/45.67 | 67.16/90.64 | 67.45/72.11 | 17.4/5.93 | 21.67/3.19 |
| | 6 | 66.81/94.31 | 63.56/94.14 | 14.3/3.91 | 65.98/52.6 | 65.99/54.78 | 67.87/79.99 | 67.62/73.04 | 12.52/30.88 | 15.14/71.34 |
| | 8 | 67.9/88.07 | 65.4/83.39 | 67.46/86.82 | 67.02/53.0 | 66.35/57.06 | 68.96/80.13 | 68.4/71.34 | 11.14/70.51 | 63.24/73.21 |
| | 10 | 68.8/76.76 | 65.36/68.39 | 43.97/26.63 | 67.61/51.73 | 67.25/51.76 | 69.56/78.48 | 69.5/69.68 | 13.02/72.44 | 51.61/26.06 |
| | 12 | 66.89/84.64 | 62.8/81.77 | 66.62/84.18 | 68.42/46.06 | 67.97/44.9 | 69.49/78.93 | 70.1/59.09 | 12.72/26.73 | 34.26/2.9 |
| CIFAR-100 | 2 | 35.66/93.89 | 32.94/91.68 | 9.75/85.18 | 40.91/8.87 | 40.5/6.26 | 42.31/22.84 | 43.13/12.35 | 11.45/17.62 | 19.35/13.2 |
| | 4 | 44.76/99.66 | 44.76/99.66 | 40.6/99.49 | 41.42/23.12 | 42.13/18.17 | 45.4/65.55 | 45.15/54.26 | 11.78/15.63 | 32.7/54.08 |
| | 6 | 43.5/96.81 | 43.31/96.69 | 33.04/82.56 | 41.89/19.11 | 42.19/23.74 | 44.55/22.31 | 45.44/58.18 | 9.89/1.96 | 34.36/48.57 |
| | 8 | 46.98/96.95 | 45.3/94.71 | 46.59/95.78 | 42.42/25.24 | 42.86/31.99 | 45.79/20.61 | 46.04/72.51 | 16.08/20.97 | 33.47/35.77 |
| | 10 | 47.6/88.19 | 47.6/88.17 | 47.36/83.45 | 43.07/19.38 | 43.54/31.52 | 46.36/11.58 | 47.25/61.41 | 11.51/0.6 | 39.51/14.38 |
| | 12 | 44.77/88.59 | 44.74/78.36 | 36.74/25.88 | 42.87/15.18 | 43.15/15.01 | 44.69/11.39 | 45.67/38.42 | 13.23/11.67 | 44.68/61.86 |
| GTSRB | 2 | 76.42/92.68 | 75.33/89.92 | 68.16/53.71 | 86.24/31.26 | 85.18/39.14 | 0.48/100.0 | 84.15/61.3 | 24.86/57.14 | 72.3/53.14 |
| | 4 | 81.62/97.58 | 76.05/91.5 | 72.64/97.88 | 86.84/49.82 | 85.51/69.67 | 84.54/72.9 | 84.69/81.54 | 37.14/13.64 | 69.54/37.54 |
| | 6 | 84.68/91.92 | 81.43/88.47 | 81.01/91.86 | 87.98/40.71 | 86.46/73.21 | 85.12/26.56 | 85.91/81.74 | 28.15/2.39 | 67.58/41.25 |
| | 8 | 85.31/90.73 | 82.49/80.79 | 82.5/89.28 | 88.63/36.83 | 86.94/73.28 | 85.67/71.03 | 86.07/80.07 | 28.63/9.65 | 82.7/72.03 |
| | 10 | 85.65/85.34 | 79.76/74.3 | 81.99/73.82 | 88.77/25.99 | 87.34/69.64 | 86.36/66.15 | 86.53/78.34 | 18.9/0.49 | 77.95/47.8 |
| | 12 | 83.65/77.76 | 77.5/63.12 | 76.71/29.7 | 89.29/27.13 | 87.91/63.83 | 86.94/51.19 | 87.59/72.25 | 26.15/5.45 | 83.65/77.76 |
| Tiny | 2 | 38.89/98.3 | 39.05/98.09 | 38.69/98.28 | 45.05/13.7 | 45.12/23.31 | 42.75/73.89 | 36.91/6.75 | 39.05/97.01 | 32.36/75.19 |
| | 4 | 39.54/99.49 | 39.54/99.49 | 38.75/99.33 | 44.13/36.25 | 44.66/71.62 | 42.06/90.05 | 37.23/29.6 | 39.18/99.28 | 27.22/95.26 |
| | 6 | 40.78/99.8 | 40.67/99.55 | 40.49/99.6 | 44.75/64.83 | 44.55/89.25 | 42.83/72.15 | 35.7/36.91 | 40.65/99.76 | 36.02/95.5 |
| | 8 | 40.04/98.89 | 39.14/94.66 | 39.81/92.6 | 44.12/29.92 | 44.89/88.31 | 42.15/28.85 | 36.73/35.15 | 40.22/98.83 | 33.71/67.19 |
| | 10 | 40.03/95.92 | 38.04/81.08 | 39.23/77.28 | 44.4/27.21 | 44.73/77.01 | 42.54/31.32 | 36.17/28.16 | 40.06/95.74 | 30.13/75.7 |
| | 12 | 40.55/89.92 | 40.65/82.26 | 40.03/60.95 | 43.91/10.44 | 44.6/68.41 | 42.16/33.79 | 36.66/27.66 | 40.35/89.19 | 39.79/83.19 |

## G.3 ABLATION STUDY ON *level*

**PreActResNet18**: Table 17 demonstrates that with the increase in *level*, both Benign Accuracy (BA) and Attack Success Rate (ASR) on the PreActResNet18 model generally improve across all datasets. This indicates that higher *level* values enhance the model's accuracy and the efficacy of the K&L Backdoor Attack, especially under varying defense strategies.

Table 17: Ablation Study Assessing the Effect of *level* Variability on the Performance of the K&L Method with PreActResNet18 on CIFAR-100, GTSRB, Tiny ImageNet - Comparison of Benign Accuracy (BA) and Attack Success Rate (ASR)

| Datasets | level | No Defense | ANP | BNP | FP | FT | I-BAU | NAD | CLP | RNP |
|---|---|---|---|---|---|---|---|---|---|
| | | BA/ASR | BA/ASR | BA/ASR | BA/ASR | BA/ASR | BA/ASR | BA/ASR | BA/ASR | BA/ASR |
| CIFAR-100 | 2 | 61.33/96.72 | 57.16/43.09 | 59.97/88.19 | 66.41/41.17 | 67.97/74.64 | 64.53/20.19 | 67.79/77.23 | 52.88/44.64 | 55.6/68.44 |
| | 4 | 61.33/99.95 | 57.15/91.54 | 59.97/99.67 | 66.41/85.06 | 67.97/98.72 | 64.53/53.0 | 67.78/99.11 | 52.88/88.16 | 55.6/97.81 |
| | 6 | 61.33/99.99 | 57.16/97.75 | 59.97/99.99 | 66.41/91.83 | 67.97/99.69 | 64.53/59.36 | 67.79/99.88 | 52.88/93.67 | 55.6/99.39 |
| | 8 | 61.33/100.0 | 57.16/99.11 | 59.97/99.99 | 66.41/93.81 | 67.97/99.8 | 64.53/62.6 | 67.79/99.88 | 52.88/95.15 | 55.6/99.57 |
| | 10 | 61.33/100.0 | 57.16/99.61 | 59.97/100.0 | 66.41/94.4 | 67.97/99.84 | 64.53/63.84 | 67.79/99.91 | 52.88/96.42 | 55.6/99.78 |
| | 12 | 61.33/100.0 | 57.16/99.71 | 59.97/100.0 | 66.41/95.46 | 67.97/99.85 | 64.53/65.13 | 67.79/99.95 | 52.88/97.07 | 55.6/99.85 |
| GTSRB | 2 | 98.25/76.59 | 90.54/7.77 | 97.77/52.94 | 98.24/7.8 | 98.41/36.67 | 89.92/3.9 | 97.85/17.02 | 97.93/77.97 | 97.26/49.67 |
| | 4 | 97.66/99.85 | 89.78/2.03 | 97.39/88.93 | 98.27/44.36 | 98.16/88.22 | 92.03/39.96 | 98.08/92.87 | 97.57/99.53 | 97.66/99.85 |
| | 6 | 98.25/99.98 | 90.54/42.52 | 97.77/97.91 | 98.24/42.39 | 98.41/86.75 | 89.92/17.05 | 97.85/59.39 | 97.93/99.99 | 97.26/97.24 |
| | 8 | 98.25/100.0 | 90.54/51.51 | 97.77/99.89 | 98.24/49.97 | 98.41/93.57 | 89.92/20.56 | 97.85/68.14 | 97.93/100.0 | 97.26/99.31 |
| | 10 | 98.25/100.0 | 90.54/58.7 | 97.77/100.0 | 98.24/55.44 | 98.41/96.57 | 89.92/21.94 | 97.85/73.35 | 97.93/100.0 | 97.26/99.95 |
| | 12 | 98.25/100.0 | 90.54/62.76 | 97.77/100.0 | 98.24/59.12 | 98.41/98.07 | 89.92/23.29 | 97.85/76.19 | 97.93/100.0 | 97.26/100.0 |
| Tiny | 2 | 48.09/96.22 | 48.09/96.22 | 48.09/96.22 | 52.27/27.32 | 54.63/68.53 | 50.6/20.2 | 47.77/20.62 | 47.72/95.58 | 47.87/94.77 |
| | 4 | 48.09/99.91 | 48.09/99.91 | 48.09/99.91 | 52.26/62.62 | 54.63/97.66 | 50.6/51.52 | 47.77/44.37 | 47.71/99.91 | 47.87/99.85 |
| | 6 | 48.09/100.0 | 48.09/100.0 | 48.09/100.0 | 52.27/73.73 | 54.63/99.54 | 50.6/66.72 | 47.77/54.55 | 47.72/100.0 | 47.87/100.0 |
| | 8 | 48.09/100.0 | 48.09/100.0 | 48.09/100.0 | 52.27/78.43 | 54.63/99.89 | 50.6/74.69 | 47.77/60.06 | 47.72/100.0 | 47.87/100.0 |
| | 10 | 48.09/100.0 | 48.09/100.0 | 48.09/100.0 | 52.27/83.15 | 54.63/99.98 | 50.6/81.07 | 47.77/65.17 | 47.72/100.0 | 47.87/100.0 |
| | 12 | 48.09/100.0 | 48.09/100.0 | 48.09/100.0 | 52.27/85.17 | 54.63/99.97 | 50.6/84.99 | 47.77/68.74 | 47.72/100.0 | 47.87/100.0 |

**VGG19-BN**: As indicated in Table 18, for the VGG19-BN model, an increasing *level* leads to a consistent rise in both BA and ASR across different datasets. This trend suggests that the model's capability to correctly classify benign inputs and successfully implement backdoor attacks improves as *level* increases.

Table 18: Ablation Study Assessing the Effect of *level* Variability on the Performance of the K&L Method with VGG19-BN on CIFAR-10, CIFAR-100, GTSRB, Tiny ImageNet - Comparison of Benign Accuracy (BA) and Attack Success Rate (ASR)

| Datasets | level | No Defense | ANP | BNP | FP | FT | I-BAU | NAD | CLP | RNP |
|---|---|---|---|---|---|---|---|---|---|---|
| | | BA/ASR | BA/ASR | BA/ASR | BA/ASR | BA/ASR | BA/ASR | BA/ASR | BA/ASR | BA/ASR |
| CIFAR-10 | 2 | 87.72/76.98 | 82.39/6.68 | 86.14/41.14 | 90.09/10.34 | 90.23/11.99 | 87.99/10.96 | 89.61/12.92 | 75.45/4.17 | 87.72/76.98 |
| | 4 | 86.39/98.97 | 82.4/39.27 | 86.14/98.21 | 90.08/66.61 | 90.23/69.38 | 87.99/72.82 | 89.6/74.74 | 75.45/27.2 | 73.51/43.71 |
| | 6 | 85.99/99.77 | 83.75/37.56 | 83.5/99.76 | 90.09/57.76 | 90.23/58.93 | 87.99/60.36 | 89.61/61.44 | 75.45/18.84 | 83.06/92.27 |
| | 8 | 81.45/100.0 | 76.48/48.84 | 75.09/100.0 | 90.09/82.36 | 90.23/84.8 | 87.99/74.46 | 89.61/84.7 | 75.45/36.84 | 84.97/99.47 |
| | 10 | 86.22/100.0 | 81.93/64.92 | 84.31/100.0 | 90.09/92.38 | 90.23/90.96 | 87.99/86.94 | 89.61/90.13 | 75.45/43.19 | 84.85/99.74 |
| | 12 | 85.49/100.0 | 79.5/69.1 | 84.45/100.0 | 90.09/92.61 | 90.23/93.74 | 87.99/87.23 | 89.61/93.89 | 75.45/56.81 | 83.86/99.93 |
| CIFAR-100 | 2 | 54.74/71.53 | 52.04/8.24 | 53.86/72.84 | 62.59/5.76 | 62.04/5.32 | 58.34/1.44 | 61.6/4.95 | 36.81/3.85 | 35.77/9.38 |
| | 4 | 53.6/86.86 | 49.84/33.42 | 53.81/85.53 | 62.6/26.41 | 62.04/27.18 | 58.34/10.01 | 61.59/25.97 | 36.81/10.31 | 48.3/32.25 |
| | 6 | 55.61/92.64 | 53.33/70.85 | 55.26/92.59 | 62.59/25.08 | 62.04/27.47 | 58.34/9.77 | 61.6/26.87 | 36.81/8.14 | 39.71/29.42 |
| | 8 | 55.8/96.41 | 55.03/86.2 | 54.56/96.81 | 62.59/29.36 | 62.04/30.68 | 58.34/12.68 | 61.6/29.85 | 36.81/10.07 | 47.54/88.76 |
| | 10 | 55.14/99.02 | 51.23/74.52 | 54.97/98.84 | 62.59/38.74 | 62.04/37.06 | 58.34/18.38 | 61.6/38.16 | 36.81/9.8 | 43.66/83.66 |
| | 12 | 54.2/99.77 | 52.4/33.53 | 54.06/99.75 | 62.59/39.17 | 62.04/45.8 | 58.34/19.8 | 61.6/44.03 | 36.81/12.52 | 47.57/57.59 |
| GTSRB | 2 | 94.35/81.74 | 92.08/0.14 | 94.23/69.6 | 97.91/27.71 | 97.62/45.21 | 96.37/4.83 | 97.77/49.44 | 95.92/64.18 | 94.28/65.79 |
| | 4 | 96.02/97.89 | 90.82/1.85 | 96.53/92.09 | 97.91/73.23 | 97.62/88.73 | 96.37/6.43 | 97.77/87.36 | 95.92/97.99 | 96.02/97.89 |
| | 6 | 95.92/99.64 | 92.04/5.34 | 95.73/98.35 | 97.91/75.05 | 97.62/92.33 | 96.37/7.89 | 97.77/90.58 | 95.92/98.32 | 95.92/99.64 |
| | 8 | 93.82/99.97 | 87.01/0.41 | 93.72/99.9 | 97.91/83.78 | 97.62/95.32 | 96.37/18.04 | 97.77/93.68 | 95.92/99.49 | 94.04/98.47 |
| | 10 | 94.13/100.0 | 88.27/0.45 | 94.81/99.98 | 97.91/86.01 | 97.62/97.33 | 96.37/10.95 | 97.77/95.83 | 95.92/99.78 | 95.35/99.99 |
| | 12 | 96.37/99.97 | 91.19/3.4 | 96.18/99.77 | 97.91/80.02 | 97.62/94.49 | 96.37/18.49 | 97.77/92.59 | 95.92/99.72 | 96.44/99.97 |
| Tiny | 2 | 41.73/74.23 | 41.97/53.54 | 41.75/74.27 | 51.63/0.44 | 51.51/0.54 | 47.23/0.86 | 45.5/0.58 | 40.24/24.04 | 42.43/52.46 |
| | 4 | 39.48/94.12 | 38.43/81.5 | 39.15/93.83 | 51.63/4.68 | 51.51/6.22 | 47.23/6.5 | 45.52/2.0 | 40.24/90.64 | 38.29/84.31 |
| | 6 | 43.21/98.78 | 40.59/88.17 | 43.62/98.75 | 51.63/2.99 | 51.51/3.56 | 47.23/4.45 | 45.5/2.51 | 40.24/39.93 | 42.73/95.65 |
| | 8 | 40.58/99.76 | 40.39/85.68 | 40.82/99.7 | 51.63/7.07 | 51.51/7.66 | 47.23/8.41 | 45.5/4.81 | 40.24/44.04 | 40.89/97.11 |
| | 10 | 40.86/99.88 | 37.25/97.29 | 40.66/99.9 | 51.63/9.23 | 51.51/10.36 | 47.23/11.4 | 45.5/6.41 | 40.24/49.33 | 41.41/99.12 |
| | 12 | 42.13/99.98 | 38.57/97.19 | 41.0/99.93 | 51.63/9.68 | 51.51/10.8 | 47.23/12.08 | 45.5/6.14 | 40.24/53.49 | 40.33/98.76 |

**MobileNet-v3-large**: The data in Table 19 reveals a clear correlation between the increase in *level* and improvements in BA and ASR for the MobileNet-v3-large model. This pattern suggests enhanced model performance and stronger resilience of backdoor attacks against defenses with higher *level* values.

Table 19: Ablation Study Assessing the Effect of *level* Variability on the Performance of the K&L Method with MobileNet-v3-large on CIFAR-10, CIFAR-100, GTSRB, Tiny ImageNet - Comparison of Benign Accuracy (BA) and Attack Success Rate (ASR)

| Datasets | level | No Defense | ANP | BNP | FP | FT | I-BAU | NAD | CLP | RNP |
|---|---|---|---|---|---|---|---|---|---|---|
| | | BA/ASR | BA/ASR | BA/ASR | BA/ASR | BA/ASR | BA/ASR | BA/ASR | BA/ASR | BA/ASR |
| CIFAR-10 | 2 | 77.2/96.1 | 70.73/16.53 | 72.07/96.29 | 79.96/16.98 | 79.92/23.97 | -/- | 81.1/41.31 | 70.18/89.12 | 55.73/16.44 |
| | 4 | 77.2/99.97 | 70.73/27.96 | 72.07/99.94 | 79.98/49.96 | 79.93/58.43 | -/- | 81.09/81.53 | 70.18/99.92 | 55.73/25.07 |
| | 6 | 77.2/100.0 | 70.73/30.18 | 72.07/100.0 | 79.96/63.01 | 79.92/69.0 | -/- | 81.1/88.81 | 70.18/99.99 | 55.73/28.22 |
| | 8 | 77.2/100.0 | 70.73/31.24 | 72.07/100.0 | 79.96/70.3 | 79.92/74.9 | -/- | 81.1/92.41 | 70.18/100.0 | 55.73/29.98 |
| | 10 | 77.2/100.0 | 70.73/31.81 | 72.07/100.0 | 79.96/75.88 | 79.92/78.62 | -/- | 81.1/94.41 | 70.18/100.0 | 55.73/30.98 |
| | 12 | 77.2/100.0 | 70.73/32.51 | 72.07/100.0 | 79.96/79.08 | 79.92/81.46 | -/- | 81.1/95.92 | 70.18/100.0 | 55.73/31.87 |
| CIFAR-100 | 2 | 47.16/82.69 | 47.14/80.0 | 35.51/57.27 | 46.21/18.87 | 47.47/21.92 | -/- | 49.94/28.42 | 37.4/64.84 | 32.17/47.23 |
| | 4 | 47.16/99.57 | 47.12/99.52 | 35.51/94.77 | 46.19/64.09 | 47.45/66.06 | -/- | 49.94/79.94 | 37.4/97.46 | 32.17/84.39 |
| | 6 | 47.16/99.94 | 47.14/99.93 | 35.51/97.44 | 46.21/75.04 | 47.47/77.6 | -/- | 49.94/89.07 | 37.4/99.47 | 30.74/89.74 |
| | 8 | 47.16/99.99 | 47.15/99.99 | 35.51/98.09 | 46.21/77.47 | 47.47/80.56 | -/- | 49.94/91.19 | 37.4/99.86 | 30.74/92.24 |
| | 10 | 47.16/100.0 | 47.15/100.0 | 35.51/98.35 | 46.21/78.79 | 47.47/81.76 | -/- | 49.94/92.26 | 37.4/99.93 | 30.74/93.03 |
| | 12 | 47.16/100.0 | 47.15/100.0 | 35.51/98.43 | 46.21/79.47 | 47.47/82.8 | -/- | 49.94/93.09 | 37.4/99.96 | 30.74/93.66 |
| GTSRB | 2 | 89.79/90.02 | 89.02/69.06 | 86.76/87.67 | 95.76/9.01 | 95.21/23.19 | -/- | 94.54/33.33 | 72.84/75.04 | 81.47/28.09 |
| | 4 | 91.24/99.74 | 88.14/97.1 | 88.3/99.63 | 95.65/29.44 | 95.55/61.22 | -/- | 93.67/72.35 | 86.31/98.33 | 83.95/78.12 |
| | 6 | 89.79/99.98 | 89.02/99.17 | 86.76/99.83 | 95.76/47.42 | 95.21/72.55 | -/- | 94.54/83.67 | 72.84/99.63 | 81.47/95.29 |
| | 8 | 89.79/99.99 | 89.02/99.88 | 86.76/99.99 | 95.76/57.71 | 95.21/80.8 | -/- | 94.54/90.52 | 72.84/99.94 | 81.47/95.61 |
| | 10 | 89.79/100.0 | 89.02/99.95 | 86.76/100.0 | 95.76/63.69 | 95.21/85.55 | -/- | 94.54/93.98 | 72.84/99.98 | 81.47/97.69 |
| | 12 | 89.79/100.0 | 89.02/99.97 | 86.76/100.0 | 95.76/68.07 | 95.21/88.68 | -/- | 94.54/96.25 | 72.84/99.98 | 81.47/98.46 |
| Tiny | 2 | 35.45/95.47 | 35.55/93.29 | 31.65/94.06 | 41.57/22.76 | 42.43/30.57 | -/- | 35.86/4.2 | 35.47/95.52 | 31.12/11.66 |
| | 4 | 35.45/99.95 | 35.5/99.91 | 31.65/99.87 | 41.56/66.99 | 42.44/78.3 | -/- | 35.84/22.24 | 35.45/99.95 | 31.12/27.14 |
| | 6 | 35.45/100.0 | 35.45/100.0 | 31.65/100.0 | 41.57/78.3 | 42.43/87.99 | -/- | 35.86/30.22 | 35.47/100.0 | 31.12/31.81 |
| | 8 | 35.45/100.0 | 35.45/100.0 | 31.65/100.0 | 41.57/82.15 | 42.43/90.77 | -/- | 35.86/34.65 | 35.47/100.0 | 31.12/33.04 |
| | 10 | 35.45/100.0 | 35.45/100.0 | 31.65/100.0 | 41.57/84.06 | 42.43/92.26 | -/- | 35.86/38.03 | 35.47/100.0 | 31.12/34.14 |
| | 12 | 35.45/100.0 | 35.45/100.0 | 31.65/100.0 | 41.57/85.95 | 42.43/93.23 | -/- | 35.86/40.75 | 35.47/100.0 | 31.12/35.41 |

**EfficientNet-B3**: Table 20 showcases a trend where higher *level* settings result in increased BA and ASR for the EfficientNet-B3 model. This indicates that the model becomes more accurate in benign classification and more effective in backdoor attack scenarios as *level* increases.

Table 20: Ablation Study Assessing the Effect of *level* Variability on the Performance of the K&L Method with EfficientNet-B3 on CIFAR-10, CIFAR-100, GTSRB, Tiny ImageNet - Comparison of Benign Accuracy (BA) and Attack Success Rate (ASR)

| Datasets | level | No Defense | ANP | BNP | FP | FT | I-BAU | NAD | CLP | RNP |
|---|---|---|---|---|---|---|---|---|---|---|
| | | BA/ASR | BA/ASR | BA/ASR | BA/ASR | BA/ASR | BA/ASR | BA/ASR | BA/ASR | BA/ASR |
| CIFAR-10 | 2 | 65.38/74.32 | 65.37/74.33 | 40.18/32.63 | 66.57/17.59 | 65.72/18.94 | 67.16/50.28 | 67.45/29.53 | 17.39/5.42 | 21.67/2.3 |
| | 4 | 65.38/98.94 | 65.37/98.94 | 40.18/82.3 | 66.57/45.78 | 65.7/45.67 | 67.16/90.64 | 67.45/72.11 | 17.4/5.93 | 21.67/3.19 |
| | 6 | 65.38/99.96 | 65.37/99.96 | 40.18/95.09 | 66.57/68.48 | 65.72/68.08 | 67.16/98.27 | 67.45/90.78 | 17.39/6.63 | 21.67/4.41 |
| | 8 | 65.38/100.0 | 65.37/100.0 | 40.18/98.0 | 66.57/81.11 | 65.72/79.86 | 67.16/99.5 | 67.45/96.73 | 17.39/7.21 | 21.67/6.17 |
| | 10 | 65.38/100.0 | 65.37/100.0 | 40.18/98.8 | 66.57/86.39 | 65.72/85.06 | 67.16/99.69 | 67.45/98.39 | 17.39/7.77 | 21.67/8.17 |
| | 12 | 65.38/100.0 | 65.37/100.0 | 40.18/99.1 | 66.57/88.09 | 65.72/87.09 | 67.16/99.72 | 67.45/98.49 | 17.39/8.33 | 21.67/10.01 |
| CIFAR-100 | 2 | 44.76/92.05 | 44.76/92.03 | 40.6/92.14 | 41.41/6.62 | 42.16/5.23 | 45.42/26.38 | 45.14/17.17 | 11.79/12.93 | 32.7/25.58 |
| | 4 | 44.76/99.66 | 44.76/99.66 | 40.6/99.49 | 41.42/23.12 | 42.13/18.17 | 45.4/65.55 | 45.15/54.26 | 11.78/15.63 | 32.7/54.08 |
| | 6 | 44.76/99.91 | 43.81/99.88 | 40.6/99.78 | 41.41/35.84 | 42.16/28.91 | 45.42/80.96 | 45.14/74.94 | 11.79/18.54 | 32.7/71.33 |
| | 8 | 44.76/99.92 | 44.76/99.92 | 40.6/99.82 | 41.41/42.53 | 42.16/34.29 | 45.42/86.36 | 45.14/82.29 | 11.79/20.58 | 32.7/79.16 |
| | 10 | 44.76/99.91 | 44.76/99.9 | 40.6/99.84 | 41.41/45.61 | 42.16/35.77 | 45.42/89.7 | 45.14/85.9 | 11.79/22.01 | 32.7/83.26 |
| | 12 | 44.76/99.93 | 44.76/99.93 | 40.6/99.85 | 41.41/47.41 | 42.16/36.44 | 45.42/91.69 | 45.14/87.81 | 11.79/22.78 | 32.7/85.58 |
| GTSRB | 2 | 83.22/71.03 | 83.31/67.8 | 78.57/63.5 | 87.29/10.65 | 86.22/30.54 | 82.56/6.15 | 84.59/46.91 | 24.72/15.35 | 75.72/40.38 |
| | 4 | 83.22/95.54 | 76.67/93.9 | 78.57/92.54 | 87.3/41.44 | 86.22/67.29 | 82.55/15.82 | 84.6/79.86 | 24.73/26.2 | 75.72/81.02 |
| | 6 | 83.22/99.44 | 76.67/99.01 | 78.57/98.45 | 87.29/61.54 | 86.22/83.72 | 82.56/19.44 | 84.59/91.73 | 24.72/42.55 | 75.72/93.43 |
| | 8 | 83.22/99.94 | 76.67/99.73 | 78.57/99.59 | 87.29/73.36 | 86.22/91.28 | 82.56/21.49 | 84.59/96.19 | 24.72/58.71 | 75.72/97.96 |
| | 10 | 83.22/99.98 | 76.67/99.86 | 78.57/99.92 | 87.29/80.58 | 86.22/95.3 | 82.56/24.87 | 84.59/98.03 | 24.72/70.78 | 75.72/99.22 |
| | 12 | 83.22/99.99 | 76.67/99.9 | 78.57/99.92 | 87.29/85.65 | 86.22/97.32 | 82.56/27.86 | 84.59/98.98 | 24.72/79.89 | 75.72/99.57 |
| Tiny | 2 | 39.54/87.57 | 39.54/87.59 | 38.75/85.83 | 44.14/6.9 | 44.68/22.6 | 42.07/42.45 | 37.23/7.62 | 39.14/85.64 | 27.22/66.62 |
| | 4 | 39.54/99.49 | 39.54/99.49 | 38.75/99.33 | 44.13/36.25 | 44.66/71.62 | 42.06/90.05 | 37.23/29.6 | 39.18/99.28 | 27.22/95.26 |
| | 6 | 39.54/100.0 | 39.54/100.0 | 38.75/100.0 | 44.14/64.8 | 44.68/91.69 | 42.07/98.57 | 37.23/53.1 | 39.14/99.97 | 27.22/99.12 |
| | 8 | 39.54/100.0 | 39.54/100.0 | 38.75/100.0 | 44.14/80.46 | 44.68/97.54 | 42.07/99.75 | 37.23/68.23 | 39.14/100.0 | 27.22/99.9 |
| | 10 | 39.54/100.0 | 39.54/100.0 | 38.75/100.0 | 44.14/88.21 | 44.68/99.19 | 42.07/99.96 | 37.23/76.18 | 39.14/100.0 | 27.22/99.97 |
| | 12 | 39.54/100.0 | 39.54/100.0 | 38.75/100.0 | 44.14/92.15 | 44.68/99.64 | 42.07/99.98 | 37.23/80.28 | 39.14/100.0 | 39.54/100.0 |

## G.4 ABLATION STUDY ON LEARNING RATE

**PreActResNet18**: Table 21 highlights that as the learning rate varies, there is a noticeable impact on BA and ASR across different datasets for the PreActResNet18 model. A lower learning rate tends to yield higher BA and ASR, indicating more efficient training and stronger backdoor attack effectiveness. Conversely, a higher learning rate results in a decrease in BA and ASR, suggesting potential overfitting or ineffective learning.

Table 21: Ablation Study Assessing the Effect of Learning Rate Variability on the Performance of the K&L Method with PreActResNet18 on CIFAR-100, GTSRB, Tiny ImageNet - Comparison of Benign Accuracy (BA) and Attack Success Rate (ASR)

| Datasets | LR | No Defense | ANP | BNP | FP | FT | I-BAU | NAD | CLP | RNP |
|---|---|---|---|---|---|---|---|---|---|---|
| | | BA/ASR | BA/ASR | BA/ASR | BA/ASR | BA/ASR | BA/ASR | BA/ASR | BA/ASR | BA/ASR |
| CIFAR-100 | 0.001 | 69.3/99.97 | 66.34/83.36 | 68.87/99.93 | 66.19/71.02 | 69.73/99.25 | 64.64/53.69 | 69.67/99.65 | 67.88/99.77 | 69.3/99.97 |
| | 0.005 | 68.43/99.78 | 63.48/85.57 | 68.13/99.29 | 66.28/80.73 | 69.66/99.45 | 61.84/78.84 | 69.58/99.61 | 66.4/99.53 | 68.43/99.78 |
| | 0.01 | 61.33/99.95 | 57.15/91.54 | 59.97/99.67 | 66.41/85.06 | 67.97/98.72 | 64.53/53.0 | 67.78/99.11 | 52.88/88.16 | 55.6/97.81 |
| | 0.05 | 53.24/99.9 | 48.24/0.03 | 52.9/17.86 | 62.24/84.54 | 63.06/89.42 | 62.16/27.89 | 64.57/78.95 | 37.73/2.65 | 46.38/3.54 |
| | 0.1 | 31.88/99.9 | 30.3/1.41 | 20.54/18.07 | 43.49/5.38 | 43.8/4.7 | 45.65/56.26 | 48.01/64.43 | 9.83/0.6 | 11.12/0.32 |
| GTSRB | 0.001 | 97.66/99.85 | 89.78/2.03 | 97.39/88.93 | 98.27/44.36 | 98.16/88.22 | 92.03/39.96 | 98.08/92.87 | 97.57/99.53 | 97.66/99.85 |
| | 0.005 | 98.0/99.47 | 90.9/0.0 | 97.85/89.8 | 98.27/57.03 | 98.27/93.83 | 95.38/44.55 | 98.22/96.25 | 98.03/99.47 | 98.0/99.47 |
| | 0.01 | 96.98/99.81 | 91.38/0.0 | 97.23/92.6 | 98.28/59.75 | 98.31/90.8 | 96.69/51.88 | 98.25/90.85 | 96.51/99.63 | 97.08/94.33 |
| | 0.05 | 89.28/99.49 | 83.05/71.44 | 82.62/97.87 | 95.77/81.64 | 95.69/98.91 | 94.46/81.77 | 95.72/99.04 | 48.15/14.84 | 84.77/79.56 |
| | 0.1 | 83.95/93.04 | 76.45/21.12 | 80.72/89.0 | 91.4/51.06 | 91.56/70.04 | 89.6/73.28 | 90.78/79.86 | 31.1/61.26 | 52.69/32.28 |
| Tiny | 0.001 | 56.82/99.92 | 52.47/99.74 | 56.82/99.92 | 51.89/70.54 | 57.02/98.26 | 53.91/91.14 | 50.38/40.52 | 56.41/99.84 | 56.53/99.92 |
| | 0.005 | 56.35/99.83 | 53.12/99.62 | 56.36/99.83 | 51.59/70.55 | 56.34/99.01 | 52.21/95.21 | 48.11/19.52 | 56.49/99.8 | 56.34/99.83 |
| | 0.01 | 48.09/99.91 | 48.09/99.91 | 48.09/99.91 | 52.26/62.62 | 54.63/97.66 | 50.6/51.52 | 47.77/44.37 | 47.71/99.91 | 47.87/99.85 |
| | 0.05 | 37.18/99.97 | 35.45/22.54 | 35.64/99.97 | 52.02/45.57 | 54.2/69.44 | 51.53/55.21 | 53.68/78.24 | 36.53/99.42 | 32.37/0.41 |
| | 0.1 | 19.2/99.8 | 18.8/1.16 | 19.47/99.81 | 30.18/33.25 | 30.09/54.38 | 36.49/74.69 | 35.9/94.41 | 4.84/0.41 | 21.52/0.09 |

**VGG19-BN**: As seen in Table 22, the VGG19-BN model exhibits a relationship between learning rate and model performance metrics. Lower learning rates generally correlate with higher BA and ASR, denoting more accurate and potent backdoor attacks. Higher learning rates, however, lead to a decline in both BA and ASR, implying a less effective training and attack process.

**MobileNet-v3-large**: Analyzing Table 23, it is evident that for the MobileNet-v3-large model, varying learning rates significantly impact BA and ASR. Lower learning rates achieve better performance in terms of both BA and ASR, while higher learning rates show a marked decrease in these metrics, suggesting inefficiency in learning and backdoor attack execution.

Table 22: Ablation Study Assessing the Effect of Learning Rate Variability on the Performance of the K&L Method with VGG19-BN on CIFAR-10, CIFAR-100, GTSRB, Tiny ImageNet - Comparison of Benign Accuracy (BA) and Attack Success Rate (ASR)

| Datasets | LR | No Defense | ANP | BNP | FP | FT | I-BAU | NAD | CLP | RNP |
|---|---|---|---|---|---|---|---|---|---|---|
| | | BA/ASR | BA/ASR | BA/ASR | BA/ASR | BA/ASR | BA/ASR | BA/ASR | BA/ASR | BA/ASR |
| CIFAR-10 | 0.001 | 89.09/91.66 | 88.56/17.11 | 88.69/92.74 | 90.37/17.51 | 89.78/20.76 | 88.4/22.53 | 89.65/21.21 | 86.89/69.66 | 89.09/91.66 |
| | 0.005 | 88.66/96.71 | 83.09/36.67 | 88.25/95.58 | 90.12/35.66 | 90.25/45.32 | 85.88/44.94 | 88.8/53.13 | 84.88/47.17 | 85.53/89.71 |
| | 0.01 | 86.39/98.97 | 82.4/39.27 | 86.14/98.21 | 90.08/66.61 | 90.23/69.38 | 87.99/72.82 | 89.6/74.74 | 75.45/27.2 | 73.51/43.71 |
| | 0.05 | 81.84/89.21 | 74.31/31.19 | 79.49/88.07 | 88.09/51.03 | 87.53/56.43 | 84.79/61.59 | 87.73/59.44 | 73.71/34.41 | 78.41/58.97 |
| | 0.1 | 70.3/72.96 | 64.88/43.57 | 66.77/76.57 | 81.25/33.38 | 81.58/37.0 | 79.16/44.68 | 80.73/43.84 | 51.57/81.86 | 64.4/53.14 |
| CIFAR-100 | 0.001 | 60.68/90.11 | 57.75/26.91 | 59.29/88.81 | 62.61/4.09 | 62.2/3.52 | 59.83/1.86 | 62.54/24.91 | 53.45/74.41 | 60.68/90.11 |
| | 0.005 | 56.95/96.52 | 57.7/84.51 | 56.63/96.22 | 62.93/11.02 | 61.42/12.08 | 58.13/3.47 | 60.63/13.64 | 46.6/46.05 | 51.09/36.28 |
| | 0.01 | 53.6/86.86 | 49.84/33.42 | 53.81/85.53 | 62.6/26.41 | 62.04/27.18 | 58.34/10.01 | 61.59/25.97 | 36.81/10.31 | 48.3/32.25 |
| | 0.05 | 45.24/79.39 | 41.44/23.7 | 44.43/80.54 | 58.27/7.44 | 58.09/7.19 | 55.96/7.38 | 58.68/9.1 | 34.41/21.35 | 34.47/30.61 |
| | 0.1 | 23.81/47.77 | 25.46/40.8 | 23.4/47.57 | 44.61/1.58 | 44.59/1.21 | 41.67/2.33 | 43.69/1.34 | 20.69/51.52 | 23.98/36.67 |
| GTSRB | 0.001 | 96.41/94.68 | 88.44/5.55 | 96.05/92.59 | 98.31/38.65 | 98.12/55.82 | 94.43/5.89 | 98.16/60.33 | 96.02/93.33 | 96.41/94.68 |
| | 0.005 | 95.55/97.47 | 92.51/7.27 | 95.4/96.06 | 98.04/57.92 | 97.85/75.46 | 95.79/2.15 | 97.15/79.16 | 95.6/95.98 | 96.75/95.47 |
| | 0.01 | 96.02/97.89 | 90.82/1.85 | 96.53/92.09 | 97.91/73.23 | 97.62/88.73 | 96.37/6.43 | 97.77/87.36 | 95.92/97.99 | 96.02/97.89 |
| | 0.05 | 94.58/95.35 | 85.18/37.1 | 94.17/94.84 | 97.46/79.32 | 97.41/84.15 | 96.66/18.69 | 97.28/84.28 | 93.99/90.75 | 95.12/93.22 |
| | 0.1 | 43.23/42.35 | 41.45/23.87 | 40.85/39.24 | 57.35/1.11 | 57.64/6.67 | 49.64/4.55 | 51.71/4.25 | 21.65/23.65 | 41.58/23.7 |
| Tiny | 0.001 | 48.59/97.82 | 50.97/34.53 | 48.06/97.39 | 53.31/47.63 | 53.36/70.99 | 45.75/5.45 | 42.33/1.35 | 48.53/97.8 | 48.59/97.82 |
| | 0.005 | 48.65/96.07 | 48.47/64.12 | 48.52/95.94 | 53.16/43.06 | 53.52/56.94 | 47.83/10.15 | 40.24/2.01 | 48.75/95.83 | 48.64/96.07 |
| | 0.01 | 39.48/94.12 | 38.43/81.5 | 39.15/93.83 | 51.63/4.68 | 51.51/6.22 | 47.23/6.5 | 45.52/2.0 | 40.24/90.64 | 38.29/84.31 |
| | 0.05 | 38.58/62.37 | 36.23/35.07 | 38.48/62.09 | 49.47/1.85 | 49.96/1.86 | 45.74/1.42 | 48.23/1.33 | 38.58/62.37 | 39.0/61.64 |
| | 0.1 | 17.02/68.42 | 17.48/57.29 | 16.16/68.67 | 32.33/2.47 | 32.79/2.67 | 31.73/4.35 | 33.66/2.9 | 16.45/70.13 | 17.06/67.26 |

Table 23: Ablation Study Assessing the Effect of Learning Rate Variability on the Performance of the K&L Method with MobileNet-v3-large on CIFAR-10, CIFAR-100, GTSRB, Tiny ImageNet - Comparison of Benign Accuracy (BA) and Attack Success Rate (ASR)

| Datasets | LR | No Defense | ANP | BNP | FP | FT | I-BAU | NAD | CLP | RNP |
|---|---|---|---|---|---|---|---|---|---|---|
| | | BA/ASR | BA/ASR | BA/ASR | BA/ASR | BA/ASR | BA/ASR | BA/ASR | BA/ASR | BA/ASR |
| CIFAR-10 | 0.001 | 83.48/99.67 | 77.88/84.86 | 81.45/96.44 | 79.59/45.44 | 79.27/53.31 | -/- | 76.29/70.64 | 81.74/99.22 | 83.48/99.67 |
| | 0.005 | 80.13/99.88 | 76.28/47.12 | 75.19/97.87 | 80.19/46.22 | 79.42/54.9 | -/- | 80.13/79.24 | 79.35/99.38 | 70.77/90.44 |
| | 0.01 | 77.2/99.97 | 70.73/27.96 | 72.07/99.94 | 79.98/49.96 | 79.93/58.43 | -/- | 81.09/81.53 | 70.18/99.92 | 55.73/25.07 |
| | 0.05 | 41.71/100.0 | 39.42/86.07 | 20.32/100.0 | 76.47/56.61 | 76.4/66.11 | -/- | 75.57/92.51 | 37.59/100.0 | 41.89/99.97 |
| | 0.1 | 44.44/99.97 | 41.02/99.94 | 44.78/99.94 | 69.93/57.97 | 70.3/67.81 | -/- | 68.14/85.97 | 39.61/99.96 | 44.44/99.97 |
| CIFAR-100 | 0.001 | 51.92/99.01 | 46.77/95.74 | 46.22/95.97 | 46.36/41.48 | 47.64/40.65 | -/- | 46.09/38.48 | 43.34/93.34 | 42.19/90.61 |
| | 0.005 | 49.25/99.09 | 46.81/96.29 | 45.81/98.34 | 46.74/54.55 | 48.35/59.85 | -/- | 48.74/73.37 | 40.02/92.44 | 34.67/71.09 |
| | 0.01 | 47.16/99.57 | 47.12/99.52 | 35.51/94.77 | 46.19/64.09 | 47.45/66.06 | -/- | 49.94/79.94 | 37.4/97.46 | 30.74/81.88 |
| | 0.05 | 26.41/99.98 | 24.02/86.13 | 22.04/99.68 | 39.76/88.46 | 40.08/90.6 | -/- | 44.46/97.45 | 23.87/99.95 | 18.64/89.53 |
| | 0.1 | 19.92/99.69 | 20.52/0.19 | 19.63/99.63 | 30.24/85.08 | 30.27/55.96 | -/- | 30.36/82.27 | 12.34/99.62 | 19.43/1.09 |
| GTSRB | 0.001 | 94.52/97.65 | 88.93/29.24 | 90.49/95.03 | 95.5/8.28 | 95.34/32.08 | -/- | 94.58/50.84 | 85.07/87.84 | 94.52/97.65 |
| | 0.005 | 93.34/99.24 | 88.71/79.17 | 89.25/98.94 | 95.86/16.8 | 95.55/56.2 | -/- | 94.62/74.69 | 88.12/96.06 | 93.34/99.24 |
| | 0.01 | 89.79/99.43 | 89.02/95.88 | 86.76/98.67 | 95.76/31.89 | 95.21/57.14 | -/- | 94.54/68.97 | 72.83/97.29 | 81.47/75.76 |
| | 0.05 | 62.68/100.0 | 64.95/99.88 | 60.29/100.0 | 95.76/77.79 | 95.26/80.76 | -/- | 94.72/92.82 | 61.57/100.0 | 71.91/18.47 |
| | 0.1 | 64.39/100.0 | 63.95/98.15 | 48.08/100.0 | 94.43/88.5 | 94.79/90.7 | -/- | 93.98/96.88 | 52.63/100.0 | 71.51/99.99 |
| Tiny | 0.001 | 46.84/98.83 | 42.81/89.16 | 44.39/97.35 | 40.77/34.81 | 43.84/51.3 | -/- | 35.13/10.69 | 46.8/98.81 | 45.7/97.86 |
| | 0.005 | 41.99/99.69 | 41.93/99.62 | 39.19/99.13 | 41.8/48.6 | 43.02/57.99 | -/- | 33.44/5.99 | 42.0/99.69 | 36.29/96.56 |
| | 0.01 | 35.45/99.95 | 35.5/99.91 | 31.65/99.87 | 41.56/66.99 | 42.44/78.3 | -/- | 35.84/22.24 | 35.45/99.95 | 31.12/27.14 |
| | 0.05 | 29.19/99.98 | 29.64/2.43 | 27.08/100.0 | 37.28/35.58 | 38.03/37.8 | -/- | 41.11/80.28 | 29.21/99.98 | 29.09/2.4 |
| | 0.1 | 17.84/99.95 | 16.28/0.0 | 15.93/99.23 | 28.77/9.42 | 29.04/8.62 | -/- | 36.85/68.96 | 17.46/99.94 | 18.81/22.97 |

**EfficientNet-B3**: The data from Table 24 indicate that for EfficientNet-B3, the learning rate has a critical role in determining BA and ASR. Lower learning rates lead to higher BA and ASR values, indicating effective learning and successful backdoor attacks, whereas higher learning rates result in poorer performance, likely due to issues like rapid convergence or overfitting.

Table 24: Ablation Study Assessing the Effect of Learning Rate Variability on the Performance of the K&L Method with EfficientNet-B3 on CIFAR-10, CIFAR-100, GTSRB, Tiny ImageNet - Comparison of Benign Accuracy (BA) and Attack Success Rate (ASR)

| Datasets | LR | No Defense | ANP | BNP | FP | FT | I-BAU | NAD | CLP | RNP |
|---|---|---|---|---|---|---|---|---|---|---|
| | | BA/ASR | BA/ASR | BA/ASR | BA/ASR | BA/ASR | BA/ASR | BA/ASR | BA/ASR | BA/ASR |
| CIFAR-10 | 0.001 | 68.23/87.07 | 68.24/87.06 | 49.45/52.41 | 64.4/31.52 | 63.81/34.61 | 66.51/63.62 | 65.8/45.14 | 18.32/10.63 | 67.68/85.9 |
| | 0.005 | 62.38/95.38 | 62.38/95.37 | 15.31/36.37 | 64.29/31.64 | 64.33/34.48 | 65.08/81.63 | 64.97/58.27 | 14.41/40.2 | 16.24/54.24 |
| | 0.01 | 65.38/98.94 | 65.37/98.94 | 40.18/82.3 | 66.57/45.78 | 65.7/45.67 | 67.16/90.64 | 67.45/72.11 | 17.4/5.93 | 21.67/3.19 |
| | 0.05 | 60.9/99.2 | 60.0/98.9 | 42.18/89.56 | 65.61/78.5 | 65.86/80.29 | 64.06/98.34 | 65.04/96.04 | 9.7/0.0 | 21.18/16.09 |
| | 0.1 | 38.38/39.21 | 39.2/33.49 | 28.95/35.46 | 47.83/14.06 | 47.68/15.5 | 44.83/17.98 | 45.67/16.59 | 11.24/1.93 | 38.38/39.2 |
| CIFAR-100 | 0.001 | 49.38/94.21 | 49.36/94.2 | 41.19/91.96 | 39.45/12.18 | 41.71/17.6 | 44.69/51.71 | 42.27/28.75 | 19.97/7.88 | 41.27/55.41 |
| | 0.005 | 46.0/97.8 | 45.98/97.81 | 41.43/95.47 | 41.24/17.84 | 41.78/17.77 | 44.42/47.25 | 44.14/34.1 | 12.45/13.65 | 26.4/53.9 |
| | 0.01 | 44.76/99.66 | 44.76/99.66 | 40.6/99.49 | 41.42/23.12 | 42.13/18.17 | 45.4/65.55 | 45.15/54.26 | 11.78/15.63 | 32.7/54.08 |
| | 0.05 | 21.54/99.81 | 21.83/99.74 | 8.99/93.47 | 27.23/24.31 | 26.62/21.06 | 26.94/87.24 | 27.1/77.14 | 3.6/0.09 | 10.61/0.86 |
| | 0.1 | 16.67/99.36 | 16.31/58.84 | 15.14/98.6 | 25.6/23.85 | 25.09/14.55 | 24.57/84.31 | 24.81/77.77 | 4.89/5.77 | 8.5/17.35 |
| GTSRB | 0.001 | 83.21/94.96 | 76.5/80.14 | 79.33/91.11 | 86.48/12.99 | 84.64/31.34 | 83.46/29.52 | 84.19/52.27 | 31.2/24.95 | 77.65/82.26 |
| | 0.005 | 83.26/96.31 | 83.4/95.04 | 65.19/29.24 | 86.85/25.47 | 84.8/55.82 | 84.05/52.53 | 83.63/75.75 | 28.69/53.22 | 71.86/88.65 |
| | 0.01 | 83.22/95.54 | 76.67/93.9 | 78.57/92.54 | 87.3/41.44 | 86.22/67.29 | 82.55/15.82 | 84.6/79.86 | 24.73/26.2 | 75.72/81.02 |
| | 0.05 | 40.8/59.48 | 41.37/45.68 | 39.56/59.64 | 66.4/0.0 | 66.84/0.14 | 49.51/0.43 | 55.27/0.3 | 13.56/41.56 | 38.5/48.2 |
| | 0.1 | 52.11/79.15 | 50.36/74.69 | 49.83/77.75 | 69.34/42.51 | 70.38/44.48 | 60.19/50.72 | 62.98/54.17 | 43.49/66.69 | 40.76/74.52 |
| Tiny | 0.001 | 44.34/99.43 | 44.8/99.22 | 44.43/99.34 | 42.87/13.85 | 45.61/81.08 | 42.25/91.85 | 35.72/6.75 | 43.81/98.99 | 44.35/99.43 |
| | 0.005 | 43.96/99.78 | 43.97/99.78 | 43.35/99.83 | 43.39/22.31 | 45.24/80.82 | 43.65/93.12 | 34.34/8.41 | 43.84/99.8 | 43.97/99.78 |
| | 0.01 | 39.54/99.49 | 39.54/99.49 | 38.75/99.33 | 44.13/36.25 | 44.66/71.62 | 42.06/90.05 | 37.23/29.6 | 39.18/99.28 | 27.22/95.26 |
| | 0.05 | 33.01/99.99 | 30.0/0.46 | 31.09/100.0 | 39.83/31.44 | 39.62/38.92 | 43.87/76.82 | 41.53/61.8 | 32.47/99.98 | 31.97/22.57 |
| | 0.1 | 23.29/99.99 | 21.23/0.0 | 23.93/99.99 | 23.96/7.12 | 23.39/6.07 | 33.4/97.24 | 33.23/48.34 | 21.19/99.99 | 23.36/2.84 |

## G.5 Ablation study on Attack Step Size $\alpha$

**PreActResNet18**: As demonstrated in Table 25, the variation in the attack step size $\alpha$ has a profound effect on the BA and ASR for the PreActResNet18 model across different datasets. Notably, a smaller $\alpha$ tends to yield a higher BA but a lower ASR, indicating a more conservative attack strategy. In contrast, increasing $\alpha$ leads to a higher ASR but at the potential cost of reduced BA, suggesting a more aggressive approach that may compromise model accuracy.

Table 25: Ablation Study Assessing the Effect of $\alpha$ Variability on the Performance of the K&L Method with PreActResNet18 on CIFAR-100, GTSRB, Tiny ImageNet - Comparison of Benign Accuracy (BA) and Attack Success Rate (ASR)

| Datasets | $\alpha$ | No Defense | ANP | BNP | FP | FT | I-BAU | NAD | CLP | RNP |
|---|---|---|---|---|---|---|---|---|---|---|
| | | BA/ASR | BA/ASR | BA/ASR | BA/ASR | BA/ASR | BA/ASR | BA/ASR | BA/ASR | BA/ASR |
| CIFAR-100 | 0.25 | 61.23/82.39 | 57.16/0.39 | 59.97/4.61 | 66.41/1.92 | 67.97/3.79 | 64.53/0.88 | 67.79/4.73 | 52.88/0.79 | 48.97/25.41 |
| | 0.5 | 59.85/99.59 | 57.16/8.46 | 59.97/44.83 | 66.41/17.9 | 67.97/35.74 | 64.53/5.78 | 67.79/40.61 | 52.88/7.39 | 50.18/19.07 |
| | 0.75 | 61.34/99.73 | 57.16/28.14 | 59.97/77.79 | 66.41/41.82 | 67.97/70.12 | 64.53/12.96 | 67.79/74.21 | 52.88/27.73 | 58.14/72.21 |
| | 1 | 61.33/99.95 | 57.15/91.54 | 59.97/99.67 | 66.41/85.06 | 67.97/98.72 | 64.53/53.0 | 67.78/99.11 | 52.88/88.16 | 55.6/97.81 |
| | 1.25 | 62.02/99.96 | 57.16/57.54 | 59.97/93.08 | 66.41/70.51 | 67.97/91.03 | 64.53/25.15 | 67.79/91.93 | 52.88/56.82 | 55.05/64.97 |
| | 1.5 | 61.49/99.97 | 57.16/67.28 | 59.97/96.68 | 66.41/78.75 | 67.97/95.15 | 64.53/31.63 | 67.79/96.07 | 52.88/71.42 | 55.48/84.97 |
| GTSRB | 0.25 | 94.11/72.72 | 90.54/0.11 | 97.77/6.25 | 98.24/0.13 | 98.41/2.58 | 89.92/0.28 | 97.85/0.48 | 97.93/23.48 | 27.17/0.18 |
| | 0.5 | 97.23/90.17 | 90.54/4.59 | 97.77/46.78 | 98.24/3.02 | 98.41/27.7 | 89.92/1.58 | 97.85/7.63 | 97.93/74.03 | 94.81/56.71 |
| | 0.75 | 98.0/97.04 | 90.54/15.61 | 97.77/76.45 | 98.24/14.13 | 98.41/51.6 | 89.92/5.27 | 97.85/25.72 | 97.93/92.47 | 97.7/91.04 |
| | 1 | 97.66/99.85 | 89.78/2.03 | 97.39/88.93 | 98.27/44.36 | 98.16/88.22 | 92.03/39.96 | 98.08/92.87 | 97.57/99.53 | 97.66/99.85 |
| | 1.25 | 98.33/98.59 | 90.54/35.26 | 97.77/92.51 | 98.24/37.88 | 98.41/78.94 | 89.92/17.31 | 97.85/54.98 | 97.93/99.16 | 98.33/98.58 |
| | 1.5 | 98.37/99.44 | 90.54/41.96 | 97.77/96.13 | 98.24/48.98 | 98.41/86.07 | 89.92/24.22 | 97.85/65.4 | 97.93/99.9 | 98.37/99.44 |
| Tiny | 0.25 | 42.36/79.11 | 48.09/5.42 | 48.09/5.42 | 52.27/0.49 | 54.63/0.81 | 50.6/0.26 | 47.77/1.38 | 47.72/4.9 | 43.3/64.57 |
| | 0.5 | 45.85/98.62 | 48.09/48.13 | 48.09/48.13 | 52.27/5.64 | 54.63/18.46 | 50.6/2.28 | 47.77/5.93 | 47.72/45.42 | 41.87/85.17 |
| | 0.75 | 44.97/99.86 | 48.09/68.88 | 48.09/68.88 | 52.27/9.15 | 54.63/31.28 | 50.6/4.08 | 47.77/9.91 | 47.72/65.79 | 45.77/99.68 |
| | 1 | 48.09/99.91 | 48.09/99.91 | 48.09/99.91 | 52.26/62.62 | 54.63/97.66 | 50.6/51.52 | 47.77/44.37 | 47.71/99.91 | 47.87/99.85 |
| | 1.25 | 46.74/99.99 | 48.09/90.25 | 48.09/90.25 | 52.27/29.45 | 54.63/70.94 | 50.6/15.86 | 47.77/31.29 | 47.72/89.16 | 46.87/99.99 |
| | 1.5 | 47.11/100.0 | 48.09/94.75 | 48.09/94.75 | 52.27/41.52 | 54.63/81.22 | 50.6/23.67 | 47.77/44.19 | 47.72/94.07 | 47.4/100.0 |

**VGG19-BN**: Table 26 reveals that for the VGG19-BN model, varying $\alpha$ significantly impacts both BA and ASR across all datasets. Lower values of $\alpha$ correlate with higher BA and lower ASR, indicative of less effective but safer attacks. Conversely, as $\alpha$ increases, ASR improves substantially, though this is sometimes at the expense of BA, reflecting a trade-off between attack effectiveness and model accuracy.

Table 26: Ablation Study Assessing the Effect of $\alpha$ Variability on the Performance of the K&L Method with VGG19-BN on CIFAR-10, CIFAR-100, GTSRB, Tiny ImageNet - Comparison of Benign Accuracy (BA) and Attack Success Rate (ASR)

| Datasets | $\alpha$ | No Defense | ANP | BNP | FP | FT | I-BAU | NAD | CLP | RNP |
|---|---|---|---|---|---|---|---|---|---|---|
| | | BA/ASR | BA/ASR | BA/ASR | BA/ASR | BA/ASR | BA/ASR | BA/ASR | BA/ASR | BA/ASR |
| CIFAR-10 | 0.25 | 87.25/10.49 | 80.36/0.86 | 86.7/12.41 | 90.09/2.72 | 90.23/3.03 | 87.99/2.21 | 89.61/3.28 | 75.45/2.19 | 87.25/10.49 |
| | 0.5 | 86.15/61.76 | 83.07/19.31 | 85.13/64.92 | 90.09/10.63 | 90.23/13.16 | 87.99/10.34 | 89.61/13.94 | 75.45/4.81 | 86.15/61.76 |
| | 0.75 | 85.82/87.16 | 77.67/9.99 | 85.87/84.29 | 90.09/19.59 | 90.23/21.76 | 87.99/22.1 | 89.61/23.16 | 75.45/6.43 | 85.82/87.16 |
| | 1 | 86.39/98.97 | 82.4/39.27 | 86.14/98.21 | 90.08/66.61 | 90.23/69.38 | 87.99/72.82 | 89.6/74.74 | 75.45/27.2 | 73.51/43.71 |
| | 1.25 | 87.52/99.02 | 81.26/20.64 | 86.92/95.43 | 90.09/53.24 | 90.23/52.36 | 87.99/61.03 | 89.61/56.87 | 75.45/17.44 | 83.93/87.53 |
| | 1.5 | 88.46/98.78 | 80.47/24.0 | 87.91/76.7 | 90.09/55.39 | 90.23/54.19 | 87.99/58.68 | 89.61/57.0 | 75.45/25.2 | 85.9/76.99 |
| CIFAR-100 | 0.25 | 56.14/27.18 | 53.26/3.72 | 56.1/22.97 | 62.59/0.39 | 62.04/0.39 | 58.34/0.31 | 61.6/0.31 | 36.81/2.33 | 51.83/9.31 |
| | 0.5 | 56.4/45.82 | 55.86/16.02 | 55.49/44.48 | 62.59/1.18 | 62.04/1.14 | 58.34/0.64 | 61.6/0.95 | 36.81/2.96 | 53.84/31.74 |
| | 0.75 | 54.19/68.03 | 52.47/35.6 | 51.55/55.66 | 62.59/6.8 | 62.04/6.27 | 58.34/2.49 | 61.6/5.63 | 36.81/3.68 | 45.24/20.37 |
| | 1 | 53.6/86.86 | 49.84/33.42 | 53.81/85.53 | 62.6/26.41 | 62.04/27.18 | 58.34/10.01 | 61.59/25.97 | 36.81/10.31 | 48.3/32.25 |
| | 1.25 | 55.41/97.91 | 55.44/84.25 | 55.4/97.63 | 62.59/15.43 | 62.04/17.33 | 58.34/4.42 | 61.6/16.42 | 36.81/6.3 | 50.04/34.8 |
| | 1.5 | 54.96/99.49 | 51.01/4.38 | 54.2/99.14 | 62.59/21.35 | 62.04/26.72 | 58.34/10.37 | 61.6/25.92 | 36.81/9.37 | 42.33/1.64 |
| GTSRB | 0.25 | 92.98/47.89 | 83.8/0.98 | 92.54/46.48 | 97.91/2.39 | 97.62/6.76 | 96.37/1.58 | 97.77/10.31 | 95.92/17.42 | 92.98/47.89 |
| | 0.5 | 95.18/79.1 | 87.54/5.28 | 94.77/73.63 | 97.91/24.67 | 97.62/44.55 | 96.37/4.65 | 97.77/49.27 | 95.92/63.28 | 95.18/79.1 |
| | 0.75 | 91.56/93.27 | 82.54/0.02 | 91.44/88.37 | 97.91/39.52 | 97.62/63.99 | 96.37/6.17 | 97.77/66.87 | 95.92/80.72 | 91.84/57.82 |
| | 1 | 96.02/97.89 | 90.82/1.85 | 96.53/92.09 | 97.91/73.23 | 97.62/88.73 | 96.37/6.43 | 97.77/87.36 | 95.92/97.99 | 96.02/97.89 |
| | 1.25 | 94.33/97.86 | 91.36/6.43 | 94.2/93.92 | 97.91/67.65 | 97.62/86.21 | 96.37/13.57 | 97.77/84.71 | 95.92/95.13 | 94.34/97.86 |
| | 1.5 | 94.91/99.55 | 88.99/0.64 | 93.82/98.54 | 97.91/73.72 | 97.62/91.16 | 96.37/11.08 | 97.77/89.2 | 95.92/98.16 | 94.91/99.55 |
| Tiny | 0.25 | 43.13/50.44 | 43.35/12.66 | 43.17/48.02 | 51.63/0.24 | 51.51/0.29 | 47.23/0.48 | 45.5/0.46 | 40.24/16.71 | 43.9/41.35 |
| | 0.5 | 42.37/84.21 | 41.35/45.4 | 42.42/82.31 | 51.63/0.53 | 51.51/0.73 | 47.23/0.95 | 45.5/0.73 | 40.24/22.11 | 42.48/62.21 |
| | 0.75 | 39.91/90.85 | 39.75/72.89 | 39.91/90.88 | 51.63/0.68 | 51.51/0.76 | 47.23/1.05 | 45.5/0.79 | 40.24/24.73 | 40.84/81.36 |
| | 1 | 39.48/94.12 | 38.43/81.5 | 39.15/93.83 | 51.63/4.68 | 51.51/6.22 | 47.23/6.5 | 45.52/2.0 | 40.24/90.64 | 38.29/84.31 |
| | 1.25 | 41.58/96.31 | 38.99/86.42 | 41.52/96.2 | 51.63/2.71 | 51.51/3.22 | 47.23/4.01 | 45.5/1.93 | 40.24/38.75 | 42.36/90.38 |
| | 1.5 | 43.47/95.01 | 41.59/80.9 | 42.78/93.68 | 51.63/3.34 | 51.51/4.03 | 47.23/5.34 | 45.5/2.83 | 40.24/43.34 | 42.74/90.63 |

**MobileNet-v3-large**: Analyzing Table 27, it is evident that for the MobileNet-v3-large model, $\alpha$ plays a crucial role in dictating BA and ASR. A smaller $\alpha$ is associated with higher BA and lower ASR, suggesting more accurate but less potent attacks. As $\alpha$ increases, there is a notable rise in ASR, but this may sometimes lead to a decrease in BA, highlighting the balance between aggressive attack strategies and maintaining model performance.

Table 27: Ablation Study Assessing the Effect of $\alpha$ Variability on the Performance of the K&L Method with MobileNet-v3-large on CIFAR-10, CIFAR-100, GTSRB, Tiny ImageNet - Comparison of Benign Accuracy (BA) and Attack Success Rate (ASR)

| Datasets | $\alpha$ | No Defense | ANP | BNP | FP | FT | I-BAU | NAD | CLP | RNP |
|---|---|---|---|---|---|---|---|---|---|---|
| | | BA/ASR | BA/ASR | BA/ASR | BA/ASR | BA/ASR | BA/ASR | BA/ASR | BA/ASR | BA/ASR |
| CIFAR-10 | 0.25 | 75.85/60.91 | 71.75/21.91 | 70.66/56.68 | 79.96/4.03 | 79.92/5.12 | -/- | 81.1/6.36 | 70.18/16.0 | 61.95/34.13 |
| | 0.5 | 75.97/99.36 | 69.79/28.71 | 69.82/93.64 | 79.96/10.68 | 79.92/15.37 | -/- | 81.1/25.11 | 70.18/57.14 | 62.39/34.59 |
| | 0.75 | 76.8/99.92 | 69.85/25.44 | 69.32/98.92 | 79.96/24.17 | 79.92/33.63 | -/- | 81.1/55.23 | 70.18/90.13 | 64.1/15.6 |
| | 1 | 77.2/99.97 | 70.73/27.96 | 72.07/99.94 | 79.98/49.96 | 79.93/58.43 | -/- | 81.09/81.53 | 70.18/99.92 | 55.73/25.07 |
| | 1.25 | 77.24/99.98 | 73.08/96.54 | 72.98/99.76 | 79.96/56.2 | 79.92/64.79 | -/- | 81.1/81.01 | 70.18/99.33 | 64.51/80.36 |
| | 1.5 | 77.54/100.0 | 72.96/98.74 | 73.64/100.0 | 79.96/70.71 | 79.92/77.82 | -/- | 81.1/91.16 | 70.18/99.76 | 72.2/88.38 |
| CIFAR-100 | 0.25 | 45.61/46.23 | 42.83/32.63 | 43.82/39.77 | 46.21/1.77 | 47.47/1.85 | -/- | 49.94/1.77 | 37.4/10.88 | 38.67/25.65 |
| | 0.5 | 44.02/93.68 | 42.18/89.59 | 37.47/80.02 | 46.21/13.19 | 47.47/15.12 | -/- | 49.94/19.3 | 37.4/43.71 | 36.72/72.01 |
| | 0.75 | 44.76/98.96 | 44.96/98.66 | 27.49/90.67 | 46.21/41.95 | 47.47/45.53 | -/- | 49.94/57.83 | 37.4/86.02 | 38.68/97.05 |
| | 1 | 47.16/99.57 | 47.12/99.52 | 35.51/94.77 | 46.19/64.09 | 47.45/66.06 | -/- | 49.94/79.94 | 37.4/97.46 | 30.74/81.88 |
| | 1.25 | 46.54/99.59 | 46.51/99.39 | 32.86/98.37 | 46.21/73.6 | 47.47/74.2 | -/- | 49.94/85.35 | 37.4/97.24 | 20.8/49.37 |
| | 1.5 | 46.53/99.88 | 44.64/99.33 | 33.99/98.53 | 46.21/83.31 | 47.47/83.45 | -/- | 49.94/92.62 | 37.4/98.82 | 34.67/72.71 |
| GTSRB | 0.25 | 88.31/63.29 | 88.8/53.54 | 83.49/59.55 | 95.76/0.66 | 95.21/1.27 | -/- | 94.54/2.15 | 72.84/10.39 | 89.37/42.87 |
| | 0.5 | 79.52/96.75 | 87.34/83.12 | 67.35/92.46 | 95.76/4.15 | 95.21/11.65 | -/- | 94.54/16.57 | 72.84/30.43 | 80.59/72.37 |
| | 0.75 | 89.77/99.68 | 88.0/92.16 | 86.11/97.64 | 95.76/12.15 | 95.21/30.68 | -/- | 94.54/40.94 | 72.84/62.49 | 68.09/20.71 |
| | 1 | 89.79/99.43 | 89.02/95.88 | 86.76/98.67 | 95.76/31.89 | 95.21/57.14 | -/- | 94.54/68.97 | 72.83/97.29 | 81.47/75.76 |
| | 1.25 | 92.42/99.62 | 90.02/93.6 | 89.08/99.41 | 95.76/34.24 | 95.21/55.46 | -/- | 94.54/64.25 | 72.84/86.25 | 92.42/99.62 |
| | 1.5 | 90.89/99.79 | 89.98/96.95 | 87.17/99.55 | 95.76/46.3 | 95.21/66.59 | -/- | 94.54/74.46 | 72.84/88.18 | 81.14/87.55 |
| Tiny | 0.25 | 37.36/62.68 | 36.27/51.05 | 37.98/49.65 | 41.57/0.65 | 42.43/0.81 | -/- | 35.86/0.34 | 35.47/1.97 | 36.64/55.71 |
| | 0.5 | 36.83/92.59 | 34.64/69.07 | 35.8/80.38 | 41.57/1.77 | 42.43/2.16 | -/- | 35.86/0.6 | 35.47/11.3 | 32.13/56.42 |
| | 0.75 | 35.14/99.22 | 32.73/95.55 | 32.86/98.65 | 41.57/24.3 | 42.43/33.56 | -/- | 35.86/4.72 | 35.47/93.31 | 30.23/74.48 |
| | 1 | 35.45/99.95 | 35.5/99.91 | 31.65/99.87 | 41.56/66.99 | 42.44/78.3 | -/- | 35.84/22.24 | 35.45/99.95 | 31.12/27.14 |
| | 1.25 | 36.88/99.88 | 34.7/99.57 | 33.75/59.8 | 41.57/79.95 | 42.43/87.01 | -/- | 35.86/43.34 | 35.47/99.88 | 28.06/53.07 |
| | 1.5 | 39.82/99.98 | 36.55/99.83 | 37.41/82.21 | 41.57/90.48 | 42.43/94.59 | -/- | 35.86/62.69 | 35.47/99.98 | 38.42/88.73 |

**EfficientNet-B3**: The data from Table 28 indicate that for EfficientNet-B3, the variation in $\alpha$ substantially affects BA and ASR across different datasets. Smaller $\alpha$ values generally result in higher BA but lower ASR, suggesting more cautious attack strategies that preserve model accuracy. Increasing $\alpha$, however, leads to higher ASR, indicating more effective but potentially riskier attacks in terms of compromising model accuracy.

Table 28: Ablation Study Assessing the Effect of $\alpha$ Variability on the Performance of the K&L Method with EfficientNet-B3 on CIFAR-10, CIFAR-100, GTSRB, Tiny ImageNet - Comparison of Benign Accuracy (BA) and Attack Success Rate (ASR)

| Datasets | $\alpha$ | No Defense | ANP | BNP | FP | FT | I-BAU | NAD | CLP | RNP |
|---|---|---|---|---|---|---|---|---|---|---|
| | | BA/ASR | BA/ASR | BA/ASR | BA/ASR | BA/ASR | BA/ASR | BA/ASR | BA/ASR | BA/ASR |
| CIFAR-10 | 0.25 | 66.29/18.43 | 65.38/18.72 | 45.3/10.06 | 66.57/6.89 | 65.72/8.72 | 67.16/10.63 | 67.45/9.34 | 17.39/5.14 | 60.41/20.09 |
| | 0.5 | 65.03/79.8 | 65.02/79.79 | 38.46/53.3 | 66.57/19.07 | 65.72/20.22 | 67.16/48.02 | 67.45/30.01 | 17.39/5.46 | 17.09/6.03 |
| | 0.75 | 63.99/95.99 | 64.0/95.99 | 35.85/76.68 | 66.57/27.67 | 65.72/28.51 | 67.16/73.58 | 67.45/47.23 | 17.39/5.59 | 24.16/13.7 |
| | 1 | 65.38/98.94 | 65.37/98.94 | 40.18/82.3 | 66.57/45.78 | 65.7/45.67 | 67.16/90.64 | 67.45/72.11 | 17.4/5.93 | 21.67/3.19 |
| | 1.25 | 66.65/98.77 | 66.65/98.77 | 14.82/19.61 | 66.57/60.52 | 65.72/62.01 | 67.16/94.24 | 67.45/83.73 | 17.39/6.31 | 26.68/3.04 |
| | 1.5 | 64.83/99.44 | 59.21/99.04 | 45.41/88.0 | 66.57/70.99 | 65.72/72.81 | 67.16/95.24 | 67.45/89.58 | 17.39/6.64 | 22.5/1.06 |
| CIFAR-100 | 0.25 | 41.26/29.17 | 44.76/15.21 | 30.95/33.66 | 41.41/1.32 | 42.16/1.15 | 45.42/1.87 | 45.14/2.26 | 11.79/11.25 | 29.99/17.32 |
| | 0.5 | 42.69/76.53 | 42.67/76.53 | 34.11/55.64 | 41.41/3.1 | 42.16/2.47 | 45.42/8.33 | 45.14/5.93 | 11.79/11.84 | 17.77/15.0 |
| | 0.75 | 42.13/97.39 | 42.12/97.39 | 39.7/96.98 | 41.41/8.68 | 42.16/7.21 | 45.42/28.36 | 45.14/22.92 | 11.79/13.24 | 31.23/22.9 |
| | 1 | 44.76/99.66 | 44.76/99.66 | 40.6/99.49 | 41.42/23.12 | 42.13/18.17 | 45.4/65.55 | 45.15/54.26 | 11.78/15.63 | 32.7/54.08 |
| | 1.25 | 43.53/98.96 | 39.24/98.59 | 42.42/98.95 | 41.41/23.9 | 42.16/19.14 | 45.42/49.46 | 45.14/47.42 | 11.79/17.12 | 21.43/37.35 |
| | 1.5 | 43.61/99.79 | 43.63/99.79 | 39.77/99.62 | 41.41/10.9 | 42.16/8.64 | 45.42/28.06 | 45.14/29.27 | 11.79/13.83 | 16.25/7.38 |
| GTSRB | 0.25 | 79.6/50.29 | 76.11/36.84 | 72.09/36.84 | 87.29/0.64 | 86.22/4.43 | 82.56/1.66 | 84.59/11.48 | 24.72/10.97 | 79.6/50.28 |
| | 0.5 | 80.37/83.43 | 74.77/72.37 | 54.57/29.5 | 87.29/10.18 | 86.22/29.84 | 82.56/4.55 | 84.59/44.59 | 24.72/13.83 | 77.35/67.25 |
| | 0.75 | 81.54/93.55 | 79.18/86.76 | 76.61/88.35 | 87.29/32.45 | 86.22/54.05 | 82.56/10.25 | 84.59/68.15 | 24.72/21.05 | 51.01/60.32 |
| | 1 | 83.22/95.54 | 76.67/93.9 | 78.57/92.54 | 87.3/41.44 | 86.22/67.29 | 82.55/15.82 | 84.6/79.86 | 24.73/26.2 | 75.72/81.02 |
| | 1.25 | 83.85/97.55 | 80.78/96.34 | 78.84/96.02 | 87.29/53.17 | 86.22/76.52 | 82.56/19.98 | 84.59/86.07 | 24.72/36.26 | 78.93/81.11 |
| | 1.5 | 84.73/98.92 | 83.36/98.48 | 82.16/95.74 | 87.29/58.01 | 86.22/83.23 | 82.56/19.74 | 84.59/91.7 | 24.72/49.0 | 72.09/89.81 |
| Tiny | 0.25 | 37.9/52.28 | 35.32/39.2 | 37.16/49.43 | 44.14/0.68 | 44.68/1.3 | 42.07/1.56 | 37.23/1.62 | 39.14/8.98 | 31.08/25.1 |
| | 0.5 | 38.15/88.72 | 38.61/88.09 | 36.53/85.72 | 44.14/2.55 | 44.68/6.71 | 42.07/9.29 | 37.23/4.29 | 39.14/39.23 | 23.76/48.92 |
| | 0.75 | 39.57/97.53 | 36.58/92.33 | 39.59/97.47 | 44.14/9.53 | 44.68/26.62 | 42.07/32.42 | 37.23/10.8 | 39.14/78.72 | 20.26/52.6 |
| | 1 | 39.54/99.49 | 39.54/99.49 | 38.75/99.33 | 44.13/36.25 | 44.66/71.62 | 42.06/90.05 | 37.23/29.6 | 39.18/99.28 | 27.22/95.26 |
| | 1.25 | 40.05/99.91 | 40.18/99.83 | 39.83/99.82 | 44.14/31.9 | 44.68/61.07 | 42.07/64.37 | 37.23/26.5 | 39.14/96.25 | 32.83/94.79 |
| | 1.5 | 40.75/100.0 | 39.19/99.9 | 40.87/99.97 | 44.14/44.96 | 44.68/74.96 | 42.07/74.84 | 37.23/38.17 | 39.14/98.41 | 22.95/97.04 |

## G.6 IMPACT OF EPOCHS AND LEARNING RATE ON AAC AND AAV

Table 29: AAV of different epochs on PreActResNet18, VGG19-BN, MobileNet-v3-large, and EfficientNet-B3

| Models | Epochs | 2 | 4 | 6 | 8 | 10 | 12 |
|---|---|---|---|---|---|---|---|
| PreActResNet18 | AAC1 | 0.81125 | 0.823472 | 0.831667 | 0.840278 | 0.834028 | 0.78375 |
| | AAC3 | 0.795139 | 0.809306 | 0.797639 | 0.79375 | 0.774583 | 0.751806 |
| | AAC5 | 0.774028 | 0.79625 | 0.781528 | 0.775972 | 0.749583 | 0.73375 |
| VGG19-BN | AAC1 | 0.455278 | 0.648333 | 0.717222 | 0.655694 | 0.629861 | 0.501944 |
| | AAC3 | 0.425833 | 0.629722 | 0.684167 | 0.595972 | 0.54 | 0.455278 |
| | AAC5 | 0.418056 | 0.585278 | 0.616944 | 0.547639 | 0.503056 | 0.444444 |
| MobileNet-v3-large | AAC1 | 0.729375 | 0.82375 | 0.825625 | 0.781563 | 0.71875 | 0.712812 |
| | AAC3 | 0.728125 | 0.820938 | 0.814375 | 0.762813 | 0.692813 | 0.6975 |
| | AAC5 | 0.725938 | 0.774062 | 0.808125 | 0.7375 | 0.68375 | 0.653437 |
| EfficientNet-B3 | AAC1 | 0.670972 | 0.780417 | 0.785278 | 0.763611 | 0.698889 | 0.616944 |
| | AAC3 | 0.663611 | 0.770556 | 0.772222 | 0.718056 | 0.651667 | 0.585278 |
| | AAC5 | 0.647222 | 0.768611 | 0.751944 | 0.7 | 0.619167 | 0.555278 |

**Impact of Epochs on AAC and AAV**

In Table 29, we present the AAV of different attack methods on PreActResNet18, VGG19-BN, MobileNet-v3-large, and EfficientNet-B3 across varying epochs. The table highlights three levels of AAC (AAC1, AAC3, and AAC5), representing the permissible loss in accuracy of 1%, 3%, and 5%, respectively.

For the PreActResNet18 model, as shown in Figure 7, the AAV generally increases with the number of epochs, peaking at 8 epochs for AAC1 and declining slightly thereafter. A similar trend is observed in AAC3 and AAC5 levels, though the peak occurs at lower epochs and the decline is more significant beyond 8 epochs.

In contrast, as shown in Figure 8, the VGG19-BN model demonstrates a peak in AAV at 6 epochs for AAC1, followed by a noticeable decrease. This trend is consistent across all AAC levels, indicating that extending training beyond 6 epochs may not be beneficial for this model in terms of AAV.

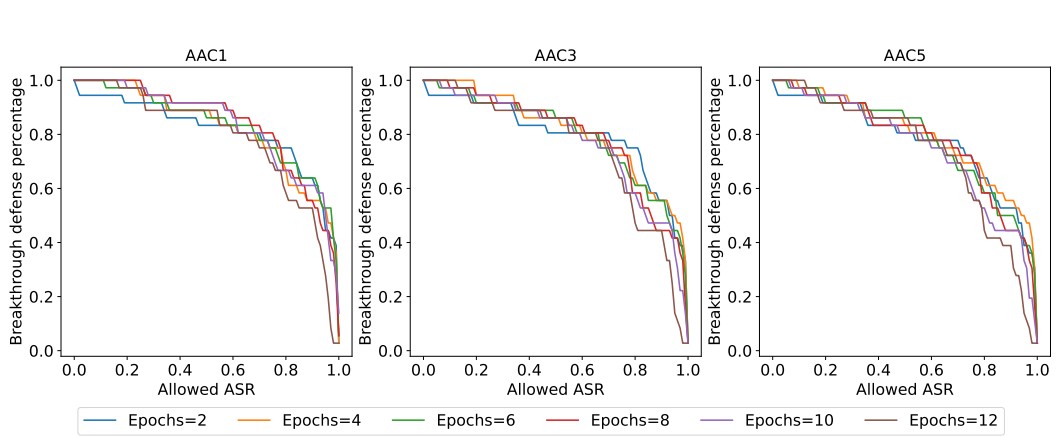

Figure 7: Impact of Epochs on AAC of PreActResNet18

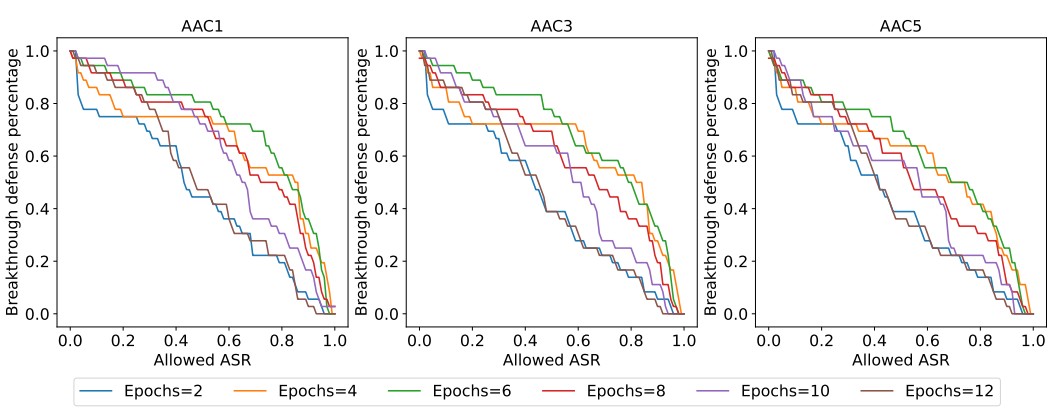

Figure 8: Impact of Epochs on AAC of VGG19-BN

As shown in Figure 9, MobileNet-v3-large shows a relatively stable AAV across epochs, with a slight peak at 4 and 6 epochs for AAC1 and AAC3. However, a gradual decrease is observed in AAC5, suggesting a diminishing return on AAV with increased epochs.

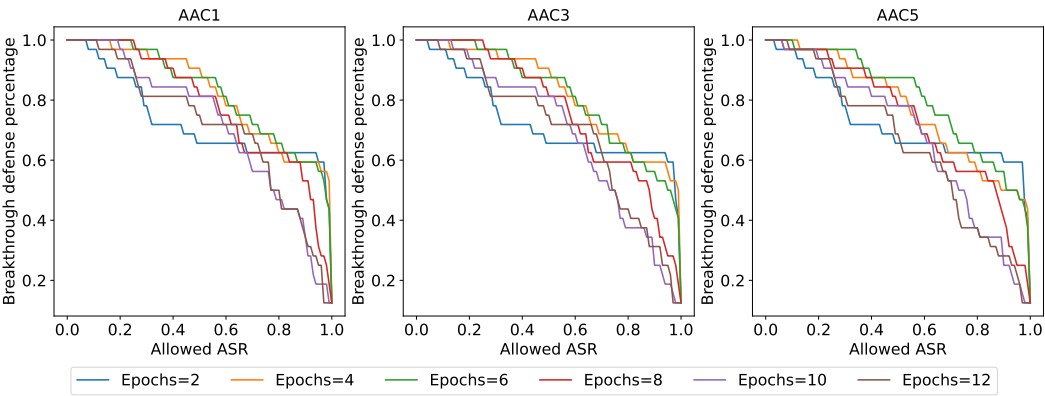

Figure 9: Impact of Epochs on AAC of MobileNet-v3-large

Lastly, as shown in Figure 10, EfficientNet-B3 reveals a peak in AAV at 4 epochs for AAC1 and AAC3, followed by a gradual decrease. The trend is similar for AAC5, with a noticeable decline in AAV after 4 epochs.

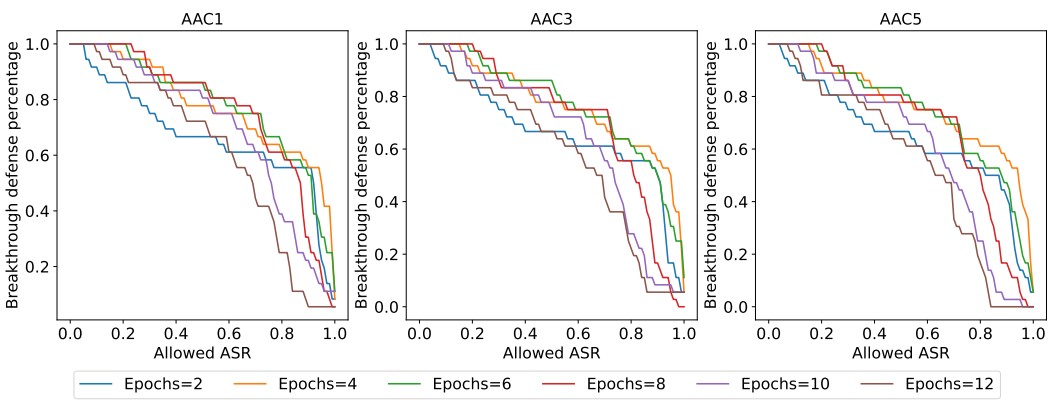

Figure 10: Impact of Epochs on AAC of EfficientNet-B3

**Impact of Learning Rate on AAC and AAV**

Table 30 illustrates the impact of different learning rates (LR) on the AAV for various models, namely PreActResNet18, VGG19-BN, MobileNet-v3-large, and EfficientNet-B3. The table showcases AAV at three different AAC levels (AAC1, AAC3, and AAC5), where AAC1, AAC3, and AAC5 correspond to allowable accuracy losses of 1%, 3%, and 5% respectively.

In the PreActResNet18 model, as shown in Figure 11, the AAV is observed to be highest at an LR of 0.005 across all AAC levels, indicating that this learning rate is optimal for the effectiveness of the attack. However, there is a noticeable decrease in AAV as the learning rate increases to 0.1.

For the VGG19-BN model, as shown in Figure 12, the AAV peaks at an LR of 0.01 for AAC1 and AAC3 levels, while AAC5 shows a similar peak at an LR of 0.005. This suggests a slightly different optimal learning rate for attacks with a higher tolerance for accuracy loss.

As shown in Figure 13, the MobileNet-v3-large model demonstrates a consistent pattern where the AAV decreases as the learning rate increases. The highest AAV is achieved at an LR of 0.001 for AAC1 and AAC3, while AAC5 has its peak at an LR of 0.005.

Table 30: AAV of different learning rates on PreActResNet18, VGG19-BN, MobileNet-v3-large, and EfficientNet-B3

| Models | LR | 0.001 | 0.005 | 0.01 | 0.05 | 0.1 |
|---|---|---|---|---|---|---|
| PreActResNet18 | AAC1 | 0.917917 | 0.950417 | 0.812778 | 0.737917 | 0.708889 |
| | AAC3 | 0.87875 | 0.8875 | 0.790833 | 0.679028 | 0.667222 |
| | AAC5 | 0.830417 | 0.854583 | 0.783333 | 0.651389 | 0.666667 |
| VGG19-BN | AAC1 | 0.537222 | 0.645139 | 0.648333 | 0.566944 | 0.346944 |
| | AAC3 | 0.521944 | 0.632083 | 0.629722 | 0.539167 | 0.3425 |
| | AAC5 | 0.502222 | 0.581389 | 0.585278 | 0.509722 | 0.338889 |
| MobileNet-v3-large | AAC1 | 0.896719 | 0.839531 | 0.82375 | 0.801719 | 0.724531 |
| | AAC3 | 0.827656 | 0.827344 | 0.820938 | 0.797344 | 0.724531 |
| | AAC5 | 0.763281 | 0.79125 | 0.774062 | 0.786563 | 0.717344 |
| EfficientNet-B3 | AAC1 | 0.808333 | 0.777083 | 0.780417 | 0.703056 | 0.525694 |
| | AAC3 | 0.746111 | 0.757222 | 0.770556 | 0.673889 | 0.515972 |
| | AAC5 | 0.680139 | 0.74 | 0.768611 | 0.635 | 0.515417 |

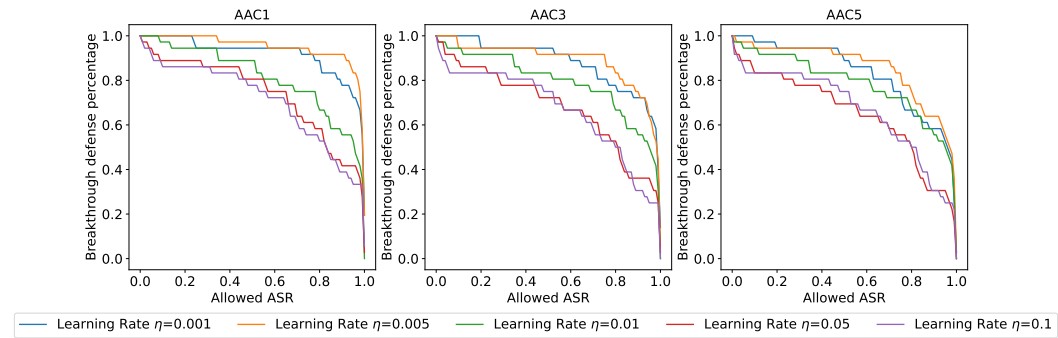

Figure 11: Impact of Learning Rates on AAC of PreActResNet18

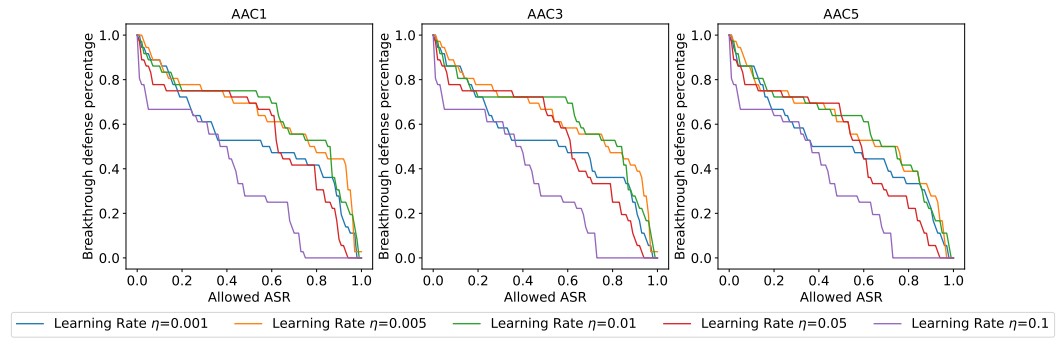

Figure 12: Impact of Learning Rates on AAC of VGG19-BN

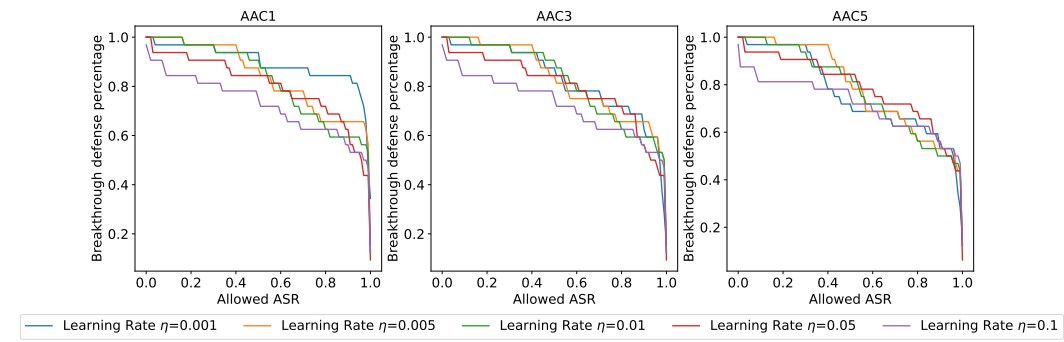

Figure 13: Impact of Learning Rates on AAC of MobileNet-v3-large

Lastly, as shown in Figure 14, the EfficientNet-B3 model displays a peak in AAV at an LR of 0.001 for all AAC levels. Similar to other models, an increase in learning rate results in a reduction of AAV, with a significant drop observed at an LR of 0.1.

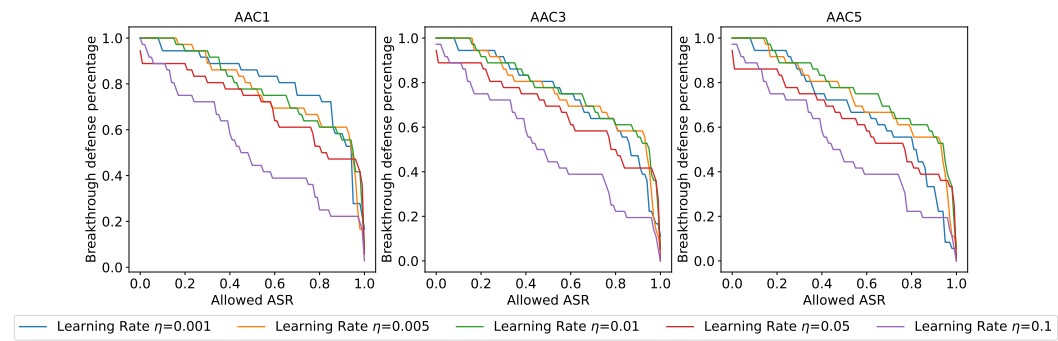

Figure 14: Impact of Learning Rates on AAC of EfficientNet-B3

# H VISUALIZATION AND INTERPRETABILITY ANALYSIS OF BACKDOOR ATTACKS

In this section, the analysis of backdoor attack methodologies was carried out using the BIG attribution method. As shown from Figure 15 to Figure 30, this assessment spanned four models: PreActRes-Net18, VGG19-BN, MobileNet-v3-large, and EfficientNet-B3, across four datasets: CIFAR-10, CIFAR-100, GTSRB, and Tiny ImageNet. The results notably highlighted the K&L method for its exceptional stealthiness. The attribution maps generated by the K&L method closely resembled those of the original, unattacked images, indicating a high degree of indiscernibility. Additionally, the trigger used in the K&L method proved to be virtually imperceptible to the naked eye, further affirming its covert nature in backdoor attacks.

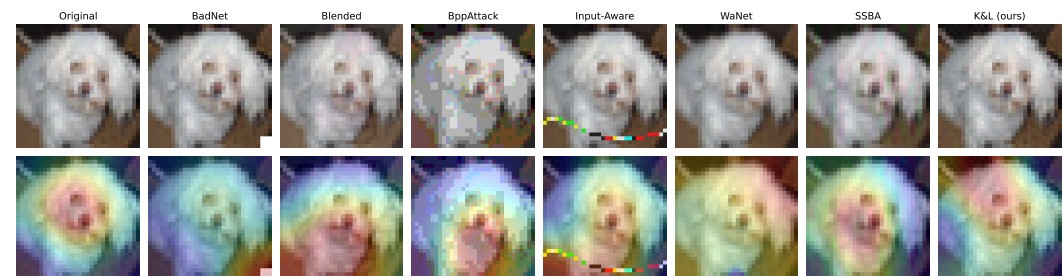

Figure 15: On the CIFAR-10 dataset, the BIG attribution is applied to the PreActResNet18 model.

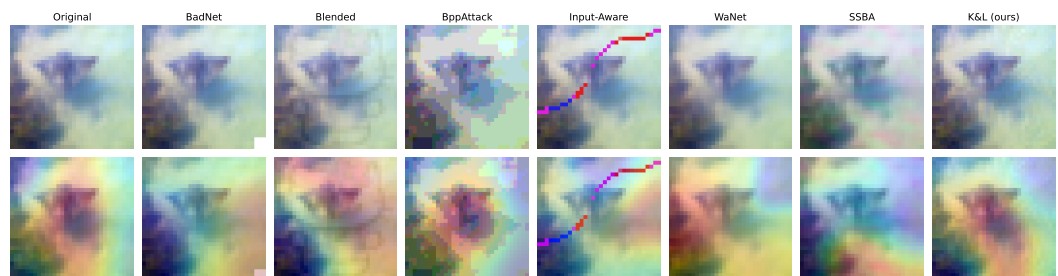

Figure 16: On the CIFAR-100 dataset, the BIG attribution is applied to the PreActResNet18 model.

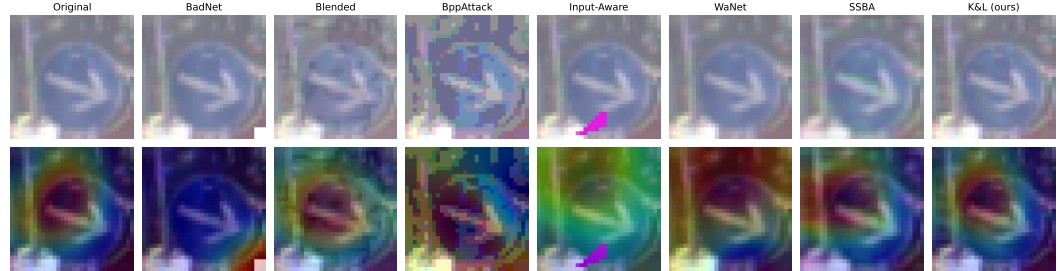

Figure 17: On the GTSRB dataset, the BIG attribution is applied to the PreActResNet18 model.

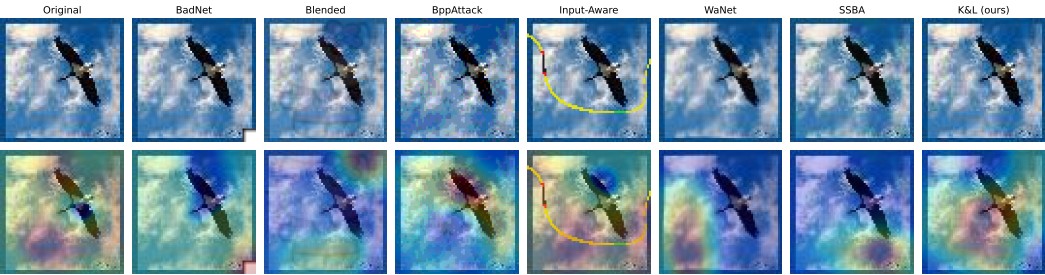

Figure 18: On the Tiny ImageNet dataset, the BIG attribution is applied to the PreActResNet18 model.

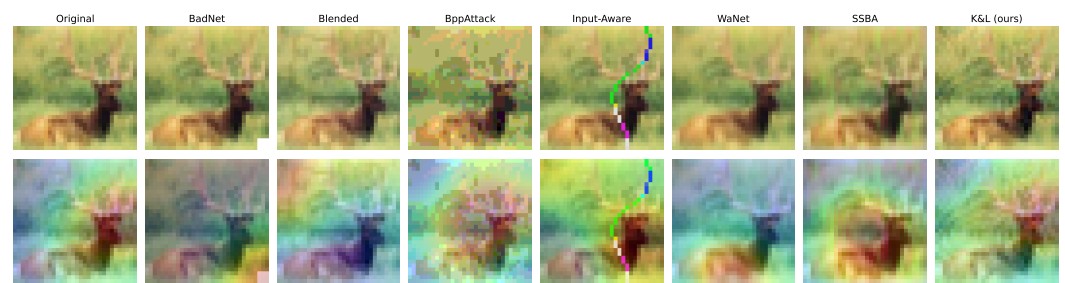

Figure 19: On the CIFAR-10 dataset, the BIG attribution is applied to the VGG19-BN model.

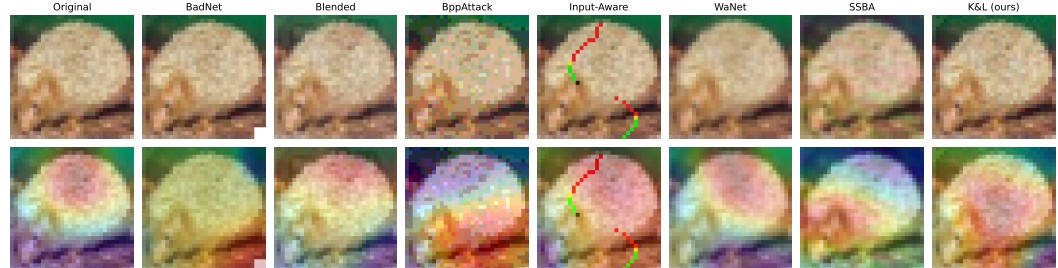

Figure 20: On the CIFAR-100 dataset, the BIG attribution is applied to the VGG19-BN model.

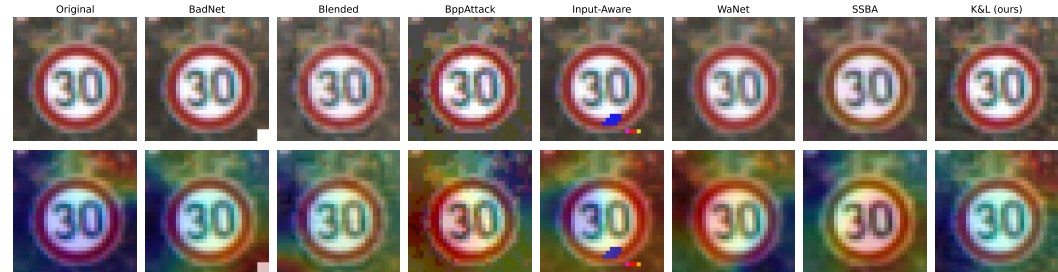

Figure 21: On the GTSRB dataset, the BIG attribution is applied to the VGG19-BN model.

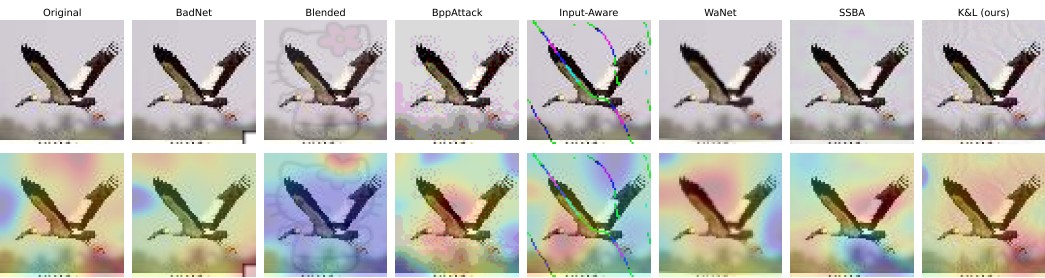

Figure 22: On the Tiny ImageNet dataset, the BIG attribution is applied to the VGG19-BN model.

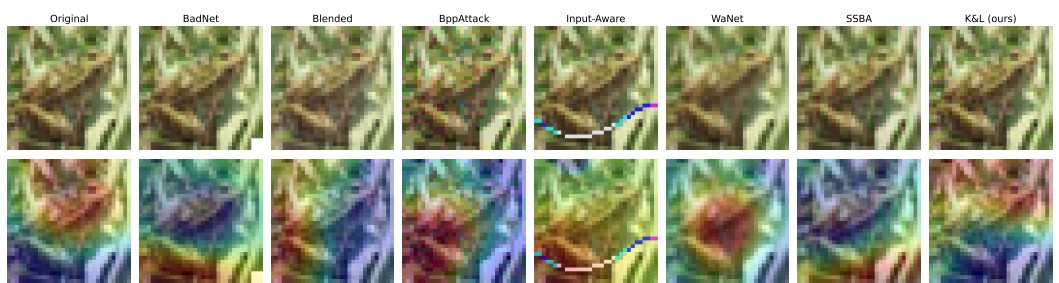

Figure 23: On the CIFAR-10 dataset, the BIG attribution is applied to the EfficientNet-B3 model.

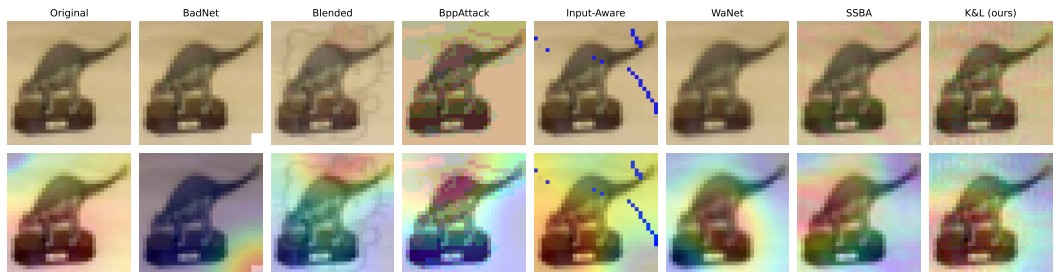

Figure 24: On the CIFAR-100 dataset, the BIG attribution is applied to the EfficientNet-B3 model.

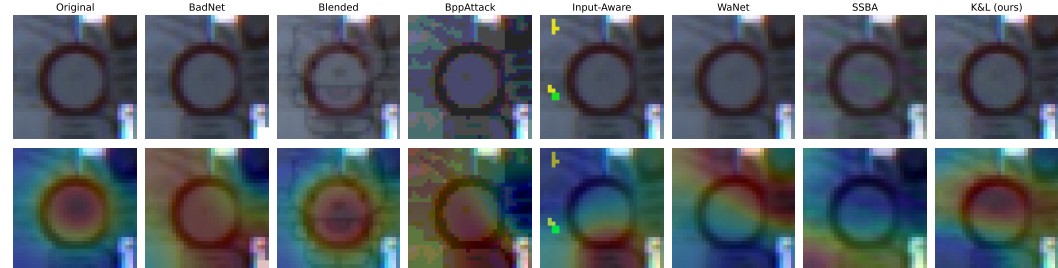

Figure 25: On the GTSRB dataset, the BIG attribution is applied to the EfficientNet-B3 model.

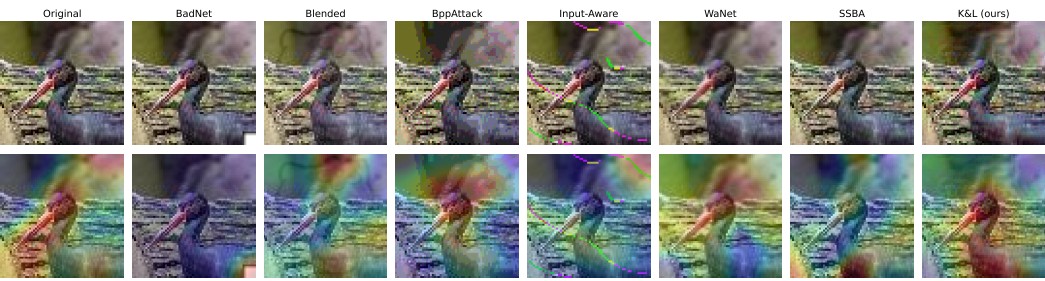

Figure 26: On the Tiny ImageNet dataset, the BIG attribution is applied to the EfficientNet-B3 model.

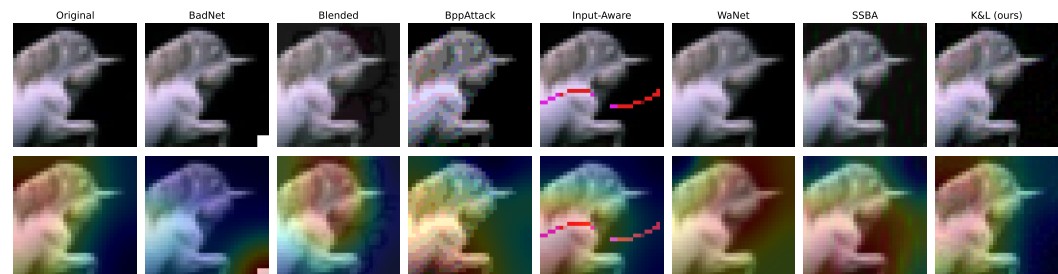

Figure 27: On the CIFAR-10 dataset, the BIG attribution is applied to the MobileNet-v3-large model.

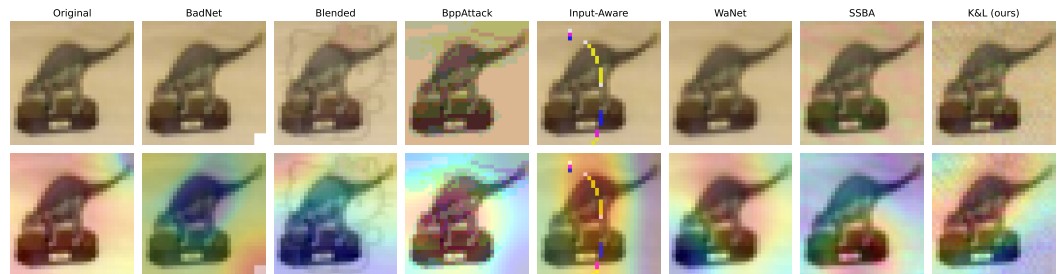

Figure 28: On the CIFAR-100 dataset, the BIG attribution is applied to the MobileNet-v3-large model.

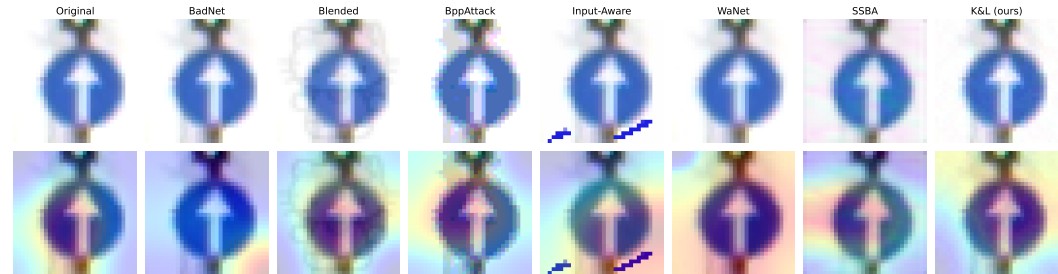

Figure 29: On the GTSRB dataset, the BIG attribution is applied to the MobileNet-v3-large model.

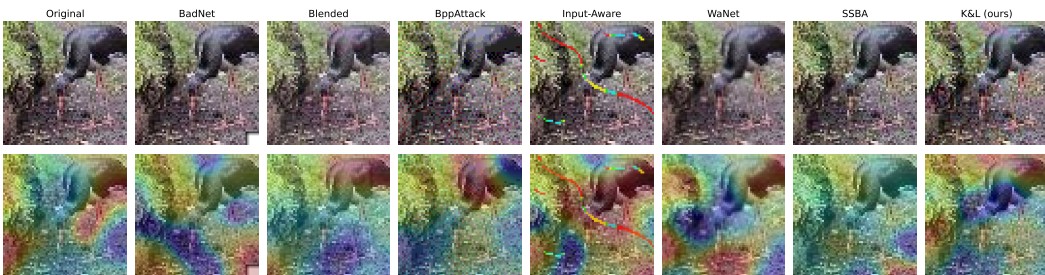

Figure 30: On the Tiny ImageNet dataset, the BIG attribution is applied to the MobileNet-v3-large model.

