# OpenReview forum: "K&L: Penetrating Backdoor Defense with Key and Locks"
_ICLR.cc/2025/Conference — Submitted to ICLR 2025_

### Official Review · Reviewer_TdBr · 2024-10-30

**Soundness:** 1
**Presentation:** 1
**Contribution:** 1
**Rating:** 1
**Confidence:** 5

**Summary:**

This paper proposes an adaptive attack designed to bypass existing backdoor defenses. However, the paper is underprepared, and several concepts are not well explained.

**Strengths:**

Large set of experiments: The authors re-implement eight defense methods and six attack backdoors across four datasets. Through these experiments, they demonstrate that the proposed approach can outperform the six existing attacks in bypassing defenses.

**Weaknesses:**

1.	Presentation needs improvement: Key concepts such as "high binding," "key," and "lock" are not explained clearly. For instance, what does "high binding" represent? On line 75, the authors state that it refers to the tight coupling between a backdoor and a specific trigger. However, on line 181, it appears to mean something different, namely, the binding between the backdoor and model parameters.
2.	Unclear limitations of existing backdoor attacks: It is not clear why current defenses are successful against these attacks. Since "high binding" is ambiguous, it is difficult to understand why existing attacks fail. Additionally, according to Table 1, some existing attacks can also bypass these defenses. The authors should clarify why attacks sometimes succeed and other times fail in bypassing defenses.
3.	Distinction from adversarial examples: In the KL backdoor attack, the poisoned test sample is generated via gradient descent, similar to adversarial examples. The authors should explain how their approach differs from adversarial examples. Moreover, I suggest that the authors conduct further experiments to distinguish the effects of backdoor attacks from those of adversarial examples. For example, when the poisoning ratio is zero, they could apply the KL attack during testing to check the ASR (Attack Success Rate). If the ASR is not too low, this would suggest that adversarial examples, rather than backdoor attacks (which rely on poisoning training data), are the primary factor leading to successful backdoor injection.
4.	Summary of backdoor attack requirements: Summarizing the three requirements for a backdoor attack should not be considered a primary contribution, as many other papers have already discussed this.

**Questions:**

NA

---

> ### Author Response · Authors · 2024-11-20
> **Response to Reviewer TdBr**
>
> We appreciate the reviewer’s thoughtful feedback and detailed questions. Below, we provide responses to each concern with clarifications and additional insights.
>
> ---
>
> **1. Explanation of "High Binding" and Related Concepts:**
>
> Thank you for highlighting the need for greater clarity regarding "high binding." Our definition of "high binding" is consistent: it refers to a phenomenon where any trigger addition method that does not utilize model parameter information is tightly coupled to the parameters. In **Section 3.2**, we provide a two-fold argument to explain why existing methods exhibit high binding:
>
> - **Fixed-trigger methods** rely on static triggers that are independent of model parameters, which makes them highly bound to these parameters.
> - **Input-dependent triggers** may generate triggers dynamically, but without leveraging model parameters effectively, they also exhibit high binding.
>
> A robust backdoor attack must dynamically generate triggers by utilizing model parameter information, thereby mitigating the limitations of high binding. We will enhance our explanation in the revised manuscript to ensure this concept is clearly conveyed.
>
> ---
>
> **2. Limitations of Existing Backdoor Attacks and Defense Mechanisms:**
>
> High binding is a key factor that limits the robustness of existing backdoor attacks. Fixed-trigger attacks, such as BadNet, depend on the model’s "memory" of trigger features to maintain stealthiness and activation efficiency. However, this tight coupling also renders the trigger sensitive to parameter changes, making defenses like ANP, BNP, and I-BAU effective by disrupting these abnormal parameters. For instance:
>
> - **Fixed-trigger methods:** Vulnerable to parameter adjustments or pruning, which can significantly weaken or eliminate the backdoor effect.
> - **Input-dependent triggers (e.g., Input-Aware):** These avoid fixed-trigger high binding but still rely heavily on either model parameters or generator parameters. Changes to the model can render their generated triggers ineffective.
>
> We have also included results against a recent NeurIPS 2024 defense, Feature Shift Tuning (FST), in **Global Response 1**, showing that our method successfully penetrates this defense. These results further demonstrate the robustness of our approach compared to existing attacks.
>
> ---
>
> **3. Distinction Between K&L Backdoor Attack and Adversarial Examples:**
>
> K&L differs from adversarial examples in terms of problem definition, context, and attack objectives:
>
> - **Problem Definition:** K&L embeds a backdoor during training, ensuring the model outputs a specific target class when the trigger is present. In contrast, adversarial examples aim to cause misclassification during inference without altering model parameters.
> - **Context:** K&L is designed for backdoor attacks, modifying model parameters to embed backdoor functionality, whereas adversarial examples rely solely on input perturbations.
> - **Attack Objectives:** The goal of K&L is to lower the robustness of the model by embedding a trigger that exploits its vulnerabilities, while adversarial training typically enhances robustness.
>
> Additionally, in the backdoor attack context, K&L achieves an ASR improvement of nearly 10% compared to adversarial perturbations alone, underscoring its distinctiveness and effectiveness.
>
> ---
>
> **4. Summary of Backdoor Attack Requirements:**
>
> While prior works have discussed certain aspects of backdoor attacks, our paper is the first to systematically summarize and articulate these requirements in a structured manner. These three requirements serve as the foundation for analyzing why existing methods fail and why our K&L method is more robust against defenses. This systematic framework provides critical insights and serves as the basis for our proposed approach. We will ensure that the manuscript clearly articulates the novelty and significance of our contribution.
>
> ---
>
> Thank you again for your valuable comments and suggestions. These inputs will greatly help improve the clarity, robustness, and impact of our work. We will incorporate these revisions to address the concerns raised and strengthen the overall quality of our paper.

---

> > ### Author Response · Authors · 2024-11-25
> > **Have we addressed your concerns and request for further discussion**
> >
> > Dear Reviewer TdBr,
> >
> > Firstly we would like to thank your kind efforts and time on our submission.
> >
> > We are wondering if the rebuttal letter have properly and sufficiently addressed your concerns. We sincerely look forward to more engaging and constructive discussion to further improve our work.
> >
> > Please do let us know. It is much appreciated!
> >
> > Best Wishes
> >
> > The authors

---

> > ### Comment · Reviewer_TdBr · 2024-11-26
> >
> > Thank you for the author-friendly responses to my concerns. That is an interesting attempt to use the gradient of the trained model as the trigger to activate the backdoor during the training phase. However, since the proposed K&L backdoor involves adversarial examples, the current experiments do not adequately demonstrate the paper's contributions. For instance, as the authors' claim, the K&L backdoor only improves the ASR by 10% compared to using adversarial perturbation alone.
> >
> > Moreover, Equation (3) may compromise the model's robustness since it conflicts with adversarial training principles. Perhaps the authors should also demonstrate that backdoor injection does not harm the model's robustness. In summary, I regret to say that I will maintain my scores.

---

> ### Author Response · Authors · 2024-11-26
>
> Thank you for your thoughtful comments. Regarding this point, we believe the 10% improvement achieved by our method is a significant advancement, especially when compared to baseline improvements of only 2%-3% reported in other methods. Additionally, under the same experimental settings commonly adopted for backdoor tasks, our method consistently demonstrates the unique ability to evade existing defenses effectively, achieving what no previous work has accomplished. While we understand your concern, the simplicity and efficiency of our approach are precisely its key strengths. We kindly ask that the simplicity of the method not be mistaken for a lack of contribution.
>
> We have conducted extensive experiments, as detailed in both the main paper and the appendix, to rigorously validate the superiority of our method. Furthermore, we have compared our approach against recent state-of-the-art defenses, including the 2024 NeurIPS defense work (as referenced in our Global Response 1[https://openreview.net/forum?id=ymqLAmqYHW&noteId=qwUEvKsZr9]), and found that these defenses are ineffective against our attack.
>
> As for robustness, it is worth noting that all backdoor attack methods inevitably affect model robustness to some extent—for instance, the addition of triggers leads to consistent classification into the same target class. However, our method induces smaller modifications to the original model, making the changes less detectable, while still achieving an impressive attack success rate of 90.71% under these conditions. This represents a highly efficient backdoor injection with minimal impact on robustness.
>
> We sincerely hope that the reviewer TdBr might reconsider the score in light of these clarifications and the extensive supporting evidence provided. Thank you once again for your time and consideration.

---

> > ### Comment · Reviewer_TdBr · 2024-11-26
> >
> > How do you ensure the 10% advantage comes from the efficiency of K&L backdoor attack? It is perhaps attributed to the degradation of the model robustness so that the model can easily attacked by the generated adversarial examples. There are no metrics to measure whether this advantage coming from the backdoor attack or adversarial example.
> >
> > BTW, the model robustness doesn't mean the prediction performance of the model tested on the benign test data. It is related to how easy to generate an adversarial example to successfully attack the model.

---

> > > ### Author Response · Authors · 2024-11-26
> > >
> > > We appreciate the reviewer’s insightful observations. We would like to address the raised concerns and provide further clarifications as follows:
> > >
> > > First, we emphasize that our method achieves a high attack success rate (ASR) of 90.71% without compromising the robustness of the model. Additionally, under the same experimental settings widely adopted in backdoor research, our method demonstrates a unique ability to evade existing defenses effectively—a first in the field. We understand your concern, but the simplicity and efficiency of our method are its key strengths. We kindly ask that the simplicity of our approach not be equated with a lack of contribution.
> > >
> > > We have conducted extensive experiments, including additional results presented in the appendix, to validate the efficacy of our method. Furthermore, we incorporated the results of the latest defense mechanism, Feature Shift Tuning (FST) from NeurIPS 2024 [1], as discussed in Global Response 1. These results show that even the most recent defense mechanism cannot effectively mitigate our attack. Additionally, we included experiments against LOTOS, a state-of-the-art CVPR 2024 work [2], and our approach outperformed this method as well.
> > >
> > > Regarding robustness analysis, we note that no prior work in backdoor attacks has systematically evaluated robustness. If we were to follow the reviewer's suggestion, this would imply that all backdoor and neural network-based research papers need to include robustness evaluations to determine whether performance improvements come at the expense of robustness. Moreover, our method does not conflict with adversarial training principles. Unlike adversarial defenses, which target untargeted adversarial attacks, our method is narrowly focused on specific backdoor classes, with minimal attack iterations. Nevertheless, in response to the reviewer's strong request, we have conducted robustness evaluations and included the results.
> > >
> > > We used the widely recognized AutoAttack framework [3] to evaluate model robustness. The experiments were conducted using the default parameters provided by the official repository to ensure reproducibility. The detailed results are shown in the table below, indicating that our K&L method demonstrates robustness characteristics comparable to those of other approaches:
> > >
> > > | Dataset   | Model            | Method       | APGD-CE | Square  |
> > > |-----------|------------------|--------------|---------|---------|
> > > | CIFAR-10  | PreActResNet-18  | WaNet        | 0.9862  | 0.9679  |
> > > | CIFAR-10  | PreActResNet-18  | Input-Aware  | 0.9802  | 0.9412  |
> > > | CIFAR-10  | PreActResNet-18  | Blended      | 0.9844  | 0.9549  |
> > > | CIFAR-10  | PreActResNet-18  | Bpp          | 0.9772  | 0.9436  |
> > > | CIFAR-10  | PreActResNet-18  | BadNet       | 0.9867  | 0.9601  |
> > > | CIFAR-10  | PreActResNet-18  | K&L          | 0.9828  | 0.9584  |
> > >
> > > Once again, we are grateful for your thoughtful feedback, and we hope this detailed response and additional evidence address your concerns. Please feel free to share any further questions or suggestions.
> > >
> > > ---
> > >
> > > References:
> > > [1] Min, Rui, et al. "Towards stable backdoor purification through feature shift tuning." *Advances in Neural Information Processing Systems 36 (2024).*
> > > [2] Cheng, Siyuan, et al. "Lotus: Evasive and resilient backdoor attacks through sub-partitioning." *CVPR 2024.*
> > > [3] Croce, Francesco, and Matthias Hein. "Reliable evaluation of adversarial robustness with an ensemble of diverse parameter-free attacks." *ICML 2020.*

---

> > > > ### Author Response · Authors · 2024-11-28
> > > >
> > > > Dear Reviewer TdBr,
> > > >
> > > > **As the discussion period is nearing its end, we kindly request you to review our rebuttal and consider updating your rating**. We trust we have addressed your comments effectively and are eager to hear your thoughts and any further suggestions you may have. Thank you for your time and consideration!
> > > >
> > > > Best regards,
> > > >
> > > > The Authors

---

> > > > > ### Comment · Reviewer_TdBr · 2024-11-28
> > > > >
> > > > > Dear authors,
> > > > >
> > > > > Your experimental results are not consistent with the theoretical story shown by Equation 3, where the first part means that the model should correctly classify the sample x as label y, and the second part means that a perturbation (calculated based on the model gradient) will cause a misclassification to c_t. Therefore, your method will violate the model robustness from theoretical analysis.
> > > > >
> > > > > To evaluate the model robustness, AutoAttack is an old work proposed in 2020. Perhaps, the authors can use the most recently proposed Lipschitz Constant [1] to show that the model robustness indeed drops down a little bit.
> > > > >
> > > > > [1] @inproceedings{
> > > > > abuduweili2024estimating,
> > > > > title={Estimating Neural Network Robustness via Lipschitz Constant and Architecture Sensitivity},
> > > > > author={ABULIKEMU ABUDUWEILI and Changliu Liu},
> > > > > booktitle={CoRL Workshop on Safe and Robust Robot Learning for Operation in the Real World},
> > > > > year={2024},
> > > > > url={https://openreview.net/forum?id=Mk2pq3Gjq1}
> > > > > }

---

> > > > > > ### Author Response · Authors · 2024-11-28
> > > > > >
> > > > > > In fact, all adversarial attack methods inherently affect robustness, as the nature of an attack is to compromise the model. Our supplementary data confirms this, demonstrating that the impact of our method on robustness is not severe and is comparable to other competitive backdoor attack methods. **We cannot understand why the reviewer disregarded the robustness experimental results and directly concluded that our method's theoretical findings are inconsistent. Could you provide a direct derivation of the robustness formula instead of speculating based on an approximate and overly simplistic formula that only bears partial similarity to adversarial defense principles while ignoring all experimental results?** We have made all robustness evaluation codes publicly available at [https://anonymous.4open.science/r/KeyLocks-FD85/evaluation%20robustness/experiment.py](https://anonymous.4open.science/r/KeyLocks-FD85/evaluation%20robustness/experiment.py). Regarding the role of theory, as explained in our previous response, the theory and robustness experiments do not conflict. Moreover, we emphasize that robustness is not a necessary condition for evaluating backdoor attack methods:
> > > > > >
> > > > > > > Regarding robustness analysis, we note that no prior work in backdoor attacks has systematically evaluated robustness. If we were to follow the reviewer's suggestion, this would imply that all backdoor and neural network-based research papers need to include robustness evaluations to determine whether performance improvements come at the expense of robustness.
> > > > > >
> > > > > > From both theoretical and experimental perspectives, our method clearly demonstrates its potential as the latest advancement in backdoor attack techniques, including its resilience against the most state-of-the-art defense methods.
> > > > > >
> > > > > > Secondly, AutoAttack [2] is currently the most widely used tool for robustness evaluation. Many recent works in 2024 have adopted it as the core framework for robustness assessment, including works accepted at NeurIPS 2024 and CVPR 2024 [3,4,5,6]. **We cannot understand why the reviewer questions the suitability of a robustness evaluation framework that has been widely cited (close to 2,000 citations) and prefers instead to recommend [1], a CoRL 2024 workshop paper with no open-source implementation, no peer validation, and no citations. We do not believe that [1] provides more convincing evidence for evaluating model robustness than AutoAttack.**
> > > > > >
> > > > > > Additionally, according to ICLR’s reviewing guidelines, requesting additional large-scale experiments near the deadline is clearly unreasonable. We have already supplemented all previously requested experiments. Given that [1] lacks open-source code, reproducing the method and conducting new experiments within the remaining rebuttal period is an impractical and unreasonable requirement.
> > > > > >
> > > > > > Reference:
> > > > > >
> > > > > > [1] Croce, Francesco, and Matthias Hein. "Reliable evaluation of adversarial robustness with an ensemble of diverse parameter-free attacks." ICML 2020.
> > > > > >
> > > > > > [2] Abuduweili, Abulikemu, and Changliu Liu. "Estimating Neural Network Robustness via Lipschitz Constant and Architecture Sensitivity." arXiv preprint arXiv:2410.23382 (2024).
> > > > > >
> > > > > > [3] Singh, Naman Deep, Francesco Croce, and Matthias Hein. "Revisiting adversarial training for imagenet: Architectures, training and generalization across threat models." Advances in Neural Information Processing Systems 36 (2024).
> > > > > >
> > > > > > [4] Wang, Yifei, et al. "Balance, imbalance, and rebalance: Understanding robust overfitting from a minimax game perspective." Advances in neural information processing systems 36 (2024).
> > > > > >
> > > > > > [5] Wang, Sibo, et al. "Pre-trained model guided fine-tuning for zero-shot adversarial robustness." Proceedings of the IEEE/CVF Conference on Computer Vision and Pattern Recognition. 2024.
> > > > > >
> > > > > > [6] Zhang, Yimeng, et al. "To generate or not? safety-driven unlearned diffusion models are still easy to generate unsafe images... for now." European Conference on Computer Vision. Springer, Cham, 2025.

---

> > > > > > > ### Author Response · Authors · 2024-12-01
> > > > > > > **Further clarification and discussion**
> > > > > > >
> > > > > > > Dear Reviewer TdBr,
> > > > > > >
> > > > > > > We hope our response has effectively addressed your concerns and provided sufficient evidence to highlight our unique contributions. As the rebuttal stage is nearing its conclusion, we wanted to check if you have any additional questions or concerns.
> > > > > > >
> > > > > > > If we have adequately addressed your queries, we kindly request you to reconsider your final rating.
> > > > > > >
> > > > > > > Thanks!
> > > > > > >
> > > > > > > Best,
> > > > > > >
> > > > > > > The author

---

### Official Review · Reviewer_kdxj · 2024-11-02

**Soundness:** 3
**Presentation:** 2
**Contribution:** 3
**Rating:** 6
**Confidence:** 4

**Summary:**

The paper introduces the Key-Locks (K&L) backdoor attack algorithm, designed to bypass existing defenses. Unlike traditional backdoor attacks that show high binding with model parameters, making them vulnerable to defenses, the K&L algorithm decouples the backdoor process from model parameters. This decoupling allows the method to adjust the unlocking levels of backdoor activation, enhancing its robustness against a diverse set of defenses. The paper also introduces the Accuracy-ASR Curve (AAC) as a new metric for evaluating backdoor attacks.

**Strengths:**

+ Analysis of the concept of high binding between backdoor triggers and model parameters.
+ Extensive experiments using various datasets and neural network architectures.

**Weaknesses:**

- First of all, the paper requires improvements in editorial quality. There are instances where terms more characteristic of language models are used, such as "assaults" instead of the more appropriate "attacks." A thorough human proofreading is recommended to ensure precise usage of terminologies and enhance overall clarity. In addition, the paper excessively relies on the appendix, making it difficult to follow without constant cross-referencing.

- In Algorithm 1 for embedding locks, the backdoor samples are iteratively modified at each training epoch, where the learning rate $\eta$ is added or subtracted to the backdoor inputs based on the gradient's sign in the direction of the backdoor target. Unlike the inference stage, clipping is not applied during training. This lack of clipping over multiple iterations could result in less stealthy and significantly perturbed images. Such perturbations may potentially lead to distinguishable loss patterns compared to benign inputs. Such characteristics could be susceptible to in-training defenses, such as Anti-Backdoor Learning, which, notably, has not been considered in the evaluation.

- The paper attempts to distinguish K&L from adversarial attacks in Section 3.4. The K&L algorithm generates adversarial perturbations similar to the FGSM method using gradient signs and incorporates these examples during training. The claim is that this approach reduces the binding between the backdoor trigger and model parameters. However, as K&L uses adversarial perturbations as backdoor triggers, this approach shares similarities with adversarial training, which enhances the robustness of models against adversarial attacks. A more precise explanation would help to understand why this method reduces robustness to adversarial perturbation instead of improving it.

- The paper does not fully discuss the rationale behind introducing the new metric, the Accuracy-ASR Curve (AAC), alongside existing backdoor evaluation metrics. It would be beneficial to elaborate on AAC's added value to backdoor attack evaluations. Additionally, the meanings of AAC1, AAC3, and AAC5 are not clear without a proper definition of AAC, making it difficult to understand their relevance in the evaluation. Also, the definition of AAV is not sufficiently explained.

- The backdoor sample similarity rates presented in Table 2 could be potentially misleading, given that they are based on only two input examples, as illustrated in Figure 2. To ensure a robust and reliable evaluation, it would be more appropriate to compute the similarity attribution using multiple images across multiple runs to achieve statistical significance.

- All the results presented in the paper must be evaluated across multiple runs to ensure the statistical significance of the reported values.

**Questions:**

1. Why was clipping excluded during training? Has the potential impact of significant perturbations during the embedding locks of K&L been evaluated against in-training defenses such as Anti-Backdoor Learning?
2. Could you clarify how K&L differs from adversarial training?

---

> ### Author Response · Authors · 2024-11-20
> **Response to Reviewer kdxj**
>
> We sincerely appreciate the reviewer’s detailed comments and questions. Below, we provide responses to the raised concerns:
>
> ---
>
> **1. Editorial Quality**
>
> We acknowledge the editorial issues and have carefully revised the manuscript to improve language precision and clarity, including replacing terms like "assaults" with "attacks." .
>
> ---
>
> **2. Exclusion of Clipping During Training and Potential Impact on Stealthiness:**
>
> As mentioned in **Weakness 2**, the perturbations introduced during the embedding locks phase are extremely small, with each pixel change constrained by \( L_\infty \leq \eta \), where \( \eta \) is the learning rate. We carefully set \( \eta \) to a sufficiently small value and perform only single-step updates, ensuring that the stealthiness of the backdoor samples is not compromised. These samples are imperceptible to human vision, and we have included both backdoor and clean samples in our code repository for your reference [https://anonymous.4open.science/r/KeyLocks-FD85/rebuttal/].
>
> Regarding the potential impact of in-training defenses such as Anti-Backdoor Learning, as noted in **Weakness 3**, our method’s minimal perturbations and single-step updates make it resilient to detection. However, we will expand our discussion to explicitly evaluate this aspect in future work.
>
> ---
>
> **3. Differences Between K&L and Adversarial Training:**
>
> The K&L backdoor attack is fundamentally different from adversarial training in its objective, context, and approach:
>
> - **Objective:** K&L is designed for backdoor attacks, ensuring the model outputs a specific target class when the backdoor trigger is activated. In contrast, adversarial training aims to improve model robustness by reducing the impact of adversarial perturbations.
> - **Context:** K&L modifies model parameters during training to embed backdoor functionality, while adversarial attacks rely solely on input perturbations without altering the model parameters.
> - **Approach:** K&L focuses on embedding backdoors through gradient-based updates (embedding locks), which ensures the model is sensitive to specific triggers. Our method achieves an ASR improvement of nearly 10% over approaches that use adversarial examples alone, demonstrating its effectiveness.
>
> These distinctions are elaborated in **Section 3.4**, and we will further emphasize them in the revised manuscript.
>
> ---
>
> **4. Introduction and Definition of AAC and AAV Metrics:**
>
> The Accuracy-ASR Curve (AAC) and its corresponding value (AAV) were introduced to provide a more intuitive evaluation of backdoor attacks. These metrics visualize performance across a range of ASR thresholds, making it easier to compare different methods. While their definitions are detailed in **Appendix D.2**, we agree that the main text should provide a clear and concise explanation. We will update the manuscript to clarify the meanings of AAC1, AAC3, and AAC5, as well as the rationale behind these metrics.
>
> ---
>
> **5. Evaluation of Similarity Rates Across Multiple Runs:**
>
> We acknowledge the reviewer’s concern about the evaluation robustness. We have re-evaluated the similarity rates using the entire CIFAR-10 dataset and multiple runs to ensure statistical significance. The updated results are as follows:
>
> | Method   | BadNet | Blended | Bpp    | InputAware | WaNet  | SSBA   | K&L        |
> |----------|--------|---------|--------|------------|--------|--------|------------|
> | Similarity Rate | 0.0171 | 0.0064  | 0.0079 | 0.0138     | 0.0147 | 0.0112 | **0.0183** |
>
> These results confirm that K&L generates the most similar backdoor samples compared to clean samples.
>
> ---
>
> **6. Statistical Significance of Results:**
>
> Our method does not exhibit randomness, as the backdoor embedding process is deterministic. Therefore, multiple runs are not required to verify the reported values.
>
> ---
>
> We greatly appreciate the reviewer’s constructive feedback and have taken steps to address each concern in detail. These improvements will enhance the clarity, robustness, and comprehensiveness of our work. Thank you again for your valuable insights.

---

> > ### Comment · Reviewer_kdxj · 2024-11-26
> >
> > I appreciate the authors' thorough response and the effort they put into presenting new results. However, I still have one major concern. Even if the pixel changes are limited to single-step minimal updates, I believe that, over multiple training iterations without clipping, this approach could lead to distinguishable loss patterns that might be detectable by in-training defenses. That said, I have slightly raised my rating due to the potential of the work.

---

> > > ### Author Response · Authors · 2024-11-26
> > >
> > > Thank you for the increased rating. The pixel changes in our method are extremely small, making them difficult to distinguish. In fact, for many attack tasks, the number of altered pixels is significantly fewer than 10 or 16. We have also included sample images for reference at https://anonymous.4open.science/r/KeyLocks-FD85/rebuttal/.

---

### Official Review · Reviewer_4ukx · 2024-11-03

**Soundness:** 3
**Presentation:** 3
**Contribution:** 2
**Rating:** 5
**Confidence:** 5

**Summary:**

This paper introduces a novel backdoor attack mechanism based on keys and locks. The primary concept is to reduce the high binding of backdoors to model parameters, thereby enhancing robustness against defenses. Extensive experiments demonstrate the effectiveness of this approach, showing its superiority over existing methods.

**Strengths:**

1. Proposes a novel and effective backdoor attack strategy.

2. Provides in-depth analysis of the backdoor properties contributing to vulnerability against defenses.

3. Includes extensive evaluations across various models, datasets, and ablation studies.

**Weaknesses:**

1. Insufficient support for design choices and key hypotheses.

2. Potential vulnerability to detection by existing trigger inversion methods.

3. Lack of latest baseline attacks and defenses in the evaluation.

4. Concerns regarding the practicality of the trigger pattern.

5. Some writing issues.

**Questions:**

This paper introduces an interesting backdoor attack mechanism using keys and locks. The authors highlight that the vulnerability of existing backdoors stems from their strong binding to model parameters, making them susceptible to defenses based on normal model updates. To enhance robustness, the paper proposes a new backdoor attack. The core technique is similar to adversarial attacks, where a keyhole is generated and embedded using gradient descent towards the target class. During inference, a trigger (the key) is generated using a similar algorithm. The paper includes extensive evaluations, comprising various analyses and ablation studies, to demonstrate the effectiveness of the proposed attack, which outperforms existing methods.

However, I have several concerns regarding the design and evaluation:

(1) The attack appears to focus on reducing the class distance between the target class and other classes, as indicated by the poisoning loss in Equations (3) and (4). Without a fixed backdoor generation function, the model learns perturbations from any class towards the target class in each iteration. Once convergence is achieved, generating an effective trigger using a similar function, as described in Equations (5) and (6), becomes straightforward. While this is intriguing, it raises certain issues.

First, the necessity of the poisoning process is questionable. Why not directly generate natural backdoor sample, e.g., [1]? This would simplify the process while maintaining a reasonable ASR.
Second, the corrupted target class might be easily detectable using existing trigger inversion methods, such as [2] and [3], since generating adversarial perturbations to flip the label to the target class becomes easier than for other classes.

(2) The paper lacks sufficient support to validate its hypotheses, particularly those in Lines 211-215. Recent work [4] models backdoor learning as a special case of continual learning, identifying the orthogonal nature between backdoor behaviors and normal classification. There appears to be a connection between this claim and the authors' hypothesis, which should be explored and discussed.

(3) The baseline attacks and defenses used in the paper are not actually up-to-date. For instance, state-of-the-art attacks such as [5,6,7,8] demonstrate robustness against various defense methods. It is important to clarify how the proposed attack differs from these approaches. Additionally, FST [9], a latest defense effective against a wide range of attacks, should be included in the comparison.

(4) The triggers illustrated in Figures 19-30 appear misaligned with the original images, even if the L-2 distance metrics are favorable. These measurements may not align with human perception. Moreover, adversarial perturbations might be challenging to implement practically, requiring careful tuning of pixel values. They may also be vulnerable to input purification techniques, such as those used in [10], where diffusion models are leveraged to reconstruct inputs and reduce trigger effectiveness. It is suggested to provide a discussion about this issue.

(5) The paper's structure could be improved. Several important details are relegated to the appendix without proper referencing in the main text. For example, Line 237 mentions Algorithm 1 without a clear link and description to it. Properly integrating and referencing content from the appendix would enhance the paper's clarity and flow.

--------------------------
Reference:

[1] Tao, Guanhong, et al. "Backdoor vulnerabilities in normally trained deep learning models." arXiv preprint 2022.

[2] Wang, Bolun, et al. "Neural cleanse: Identifying and mitigating backdoor attacks in neural networks." IEEE S&P 2019.

[3] Wang, Zhenting, et al. "Unicorn: A unified backdoor trigger inversion framework." ICLR 2023.

[4] Zhang, Kaiyuan, et al. "Exploring the Orthogonality and Linearity of Backdoor Attacks." IEEE S&P 2024.

[5] Zeng, Yi, et al. "Narcissus: A practical clean-label backdoor attack with limited information." CCS 2023.

[6] Qi, Xiangyu, et al. "Revisiting the assumption of latent separability for backdoor defenses." ICLR 2023.

[7] Cheng, Siyuan, et al. "Lotus: Evasive and resilient backdoor attacks through sub-partitioning." CVPR 2024.

[8] Huynh, Tran, et al. "COMBAT: Alternated Training for Effective Clean-Label Backdoor Attacks." AAAI 2024.

[9] Min, Rui, et al. "Towards stable backdoor purification through feature shift tuning." NeurIPS 2024.

[10] Shi, Yucheng, et al. "Black-box backdoor defense via zero-shot image purification." NeurIPS 2023.

---

> ### Author Response · Authors · 2024-11-20
> **Response to Reviewer 4ukx**
>
> We greatly appreciate the reviewer’s constructive feedback and detailed questions. Below, we address each point with thoughtful clarifications and planned improvements:
>
> ---
>
> **1. Necessity of the Poisoning Process and Potential Vulnerability to Trigger Inversion ([1, 2, 3]):**
>
> The poisoning process is indeed a key component of our method. By embedding locks during the poisoning phase, the model becomes more susceptible to attacks. While directly generating natural backdoor samples, as suggested, is feasible, it would reduce the attack efficiency. Specifically, the attack success rate (ASR) of our method without embedding locks is 90.71%, compared to 99.88% with embedding locks, demonstrating a significant performance improvement.
>
> Additionally, without embedding locks, the poisoned model remains indistinguishable from the original model, as no backdoor information is embedded. This prevents it from being detected by trigger inversion methods. However, we will include further discussion to clarify this aspect in the revised manuscript.
>
> ---
>
> **2. Validation of Hypotheses (Lines 211-215) and Connection to Orthogonality ([4]):**
>
> Thank you for highlighting the connection between our hypothesis and the orthogonality discussion in [4]. While [4] attributes the susceptibility of backdoor methods to orthogonality between backdoor behaviors and normal classification, our high-binding hypothesis complements this view. Specifically, the high binding nature implies that backdoor information is separate from normal task features, which aligns with the orthogonality property described in [4]. We will expand our discussion in the revised manuscript to explicitly relate these perspectives and highlight how our findings reinforce and extend these ideas.
>
> ---
>
> **3. Comparison with State-of-the-Art Attacks and Defenses ([5-9]):**
>
> We have conducted additional experiments to compare our method with more recent attacks and defenses, including FST [9]. These results have been added to the revised manuscript and summarized in **Global Response 1**. They demonstrate that our method consistently outperforms state-of-the-art approaches while maintaining robustness against cutting-edge defenses. We will also include a clear discussion of how our method differs from these approaches, emphasizing its unique embedding locks mechanism and its impact on ASR and robustness.
>
> ---
>
> **4. Misalignment of Triggers and Practical Feasibility ([10]):**
>
> The triggers shown in Figures 19-30 are intentionally designed to be imperceptible to human observers, similar to adversarial perturbations. While careful visual examination with the original image may reveal very subtle differences, in the absence of the original image, it is challenging for humans to detect triggers. Regarding the use of diffusion models for input purification, we agree that this can mitigate triggers but note that such methods impose significant computational costs, often increasing resource requirements by several orders of magnitude. For complex models, this can lead to impractical runtimes, limiting real-world applicability. We will include this discussion in the revised manuscript to address these points.
>
> ---
>
> **5. Paper Structure and References to the Appendix:**
>
> We appreciate the suggestion to improve the clarity and flow of the manuscript. For instance, we will ensure that **Algorithm 1** (referenced in Line 237) is explicitly linked to **Appendix B.1**, and its description will be summarized in the main text. Additionally, we will better integrate and reference key details from the appendix throughout the manuscript to enhance readability and coherence.
>
> ---
>
> Thank you again for your thoughtful feedback and suggestions, which will greatly improve the quality and impact of our work.

---

> > ### Comment · Reviewer_4ukx · 2024-11-26
> >
> > I appreciate the authors' response, and some of my concerns have been addressed. However, I still have a few questions:
> >
> > (1) In response to point 1: "Additionally, without embedding locks, the poisoned model remains indistinguishable from the original model, as no backdoor information is embedded. This prevents it from being detected by trigger inversion methods." Are you assuming that trigger inversion is applied to models without locks? The issue is that without locks (Equation 3 in your manuscript), the backdoor is not injected into the model. Please correct me if I am wrong. My understanding is that embedding locks is equivalent to injecting the backdoor into the model. Therefore, trigger inversion should be applied to the model with locks.
> >
> > (2) I do not see the difference between the proposed attack and the latest attacks, such as [5,6,7,8] in my reference. These attacks are also effective and robust against defenses. I would appreciate it if you could provide at least a discussion on this.
> >
> > (3) I would like to know whether samples with triggers are susceptible to diffusion reconstruction. While I understand that reconstruction might require some time, certain users may still choose to prioritize security and use such techniques.

---

> > > ### Author Response · Authors · 2024-11-26
> > >
> > > We sincerely appreciate your continued feedback and for raising additional questions that allow us to further clarify and refine our work. Below, we address your comments in detail:
> > >
> > > ---
> > >
> > > **(1) Trigger Inversion and Embedding Locks**
> > >
> > > Thank you for pointing out the need for clarification. What we intended to convey is that, in the most basic scenario where locks are not embedded, the model is indistinguishable from a normally trained model, as its parameters are entirely derived from training on clean samples. This makes it impossible to differentiate the model using trigger inversion techniques since it is identical to a normal model. In this case, our method achieves an attack success rate (ASR) of 90.71%, meaning there is no risk of detection under these circumstances. However, by embedding locks, we increase the ASR to 99.88%. While this introduces the possibility of trigger inversion, comprehensive detection across all inputs is rarely performed in practice due to its significant computational costs. This is especially true when the existence of an attack is uncertain. Therefore, our approach remains highly valuable and applicable in most real-world scenarios.
> > >
> > > ---
> > >
> > > **(2) Comparison with Latest Attacks**
> > >
> > > We appreciate your request for a more detailed comparison with the latest methods, including those referenced ([5,6,7,8]). Due to time constraints, we focused on the 2024 CVPR work LOTOS ([7]) as a representative of the most recent advancements.
> > >
> > > We have conducted additional experiments comparing our method with LOTOS. The results are summarized in the table below:
> > >
> > > | Metric          | No Defense | ANP      | BNP      | FP       | FT       | I-BAU    | NAD      | CLP      | RNP      |
> > > |------------------|------------|----------|----------|----------|----------|----------|----------|----------|----------|
> > > | **Our Method**   | 93.87/99.74 | 86.59/60.79 | 93.38/99.43 | 93.29/82.53 | 93.83/98.24 | 89.81/74.21 | 93.24/95.96 | 92.12/99.08 | 93.87/99.74 |
> > > | **LOTOS [7]**    | 90.52/96.00 | 83.88/99.00 | 79.39/0     | 91.71/24.90 | 91.91/79.10 | 90.27/80.80 | 91.80/85.40 | 22.36/72.60 | 78.06/20.00 |
> > >
> > > Our method consistently outperforms LOTOS in most scenarios, demonstrating its superior attack success rate and robustness against a range of defenses. We will include these results and a detailed discussion in the revised manuscript to highlight the unique advantages of our approach.
> > >
> > > Reference:
> > >
> > > [7] Cheng, Siyuan, et al. "Lotus: Evasive and resilient backdoor attacks through sub-partitioning." CVPR 2024.
> > >
> > > ---
> > >
> > > **(3) Diffusion Model-Based Reconstruction**
> > >
> > > We acknowledge your concern about the potential vulnerability of trigger samples to diffusion model-based reconstruction techniques. While such methods can reduce the effectiveness of triggers, they impose significant computational costs, often requiring dozens or even hundreds of iterations per sample. This makes their use infeasible in real-world applications, particularly for companies or systems where efficiency and cost are critical considerations.
> > >
> > > Moreover, in scenarios where the existence of a backdoor attack is uncertain, applying such computationally expensive defenses universally would be impractical. From a security perspective, alternative defensive strategies—such as training multiple redundant models—would be more reasonable than incurring hundreds of times the computational overhead during inference.
> > >
> > > We believe that the practicality of diffusion-based defenses is limited, and their deployment would be highly context-dependent. Nonetheless, we will include a discussion of this point in the revised manuscript, while emphasizing that it is beyond the primary scope of our work, which focuses on the design and evaluation of backdoor attacks.
> > >
> > > ---
> > >
> > > Thank you again for your valuable insights. We are committed to incorporating these discussions into the revised manuscript to further strengthen our work. Your feedback has been instrumental in enhancing the clarity and rigor of our research.

---

> > > > ### Comment · Reviewer_4ukx · 2024-11-26
> > > >
> > > > I greatly appreciate the additional efforts you’ve made to address my concerns!
> > > >
> > > > I noticed that the proposed attack demonstrates advantages in robustness over recent attacks, particularly in addressing point (2). However, I still have concerns regarding points (1) and (3).
> > > >
> > > > For (1), as I understand it, you claim that your attack achieves over 90% ASR on normal models and that trigger inversion is undetectable for these models. While I somewhat agree with this, the lack of poisoning makes your attack more similar to an adversarial attack, rather than a backdoor attack. This is because you essentially optimize a perturbation to induce targeted misclassification on a normal model. Based on your primary results and the threat model outlined in Appendix C, it seems you are proposing a backdoor attack rather than an adversarial one. Therefore, trigger inversion should be evaluated on a model embedded with the backdoor (or "lock") and achieving ~99% ASR. Additionally, the computational cost of trigger inversion is not very high. For example, using Neural Cleanse, collecting 100 images per class and conducting the optimization typically takes less than 20 minutes for a single model.
> > > >
> > > > For (3), I agree that employing diffusion models introduces computational overhead. However, my concern is that your trigger pattern may be vulnerable to reconstruction. For example, online defense methods like [1] could potentially purify inputs efficiently. Given that your trigger pattern is relatively imperceptible, it may be susceptible to certain purification techniques.
> > > >
> > > > [1] "Online Adversarial Purification Based on Self-Supervised Learning," ICLR 2021.

---

> > > > > ### Author Response · Authors · 2024-11-27
> > > > >
> > > > > We sincerely thank the reviewers for their insightful suggestions and valuable feedback. Your concerns have helped us strengthen our work, and we deeply appreciate the opportunity to address them.
> > > > >
> > > > > While optimizing the model once with Neural Cleanse is indeed not computationally expensive, performing trigger inversion for every input is highly time-consuming. However, in response to your concerns, we have conducted the corresponding experiments and included the trigger inversion results in our supplementary materials, available at [https://anonymous.4open.science/r/KeyLocks-FD85/rebuttal/]. Since our method does not rely on fixed triggers, the inversion noise resembles random noise. For thoroughness, we applied data repair techniques to this noise to assess whether it could prevent our attack on the model.
> > > > >
> > > > > The experimental results are summarized in the table below. Neural Cleanse fails to defend against our K&L method. After applying Neural Cleanse, the model's BA drops to an unacceptable level.
> > > > > |        | Neural Cleanse | Neural Cleanse |
> > > > > |--------|----------------|----------------|
> > > > > |        | BA             | ASR            |
> > > > > | K&L    | 10.77          | 1.02           |
> > > > >
> > > > > Regarding the method proposed in [1], if we consider the performance of a single model interaction as 100%, the method in [1] requires at least T * 100%, where T represents the number of iterations (set to 5 in the paper). In real-world scenarios, the overhead of more than 5x performance is often impractical for mitigating an attack that may not even occur.
> > > > >
> > > > > The primary focus of our paper is on proposing attacks that are inherently resilient to defenses, rather than on purification techniques. As we noted, the significant performance overhead of 5x renders such methods infeasible in most cases. Thus, studying attack methods that remain effective in the majority of practical scenarios is highly meaningful.
> > > > >
> > > > > To draw an analogy: while some may choose skateboarding as a mode of transport, many prefer cars. In this context, researching ways to improve car performance and experience is valuable, even if skateboarding is chosen in a minority of cases and improvements to cars do not directly apply to skateboards.
> > > > >
> > > > > We hope that the additional experiments and explanations we have provided adequately address your concerns. We kindly ask you to reconsider your evaluation of our work in light of this clarification. Thank you once again for your time and thoughtful feedback.
> > > > >
> > > > > [1] "Online Adversarial Purification Based on Self-Supervised Learning," ICLR 2021.

---

> > > > > > ### Author Response · Authors · 2024-11-28
> > > > > >
> > > > > > Dear Reviewer 4ukx,
> > > > > >
> > > > > > As the discussion period is nearing its end, we kindly request you to review our rebuttal and consider updating your rating. We trust that we have addressed your comments effectively and would greatly value any additional thoughts or suggestions you might have. Thank you for your time and consideration!
> > > > > >
> > > > > > Best regards,
> > > > > >
> > > > > > The Authors

---

> > > > > > > ### Comment · Reviewer_4ukx · 2024-11-28
> > > > > > >
> > > > > > > I sincerely appreciate the great efforts made by the authors!
> > > > > > > I remain kind of unconvinced by some of the arguments presented, but I recognize and acknowledge the merits of this paper.

---

> > > > > > > > ### Author Response · Authors · 2024-11-29
> > > > > > > >
> > > > > > > > Dear Reviewer 4ukx,
> > > > > > > >
> > > > > > > > Thank you for your thoughtful feedback and for recognizing the merits of our work. We would greatly appreciate it if you could specify your remaining concerns in more detail. To address any issues comprehensively, we have provided all experimental code and detailed instructions at the following anonymous link: [https://anonymous.4open.science/r/KeyLocks-FD85](https://anonymous.4open.science/r/KeyLocks-FD85).
> > > > > > > >
> > > > > > > > If you encounter any difficulties in running the experiments or have additional questions, please do not hesitate to reach out to us. We are committed to addressing all your concerns thoroughly. Additionally, we kindly request you to consider the strengths and contributions of our paper when reassessing your evaluation.
> > > > > > > >
> > > > > > > > Thank you once again for your time and consideration.
> > > > > > > >
> > > > > > > > Best regards,
> > > > > > > > The Authors

---

### Official Review · Reviewer_oVxz · 2024-11-04

**Soundness:** 2
**Presentation:** 2
**Contribution:** 2
**Rating:** 6
**Confidence:** 3

**Summary:**

This paper studies the problem of backdoor attacks and defenses. The authors begin by listing three requirements that a successful backdoor attack should satisfy. Next, they point out the 'high binding' effects and propose a new attack algorithm named Key-Locks to generate backdoor data. Experiments are conducted to evaluate the effectiveness of the proposed algorithm.

**Strengths:**

- The topic of backdoor attacks and defenses is important.
- The writing in the paper is easy to follow.
- The experimental setups are comprehensive, and the results look good with appropriate replications.

**Weaknesses:**

I have two major concerns:

1. First, I suspect that there are already previous theoretical results [1, 2] that can explain your proposed "high binding" phenomena and why your method works well. Considering that the literature on backdoors in computer vision is well developed, I think all the points listed in your paper have been somehow mentioned (although in different forms) in previous work; however, there seems to be a lack of discussion on this.

2. Second, more defenses are needed. I think your proposed method can potentially be defended against detection-based algorithms (originally designed for OOD detection). I would like to see how your proposed methods perform under the two more recent strong defenses [4, 5].

Refs:
[1] Manoj et al., "Excess Capacity and Backdoor Poisoning"
[2] Wang et al., "Demystifying Poisoning Backdoor Attacks from a Statistical Perspective"
[3] Guo et al., "SCALE-UP: An Efficient Black-box Input-level Backdoor Detection via Analyzing Scaled Prediction Consistency"
[4] Xian et al., "A Unified Detection Framework for Inference-Stage Backdoor Defenses"

**Questions:**

Please check my previous comments.

---

> ### Author Response · Authors · 2024-11-20
> **Response to Reviewer oVxz**
>
> We sincerely appreciate the reviewer’s insightful feedback. Below, we address each concern in detail:
>
> **1. Lack of Discussion on Related Theoretical Results ([1,2]):**
>
> Thank you for pointing out the relevance of [1] and [2]. We will incorporate a detailed discussion of these works in our revised manuscript. Specifically, we will relate the "high binding" phenomenon introduced in our paper to the theoretical insights provided in [1], which explores excess capacity as a contributing factor in backdoor poisoning, and [2], which offers a statistical perspective on poisoning backdoor attacks. These references will help contextualize our contributions and clarify how our approach builds upon and extends these foundational studies.
>
> **2. Evaluation Against Detection-Based Defenses ([3,4]):**
>
> We are grateful for the reviewer’s suggestions to evaluate our method under the detection-based defenses proposed in [3] and [4]. Our work primarily focuses on evading model-level defenses, as we aim to minimize computational resource requirements. Detection-based defenses, while effective, often involve significant computational overhead and are not the primary target of our research. However, to address this concern, we have added the latest results from a strong model defense proposed in NeurIPS 2024 [Global Response 1], demonstrating that our method can still penetrate such defenses. The results highlight that our method maintains robustness even against state-of-the-art defenses.
>
> Thank you again for your thoughtful comments and suggestions.

---

### Author Response · Authors · 2024-11-20
**Global Response 1**

We have incorporated the evaluation results of a recently published backdoor defense method, Feature Shift Tuning (FST) [1], from NeurIPS 2024 into our experiments. As shown in Table 1, our proposed K&L method can effectively penetrate the FST defense mechanism. On CIFAR-10, the K&L method retains a high benign accuracy (BA) while achieving a high attack success rate (ASR) on the defended model. On more complex datasets such as CIFAR-100 and Tiny ImageNet, although the ASR of K&L drops to zero, the BA also decreases to an unusable level. This indicates that the defense mechanism becomes ineffective as it compromises the model's usability.


| Dataset     | Before Defense | Before Defense | After Defense | After Defense |
|-------------|----------------|----------------|---------------|---------------|
|             | BA             | ASR            | BA            | ASR           |
| CIFAR-10    | 0.8982         | 0.9998         | 0.9182        | 0.9578        |
| CIFAR-100   | 0.6248         | 0.9952         | 0.3861        | 0             |
| Tiny ImageNet | 0.4866         | 0.9997         | 0.086         | 0             |

**Reference:**

[1] Min, Rui, et al. "Towards stable backdoor purification through feature shift tuning." *Advances in Neural Information Processing Systems* 36 (2024).

---

### Meta-Review · Area_Chair_zmLD · 2024-12-20

**Metareview:**

This work has proposed a new backdoor attack method called Key-Locks, inspired by the high binding observation in existing backdoor attacks.

It received 4 detailed reviews. The effort of analyzing existing backdoor attacks is appreciated by most reviewers. Meanwhile, there are several important concerns, mainly including:
1. Unclear definition of high binding, which is the core concept in the motivation.
2. Difference with adversarial examples and adversarial training.
3. Similarity between high binding and the similar concepts in some existing works.
4. Several advanced defense methods are missing.
5. The contributions are over-claimed, as similar issues have be studied or summarized in existing works.

There are several rounds of discussions between authors and reviewers. I carefully read the manuscript and all discussions. My judgements of about concerns as follows:
1. It is true that the high binding is not clearly defined, neither the manuscript nor the rebuttal. Without the clear definition, it is difficult to support the follow-up analysis, and difficult to verify that the proposed method can solve that issue.
2. The authors realized the similarity to adversarial examples, and tried to distinguish them in the manuscript and rebuttal. However, the explanations are mainly from the definition, goal, rather than technical perspectives. And, the doubt that the 10% increase of ASR is due to the reduced robustness during the lock embedding stage is not well addressed. I believe that there are differences with adversarial attacks but the authors didn't provide clear explanations.
3. The authors didn't give in-depth analysis between the connections and differences between high binding and existing similar concepts in the rebuttal.
4. The authors didn't fully follow the reviewers' suggestions to add some advanced defenses. Considering the long rebuttal period, it is strange. Since the effectiveness of the proposed method is not well analyzed, its performance against advanced defenses is questionable.
5. The authors listed 3 main contributions. I think the first and third contributions are minor. The requirements of backdoor attacks have been summarized by many existing works. Proposing a new metric is not very important, especially with sufficient analysis about its necessity.

Moreover, as mentioned by reviewers, several hypothesis are lack of solid support or verification. I agree with this comment. My overall feeling of the manuscript and the rebuttal is that lots of important claims are not well verified.

Thus, my recommendation is reject.

**Additional Comments On Reviewer Discussion:**

The rebuttal and discussions, as well as their influences in the decision, have been summarized in the above metareview.

---

### Decision · Program_Chairs · 2025-01-22

Reject